

# Projected future changes in extreme precipitation over China under stratospheric aerosol intervention

Ou Wang[1], Ju Liang[2], Yuchen Gu[3], Jim M. Haywood[4,5*], Ying Chen[6], Chenwei Fang[7], Qin`geng Wang[1*]

[1]State Key Laboratory of Pollution Control and Resources Reuse, School of the Environment, Nanjing University, Nanjing, 210023, China
[2]Department of Agricultural Meteorology, Colloege of Resources and Environmental Sciences, China Agriculture University, Beijing 100193, China
[3]Department of Earth Science, Mathematical and Physical Sciences, University College London, London WC1E 6BT, UK
[4]Dept of Mathematics, School of Environment, Science and the Economy, University of Exeter, Exeter EX4 4QE, UK
[5]Met Office Hadley Centre, Exeter EX1 3PB, UK
[6]School of Geography Earth and Environment Sciences, University of Birmingham, Birmingham B15 2TT, UK
[7]Key Laboratory of Meteorological Disaster, Ministry of Education (KLME), Joint International Research Laboratory of Climate and Environment Change (ILCEC), Collaborative Innovation Center on Forecast and Evaluation of Meteorological Disasters, Key Laboratory for Aerosol-Cloud-Precipitation of China Meteorological Administration, Nanjing University of Information Science & Technology, Nanjing, 210044, China

*Correspondence to*: Ou Wang (dg1925035@smail.nju.edu.cn)

**Abstract.** Extreme precipitation events are linked to severe economic losses and casualties in China every year; hence, exploring the potential mitigation strategies to minimize these events and their changes in frequency and intensity under global warming is of importance, particularly for the populous subregions. In addition to global warming scenarios, this study examines the effects of the potential deployment of stratospheric aerosol injection (SAI) on hydrological extremes in China based on the SAI simulations (G6sulfur) of the Geoengineering Model Intercomparison Project (GeoMIP) from UKESM1 (The UK Earth System Model) simulations. The simulated SAI deployment is compared with simulations of the future climate under two different emission scenarios (SSP5-8.5 and SSP2-4.5) and reduction in the solar constant (G6solar) to understand the effect of SAI on extreme precipitation patterns. The results show that, under future global warming scenarios, precipitation and extreme wet climate events are projected to increase by 2100 relative to the present day across all the subregions in China. Additionally, analyses of extreme drought events show a projected increase in southern China. The G6sulfur and G6solar experiments ameliorate the increases in extreme rainfall intensities, especially for the eastern subregions of China. The impacts of SAI in decreasing extreme precipitation events and in consecutive wet days are more pronounced than in G6solar. While the results from both G6sulfur and G6solar show encouraging abatement of many of the impacts on detrimental extreme events that are evident in SSP5-8.5 there are some exceptions. Both G6sulfur and G6solar show drying trends at high latitudes within the region, which is consistent with our understanding of the spin-down of the hydrological cycle under SRM. For instance, the projected dry days increase for G6sulfur compared to SSP5-8.5. These side effects imply that a cautionary approach and further optimization may be required should any future SRM deployment be considered.



## 1 Introduction

China, as a country that hosts the world's second largest population, is acutely vulnerable to extreme hydrological events caused by climate change. For example, climate change can cause sea-level rise which could significantly impact flooding hazards for coastal cities in China, and flooding events in China are consistently predicted to increase under the influence of rising GHG (high-level greenhouse gas) emissions. For example, studies show that precipitation will increase across China by the end of the 21st century with previous research projecting that heavy rainfall events cause increased flooding in the 21st century (Yang et al., 2021). In addition to an increase in extreme precipitation, it has also been suggested that the most populated south-eastern China would experience higher flood hazard risks across all future periods (2016-2035, 2046-2065, and 2080-2099) (Ying et al., 2014). Extreme precipitation events appear to have impacted China more often in recent years; the summer of 2020 was anomalous, with flooding in southern, eastern, and parts of central China (Jia et al., 2022). In the summer of 2021, unprecedented rainfall hit Zhengzhou, the capital and central city of Henan province, causing severe flooding with a rainfall intensity of over 200 mm per hour (Zhao et al., 2021). The leading cause of the heavy rainfall was Typhoon In-fa and continuous subtropical high pressure (Jackson and Shawn, 2021). Strong low-level easterly or south-easterly jets developed between the western Pacific subtropical high (WPSH) and Typhoon In-Fa (Rao et al., 2023) bringing a large amount of moisture from the ocean to Henan Province that caused heavy rainfall (Deng et al., 2022). The precipitation further accumulated, exceeding 550 mm within 24 hours. It has been designated as the '7.20 Henan extreme urban flooding event' (Dong et al., 2022). In 2022, although southern China continued to suffer from extreme precipitation, the flooding risk has been centred to the west of the Yangtze River, including provinces such as Sichuan and Qinghai which have experienced sudden heavy rainfall events. Although not statistically robust, these events might tentatively suggest a potential expansion of the regions that could be influenced by increasing precipitation in China under the changing climate. Commencing on July 29, 2023, a record-breaking episode of rainfall and flooding unfolded across at least 16 cities and provinces in north-eastern China. Notably, Beijing encountered its most substantial precipitation event in 140 years, with the accumulated rainfall exceeding 60% of the region's typical annual precipitation within a remarkably brief span of 83 hours (CDP, 2023). This led to flooding in cities, including Beijing, resulting in at least 60 fatalities and significant damage to homes, crops, livestock, and infrastructure (REUTERS, 2023).

On a global scale, climate change appears to have been influencing hydroclimatic conditions. The direct influence of global warming is that rising atmospheric temperatures induce stronger evapotranspiration and the atmosphere can hold more water vapour. The intensified hydrological cycle exacerbates heavy rainfall and flooding but can also contribute to further drying over land areas and prolonged drought periods (IPCC, 2021). Climate-change-induced faster evaporation causes and higher atmospheric temperatures induce more moisture-laden air in the storm tracks. Consequently, in general, wet areas become wetter while dry areas become drier (Held and Soden, 2006). Extreme weather events including droughts and flooding could be worsened by global warming. A global-scale study indicates that global warming will potentially increase drought severity



as well as drought frequency in the future (Qi et al., 2022). Flooding events also occur at higher frequency and intensity under extreme precipitation amplification (Tabari, 2020). Weather and climate disasters such as extreme temperatures and severe
snowstorms, have caused serious economic losses in densely populated East Asian countries (Huang et al., 2007; Li et al., 2016). An increase in forecast precipitation in current climate models, particularly over the populated areas in East Asia, such as China (Liang and Haywood., 2023) implies strategies are urgently needed to mitigate changes in hydrological extremes.

Owing to the difficulties in achieving climate targets such as the 1.5°C or 2°C above pre-industrial levels targets proposed by the Paris Conference of Parties (IPCC, 2018), Solar Radiation Modification (SRM) proposals, i.e. strategies to mitigate the
worst impacts of climate change by brightening the planet have been studied (Haywood et al., 2022). To understand the robust climate model responses to geoengineering, the Geoengineering Model Intercomparison Project (GeoMIP) was established to provide a comprehensive multi-model assessment of the effects of SRM (Kravitz et al., 2013; Kravitz et al., 2011). Modelling SAI geoengineering (e.g. G6sulfur experiment) is one of the most prominent SRM strategies within the GeoMIP suite of recent simulations (Visioni et al., 2022). SAI mitigates anthropogenic climate warming by injecting reflective particles, or their
gaseous precursors, into the stratosphere. The resultant aerosols reflect and scatter solar radiation back into space leading to a cooling that counterbalances the warming from increased concentrations of greenhouse gases (e.g. Stenchikov et al., 1998). Such simulations mimic the explosive 1991 eruption of Mount Pinatubo in the Philippines; this volcanic eruption created a layer of stratospheric aerosol that induced a cooling of global average surface temperature by around 0.5 degrees Celsius for around two years (Bluth et al., 1992; Self et al., 1996; Robock, 2000). There have been other smaller eruptions since then that
have also been modelled to exert a cooling influence, the climate (Soden et al., 2002; Haywood et al., 2014; Schmidt et al., 2018). In addition to reducing the temperature, SAI also influences tropospheric and stratospheric ozone, terrestrial ecosystem, terrestrial carbon, and hydrological cycle by changing the physical climate system and atmospheric chemistry (Liang and Haywood., 2023; Jones et al., 2018; Jones et al., 2020; Cao, 2018; Plazzotta et al., 2019; Lee et al., 2021; Visioni et al., 2022; Imai et al., 2020; Mclandress et al., 2011).

The latest phase of policy-relevant GeoMIP simulations (GeoMIP6) proposed two new experiments, G6sulfur and G6solar (Kravitz et al., 2015), which are designed to simulate the influence of SAI and solar constant reduction to the end of the 21st century based on predicted future emission pathways (Shared Socioeconomic Pathway; SSPs). G6sulfur and G6solar aim to lower global mean surface temperatures from a high-tier emission scenario (SSP5-8.5; (Meinshausen et al., 2020)) to a medium-tier emission scenario (SSP2-4.5). These SSP scenarios are developed by the Coupled Model Intercomparison Project
Phase 6 (CMIP6; Eyring et al., 2016), which provides multi-model climate predictions based on alternative scenarios of future emissions and land use changes produced by integrated assessment models (O'neill et al., 2016). Studies such as Jones et al. (2021) and Ji et al. (2018) included detailed descriptions and explanations of the CMIP6, GeoMIP, and the differences in models' assumptions. SAI will exert a negative radiative forcing and reduce near-surface air temperature (including temperature means and extremes) (Pinto et al., 2020), and precipitation (Liu et al., 2021). To date, only a few studies have





concentrated on the impact of SAI on the future changes in weather systems over East Asia (Liang and Haywood., 2023; Liu et al., 2023).

In this study, we analyse the projected change of precipitation over China in the 21st century, particularly focused on the effects of G6sulfur and G6solar on hydrological extremes, using the UKESM1 model and CMIP6 experiments. Our study explores the change in frequency and intensity of extreme precipitation between future GHG emissions and solar
geoengineering scenarios. We focus on the period towards the end of this century, 2070-2100, to maximize the signal-to-noise in the simulations as the SAI injection rates increase throughout the century. Section 2 describes the experimental design and details of the model used in this study. In Section 3, results are presented to show the changes in extreme indices in China both under the use of SAI (G6sulfur) and solar constant reduction (G6solar) compared to present-day and climate change scenarios. The regional analysis of China's extreme precipitation and cumulative distribution function is also provided in Section 3.
Section 4 summarises and discusses the findings.

## 2. Data and methods

### 2.1 Study area

To quantitatively examine regional differences and better visualize the future extreme climate features, China is divided into 7 different sub-regions (Fig.1) according to distinguish extreme climates across the following regions: Northeast China (NEC),
North China (NC), Northwest China (NWC), Centre China (CC), East China (EC), South China (SC), and Southwest China (SWC) following Liang et al. (2023).

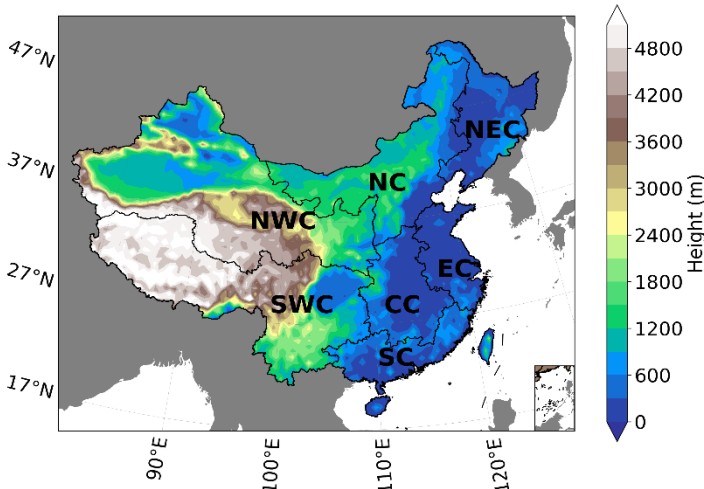

**Figure 1: Geological map of elevation and 7 sub-regions in China (unit: m), data from (Liang et al., 2023)**



## 2.2 UKSEM1 model and model simulations

In this study, data from G6sulfur and G6solar experiments are used from the sixth phase of GeoMIP from the U.K. Earth
System Model UKESM1 (Sellar et al., 2019). UKESM1 is a fully coupled Earth system model with an atmospheric resolution
of $1.25°$ latitude by $1.875°$ longitude (Storkey et al., 2018; Walters et al., 2019; Mulcahy et al., 2018, Sellar et al., 2019), and
contributes to both CMIP6 and GeoMIP6 (Jones et al., 2020). The Scenario MIP high GHG forcing scenario SSP5-8.5 (O'neill
et al., 2016) is used as the baseline scenario of both G6solar and G6sulfur experiments (Kravitz et al., 2015). The G6sulfur

experiment injects $SO_2$ into the stratosphere to adjust the global mean surface temperature from SSP5-8.5 to that of the SSP2-
4.5 medium-forcing scenario over the period 2020-2100 (Haywood et al., 2022). The GeoMIP G6sulfur simulations that reduce
global mean temperatures from the SSP5-8.5 scenario to the SSP2-4.5 scenario are described in detail elsewhere (Kravitz et
al., 2015). UKESM1 simulates $SO_2$ injection in the stratosphere along the Greenwich meridian at an altitude of 18-20 km
between 10° N and 10° S in G6sulfur (Kravitz et al., 2021). UKESM1 contains the sophisticated United Kingdom Chemistry

and Aerosols (UKCA) module that represents the sulphur cycle in the troposphere and stratosphere (Archibald et al., 2020). A
parallel experiment to G6sulfur, the G6solar experiment, reduces ScenarioMIP Tier 1 high forcing scenario to the medium
forcing scenario by reducing solar irradiance (Kravitz et al., 2015). Reductions in the global mean near-surface
temperature,from those of SSP5-8.5 to those of SSP2-4.5 are achieved by reducing the solar constantor increasing SAI by trial
and error to within a tolerance of ±0.2 K (Visioni et al., 2021). Notably, it is anticipated that G6solar will exhibit reduced inter-

model disparities in the spatial distribution of forcing when compared to G6sulfur owing to model differences in representing
the complexities of the sulfur cycle within global models. Therefore, G6solar is proposed as a parallel experiment to G6sulfur
for the purpose of comparing the impacts of solar reduction with those of stratospheric aerosols. (Kravitz et al., 2015).

In the UKESM1 model, three ensemble members, 'r1i1p1f2', 'r4i1p1f2', and 'r8i1p1f2', are run for G6sulfur and G6solar as
specified in the GeoMIP protocol (Kravitz et al., 2015). We calculated the ensemble mean for all simulations, including SSP5-

8.5, SSP2-4.5, G6sulfur, G6solar and Historical. The future changes in extreme climates are assessed by comparing the future
simulations (SSP5-8.5, SSP2-4.5, G6sulfur, G6solar) for the period 2071-2100 with the present-day period 1981-2010
(Historical), using the UKESM1 historical simulations for CMIP6. The results for the final 30 years of the 21st century (2071-
2100) from the simulations are used to investigate the influences of G6sulfur and G6solar. The rationale for using the present-
day period as a baseline for comparison is that the model performance for the present-day can be assessed against observed

climate metrics to provide guidance of model fidelity as described in the following subsections.

## 2.3 Aphrodite precipitation data

The APHRODITE (Asian Precipitation-Highly-Resolved Observational Data Integration Towards Evaluation) is interpolated
from gauge-observation data with a resolution of $0.25° × 0.25°$ (Lai et al., 2020; Yatagai et al., 2012), and are used to validate
the performance of the UKESM1 model in simulating present-day extremes. APHRODITE is a dataset containing long-term



gridded daily precipitation (1951-2015). The high-resolution daily product of APHRODITE is developed based on the rain-gauge data across Asia presented on a continental scale (Sunilkumar et al., 2019). In this study, we apply APHRODITE's climatological daily mean precipitation as the observation data and validate the UKEMS1 historical simulations.

## 2.4 Extremes precipitation indices

The results were assessed according to the IPCC defined extreme precipitation indices to quantify the precipitation extreme
responses in each experiment. We selected eight extreme indices defined by the Expert Team on Climate Change Detection and Indices (ETCCDI) as shown in Table 1 to perform the analysis on all of the simulations.

**Table 1: The definition of selected extreme indices based on ETCCDI (Firch et al, 2002; Tank et al., 2002)**

| Indices | Descriptive name | Definition | Units | Type |
|---------|------------------|------------|-------|------|
| DD | Dry days | Count of days when precipitation<1 mm | days | Fixed Threshold |
| CDD | Consecutive dry days | Maximum number of consecutive days with < 1 mm of precipitation | days | Fixed Index/ Spell |
| CWD | Consecutive wet days | Maximum number of consecutive days with ≥ 1 mm of precipitation | days | Fixed Index |
| R50MM | Rainstorm days | Count of days when precipitation≥50 mm | days | Fixed Threshold |
| RX1DAY | Maximum 1-day precipitation | Annual maximum 1-day precipitation | mm | Fixed Index |
| RX5DAY | Maximum 5-day precipitation | Annual maximum 5-day precipitation | mm | Fixed Index/Spell |
| R95p | Precipitation due to very wet days | Annual precipitation amount accumulated on days when daily precipitation is greater than the 95th percentile threshold of the wet-day precipitation | mm | Percentile-based Threshold |

## 2.5 Statistical methods and Cumulative Distribution Function (CDFs)

In the validation section comparing the experiment's ensemble mean with observed Aphrodite data, the ensemble mean was re-gridded to the resolution of the Aphrodite data after averaging.



This study simulates the changes in precipitation between different experiments in terms of annual mean differences. To examine the statistical importance of the results, we performed the Wilcoxon Rank Sum Test instead of the more commonly used Student's *t* test. Wilcoxon Rank Sum Tests works as a non-parametric two-sample t-test and is more appropriate for use with atmospheric data (Wilks, 2011), with p-value < 0.05 suggesting statistical significance.

To better visualize the future extreme climate features and the effects of SAI, the Cumulative Distribution Functions (CDFs) for rainfall have been calculated. CDFs is frequently employed for bias correction to enhance the accuracy of rainfall analysis (e.g., Apurv et al., 2015; Rana et al., 2014; Xiong et al., 2019). This decision accounts for the mixed nature of our data, as extreme precipitation can be viewed from both continuous and discrete perspectives. While Probability Density Functions (PDFs) are conventionally used for continuous variables (Vidhya, 2023), our focus on annual extreme precipitation, often treated as discrete events, aligns well with the suitability of CDFs. This approach allows for a nuanced exploration of the distribution, accommodating the continuous and discrete aspects of our dataset. The calculated CDFs offer a holistic perspective, providing insights into the probability distribution patterns for various events over the study period.

Unlike Tung et al. (2022), where CDFs were employed, a choice was made to use reversed CDFs in our study to better illustrate the thresholds for rainfall events exceeding certain values. To achieve this, during the continuous 30-year study period, we computed the average annual extreme precipitation index values for each grid point and plotted their CDFs. This analysis facilitated the observation of continuous probability distribution patterns and the assessment of tail-end magnitudes, providing insights into the continuous likelihood of varying precipitation levels and revealing extremes throughout the studied period.



## 3. Results

In order to comprehensively assess the statistical significance of our findings, a 'field significance' analysis based on the method proposed by Wilks (2006) was employed. This approach allows us to evaluate the significance in regions without observed data points. The calculations were conducted using the following Eq. (1):

$$\Pr(M \geq m) = \int_{i=m}^{K} \frac{K!}{i!(K-i)!} \cdot (\alpha_{local})^i \cdot (1 - \alpha_{local})^{K-i} \tag{1}$$


where 'm' represents the number of observed data points identified as statistically significant through significance testing, 'K' is the total number of grid points, and 'α_local' is the local significance level.

According to the calculation using this formula, all non-dotted regions exhibit statistical significance. However, detailed explanations of this process are omitted for brevity in this paper.

### 3.1. Precipitation changes

The spatial pattern of mean precipitation bias between simulated present-day (PD; mean of 1981-2010) data and observed Aphrodite data of 30 years is shown in Fig.2. The PD data is derived from the UKESM1 historical simulations. Evaluation of annual mean land precipitation simulations of UKESM1 model (PD) (Fig.2b) against observed data (Obs) (Fig.2a) reverals that, while the general pattern of simulated precipitation was accurate, the amount of precipitation tends to be somewhat larger

than that observed over much of China. The general decrease from the southeast coastal regions to the northwest inland areas is well simulated. The wet bias in daily precipitation is evident in most parts of southern China (SC, CC, and SWC), particularly on the south-eastern flanks of the Qinghai-Tibet Plateau (QTP). Regions with dry biases are relatively small (shown by the negative values and brown contours in Fig.2c), with bias values less than 1mm/day.

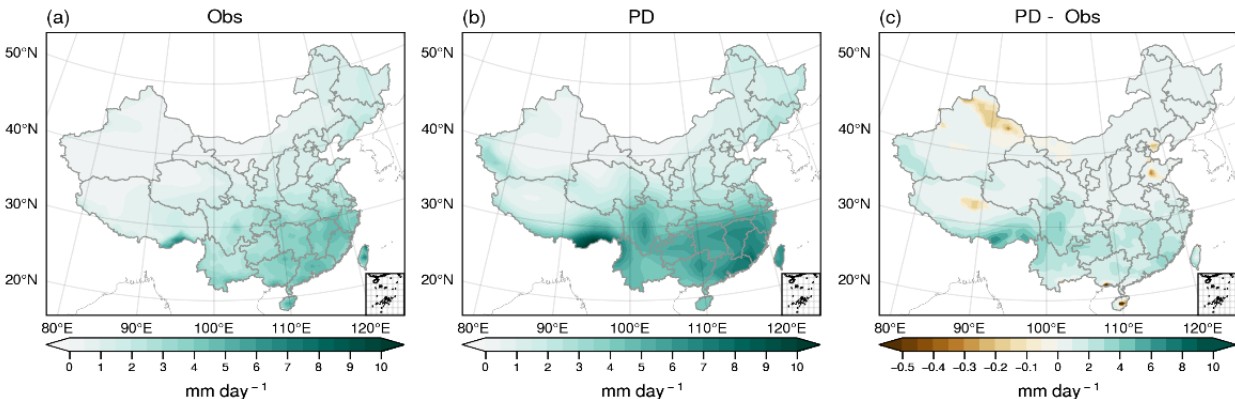




**Figure 2. Spatial distributions of mean land precipitation (units: mm day⁻¹) over China during the period of 1981-2010 from (a) Obs (observed Aphrodite data), (b) PD data (simulated multi-ensembles historical data) and (c) the bias between PD and observed.**

Figure 3 illustrates the spatial distribution of mean precipitation changes in 'mm/day' units over land areas for 2071-2100 compared with PD for UKESM1. Clearly, most regions experience precipitation increases under all four potential future scenarios. Simulations under SSP5-8.5 scenario project a significant precipitation increase in southern China except in Hainan and Taiwan province (Fig.3a), but the precipitation in SC, northern Taiwan and Hainan is projected to increase in the future. Similar patterns are noted under SSP2-4.5 (Fig.3b) while the magnitude of increases in precipitation is reduced by about a half when compared to SSP5-8.5. Both G6sulfur (Fig.3c) and G6solar (Fig.3d) (SRM) show ameliorated change with respect to SSP5-8.5, meaning that the SRM mitigate the increase in precipitation in China that occurs in SSP5-8.5.

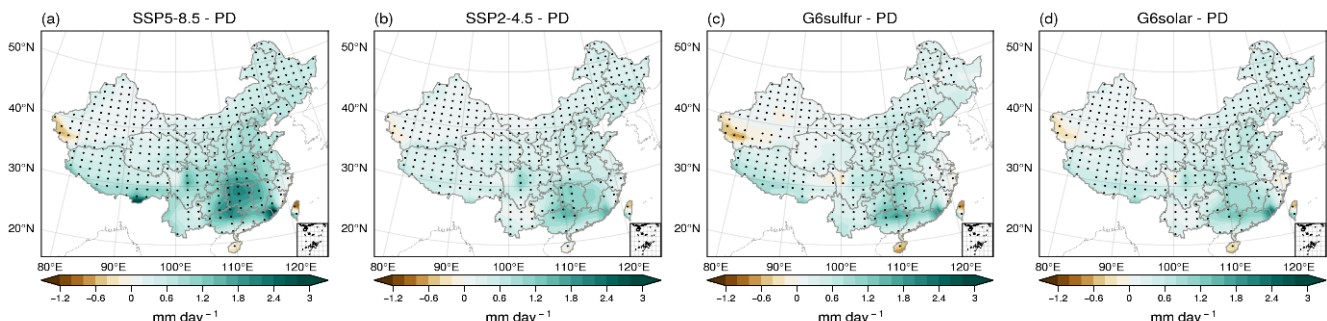

**Figure 3. Absolute change in land precipitation (mm day⁻¹) for the period 2071-2100 relative to the PD (a) SSP5-8.5, (b) SSP2-4.5, (c) G6sulfur, and (d) G6solar. The PD data comprise years 1981-2010 from the UKESM1 historical simulations. The dotted areas indicate where the difference is statistically significant at 95% confidence level using a Wilcoxon rank sum test.**

Figure S1 compares the simulated future precipitation between G6sulfur and other experiments. Compared with SSP5-8.5 (Fig. S1(a)), the simulated SAI by G6sulfur leads to a decrease in precipitation over almost the entire China. This suggests that SAI is sensitive to global warming in China, particularly over the SWC (southeast QTP) and CC regions. The difference in precipitation between G6sulfur and SSP2–4.5 (Fig. S1(b))/G6solar (Fig. S1(c)) is smaller compared to the difference between G6sulfur and SSP5-8.5 (Fig. S1(a)). This indicates SAI effectively mitigates the increase in mean precipitation from the high GHG SSP5-8.5 scenario to the medium GHG SSP2-4.5 scenario across most of China.



## 3.2. Hydrological extreme changes

In this section, we examine impacts on hydrological extreme indices changes by presenting geographic maps of the chosen variables. As extreme rainfall events are associated with the tail of the CDFs, the tails of the CDFs have been focused on by adjusting the vertical axis to range from 0% to 10%, and the Aphrodite data has been excluded for clarity.

### 3.2.1 Wet extreme changes

The small-scale flooding risk is now being assessed by the RX1day index, and the extreme threshold index of very wet day precipitation represented by the R95p is being used. The ensemble mean of absolute changes in RX1day and R95p for the future period (2071-2100) relative to the historical baseline period (1981-2010) are shown in Fig.4. Simulations under the SSP5-8.5 scenario project significant increases ($p$-value $< 0.05$) in RX1day in east China (Fig.4a). The greatest magnitude of increase (above 50 mm) is seen in SC, CC, east coastal NC and a small part of SWC regions. Under SSP2-4.5 (Fig.4b), a similar pattern of the RX1day change to SSP5-8.5 is projected but with smaller magnitudes. G6sulfur (Fig.4c) and G6solar (Fig.4d) show generally ameliorated changes compared to SSP5-8.5.

In the future, an increase in RX5day is anticipated across most of China, with the most substantial increments occurring in the eastern part of the country and on the QTP (Fig. S2a-d). The largest increases are anticipated under the SSP2-8.5 scenario (Fig.S2a), reaching a maximum of over 100 mm. In the other three G6 models, the rise in RX5day is considerably smaller under SSP5-8.5, with none exceeding 100 mm. This suggests a mitigated future RX5day simulation compared to SSP5-8.5 in these three models. It is noteworthy that under G6solar (Fig.S2d), the maximum RX5day is observed in the south-eastern part of the SC region.

R95p is projected to significantly increase ($p$-value $< 0.05$) in CC, SC, and south SWC regions under SSP5-8.5 scenario (Fig.4e) which is generally consistent with a previous study (Wang et al., 2016). This increase has been attributed to the strengthened south-westerly winds across south China caused by the land-sea contrast between China and adjacent oceans. Furthermore, global warming also contributes to the increased water vapor, thereby enhancing the likelihood of precipitation and related extremes (Tang et al., 2021). The SSP2-4.5, G6sulfur and G6solar experiments present similar spatial distributions, but smaller magnitudes of changes (Fig.4f-h). G6sulfur (Fig.4g) shows a significant decrease in highly populated areas of Hainan (south SC) compared to SSP5-8.5. This suggests that SAI SRM appears to effectively mitigate the R95p increase relative to the high SSP5-8.5 scenario in this area, although we caution over-interpretation of this result over such a small geographical area.





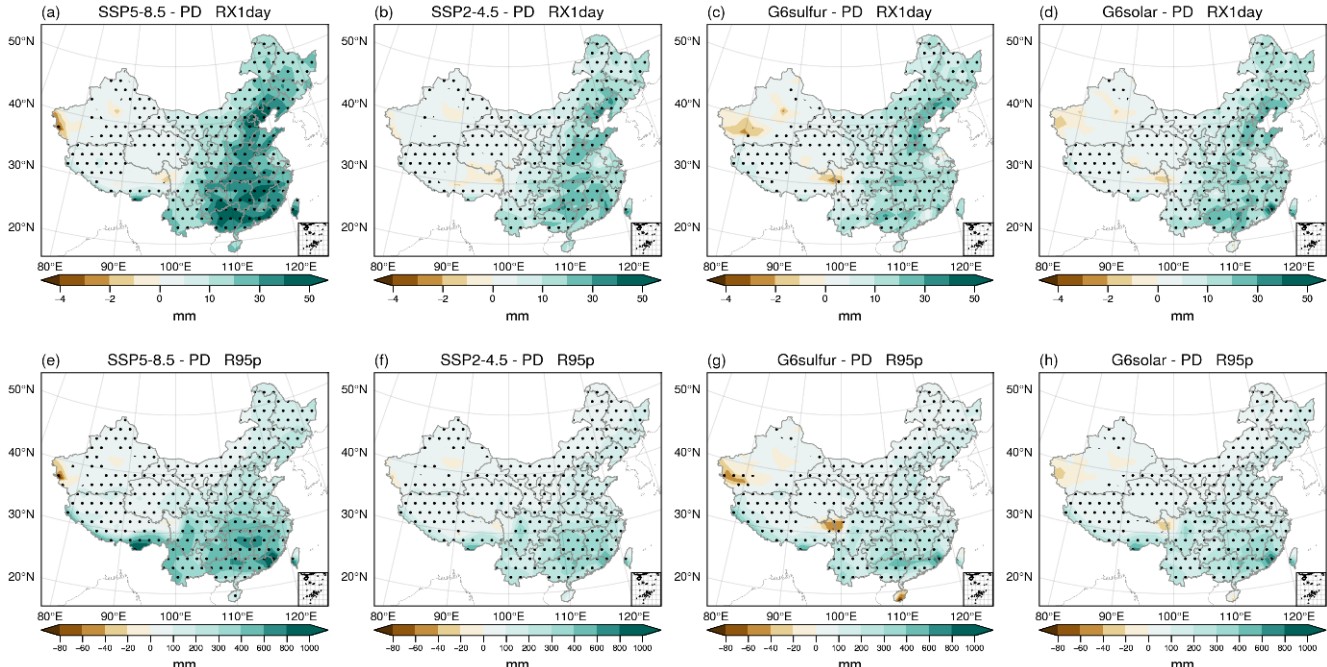


**Figure 4. Absolute changes in RX1Day (a-d) and R95p (e-h) for the future period of 2071 – 2100 relative to the PD. The dotted areas indicate where the difference is statistically significant at 95% confidence level using a Wilcoxon rank sum test.**

Figure 5 illustrates how SAI modulates the distribution of extreme precipitation intensity indices by dicipting differences between G6sulfur and other experiments. The comparisons confirm that G6sulfur can effectively ameliorate the increase in the intensity indices in eastern China (shown by the negative brown shaded areas in Fig.5 a and d). A previous study demonstrated that the reduction of emissions of anthropogenic aerosols is a key factor leading to increased extreme intensity indices over most of China by the end of 21st century (Wang et al., 2016). Compared with SSP2-4.5 (Fig.5b) and G6solar

(Fig.5c), G6sulfur leads to only small changes of less than 20mm in RX1day. The similarity of G6sulfur to SSP2-4.5 and G6solar suggests that the primary impact on RX1day over China is driven simply by the temperature; a global mean temperature of the standard CMIP6 SSP2-4.5 scenario gives very similar results to those achieved when SSP5-8.5 temperatures are brought down to those of SSP2-4.5. However, the same cannot be said for R95p, where the ensemble mean of UKESMI projects a significant increase in SC and southwestern SWC (Fig.5e) for the G6sulfur (2071-2100) relative to the SSP2-4.5

(2071-2100), and in north SC, eastern, and southwestern SWC relative to G6solar (Fig.5f).

From Figure S3a, it is evident that G6sulfur effectively mitigates RX5day under SSP5-8.5, particularly in the eastern and south-western regions. The most notable impact is observed in the SC region and the south QTP, with a mitigation of up to 80mm. This suggests that G6sulfur can efficiently mitigate future flood risks in these areas. In comparison to SSP2-4.5



(Fig.S3b), G6sulfur exhibits an increase in RX5day, primarily in the region between 100°E and 120°E. This implies that SSP2-4.5 performs better than G6sulfur in these areas compared to the mitigation effect under SSP5-8.5. For 'G6sulfur-G6solar' (Fig.S3c), positive values of RX5day are more pronounced in certain areas between 100°E and 120°E, especially in the low latitude zone between 20°N and 30°N. This indicates that G6solar achieves the most effective mitigation of RX5day under SSP5-8.5.

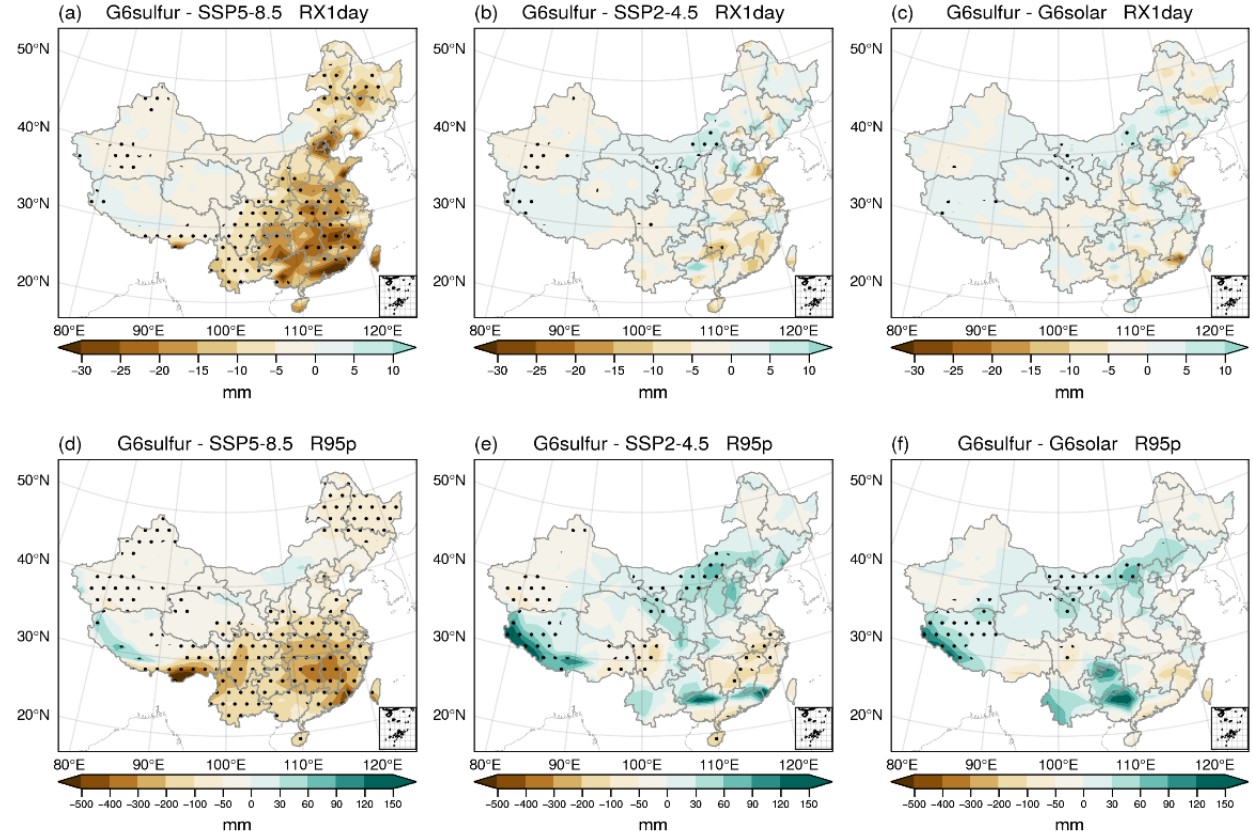


**Figure 5. Differences in RX1Day (a-c) and R95p (d-f) for the future period of 2071 – 2100 between G6sulfur and SSP5-8.5 (a, d), SSP2-4.5 (b, e), and G6solar (c, f). The dotted areas indicate where the difference is statistically significant at a 95% confidence level using a Wilcoxon rank sum test.**


To better illustrate the ameliorating impact of SAI on extreme precipitation under the SSP5-8.5 scenario, the differences between the maximum values of each index under G6sulfur and the SSP5-8.5 scenario, compared to the maximum values under the PD, were aggregated. This process aims to emphasize the mitigation effect of SAI on extreme high values of each index under the SSP5-8.5 scenario. As outlined in Table 2, positive values indicate mitigation, while negative values signify

the opposite. G6solar's ameliorating impact under the SSP5-8.5 scenario has been presented in Supplementary Table S1 (In calculations involving decimal fractions, rounding to the nearest value was applied).



**Table 2: Amelioration effect of G6sulfur in comparison to SSP5-8.5.**

|  | China | NEC | NC | NWC | EC | CC | SC | SWC |
|---|---|---|---|---|---|---|---|---|
| **RX1day** | + | + | + | + | + | + | + | + |
| **RX5day** | + | + | + | + | + | + | + | + |
| **R50mm** | + | **0** | + | **0** | + | + | + | + |
| **CWD** | - | - | + | + | - | - | - | - |
| **R95p** | + | + | + | + | + | + | + | + |
| **DD** | - | - | - | - | - | - | + | - |
| **CDD** | - | - | - | - | - | - | + | + |

RX1day (Fig.6a) and R95p (Fig.6b) show consistent increases in the future relative to PD (the blue line). For RX1day the CDFs for precipitation SSP5-8.5 surpasses those from all other scenarios as might be expected from the spatial analyses presented in Fig.4. Additionally, southeastern China (EC, CC, and SC) shows higher values than the northern and western inland regions (NEC, NC, NWC).

The tail of the RX1day CDFs in experiment G6sulfur (purple) is close to that of the SSP2-4.5 scenario (green) in NEC, NC, NWC, and SWC (Fig.6). Combined with the small and evenly distributed magnitudes shown in Fig.5b, this shows that G6sulfur effectively mitigates SSP5-8.5 to similar the SSP2-4.5 in these regions. In EC, SC and CC, the RX1day CDFs is reduced from that of SSP2-4.5 by between 5-10mm and moves further away from the values seen in the high-end SSP5-8.5 scenarios towards those seen in the historical simulations. The RX1day CDFs for G6solar are indistinguishable from those for SSP2-4.5 in many
regions, but for CC the RX1day lies to the left of the SSP2-4.5 curve but by not as far as that for G6sulfur indicating less abatement of RX1day extremes. Interestingly, for SC, the G6solar CDFs curve lies to the right of the SSP2-4.5 curve. Combining the negative value for the SC region in Table S2 reveals that the G6solar distribution is even further from that of the PD compared to SSP5-8.5, suggesting that G6solar does not mitigate the extreme high values of the RX1day index in the SC region.


The tail of RX5day CDFs across all regions suggests a future increase in RX5day under four G6 scenarios, with a more pronounced rise under the high SSP5-8.5 scenario. This phenomenon is consistent with the spatial distribution change observed in Fig.4(a-d). In the NEC and CC regions, G6sulfur closely aligns with the SSP2-4.5 scenario (Fig. S4). Additionally, in the NC region, G6solar closely mirrors the conditions of the SSP2-4.5 scenario. Combined with the differences in RX5day of
spatial distribution in Fig.5, it is evident that G6sulfur mitigates RX5day, resembling the SSP2-4.5 scenario, in NEC and CC regions. Simultaneously, G6solar alleviates RX5day in the NC region, resembling the conditions of the SSP2-4.5 scenario.



For R95p, the CDFs of G6sulfur and SSP2-4.5 show no apparent differences for SC, SWC and NWC; however, in EC and CC, there are substantial decreases of more than 100mm, whereas in NEC and NC, there are some increases, although the values

only reach 50mm. It should be stressed here that the regions that are chosen for aggregation are somewhat arbitrary and the results could well change should smaller sub-regions be chosen for analysis. For example for the SWC region, Figure 5e reveals that R95p for G6sulfur-SSP2-4.5 shows statistically significant negative values centred at around 30°N 100°E, and statistically significant positive values west of 90°E. Because these regions are aggregated together in the SWC region, there is not a discernible influence on the CDFs of R95p. Thus, the spatial maps and the CDFs should be used in combination when

presenting quantitative results.

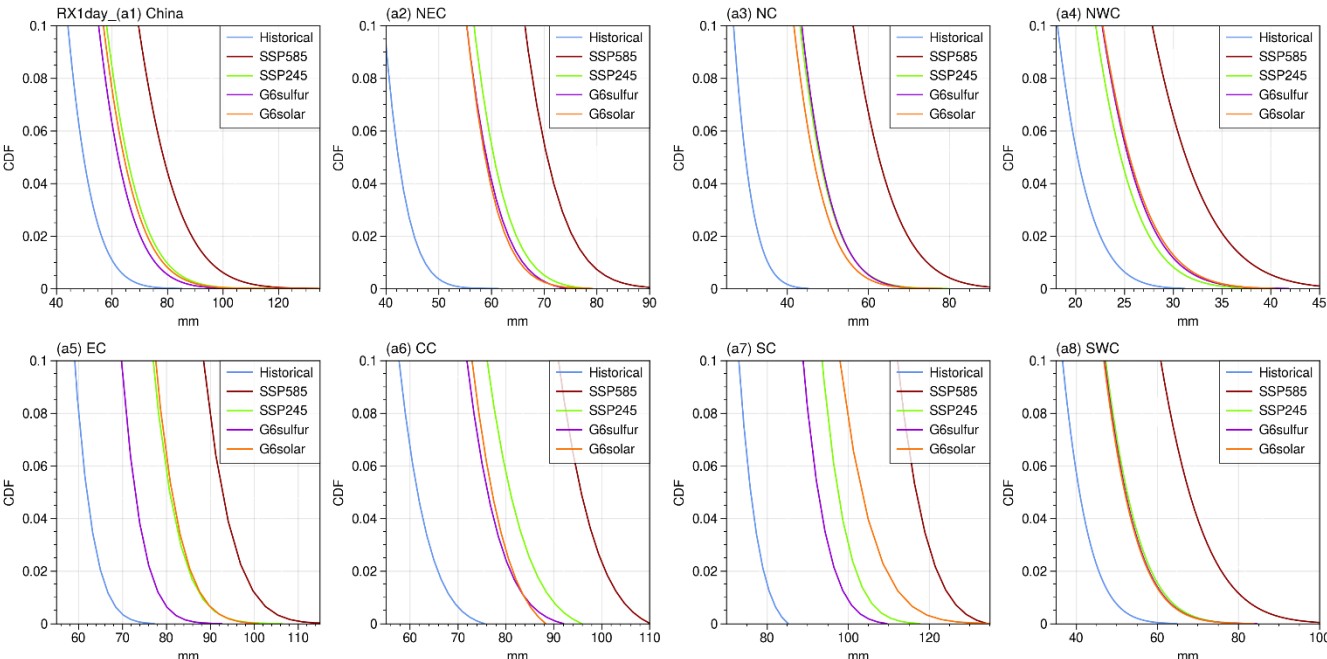



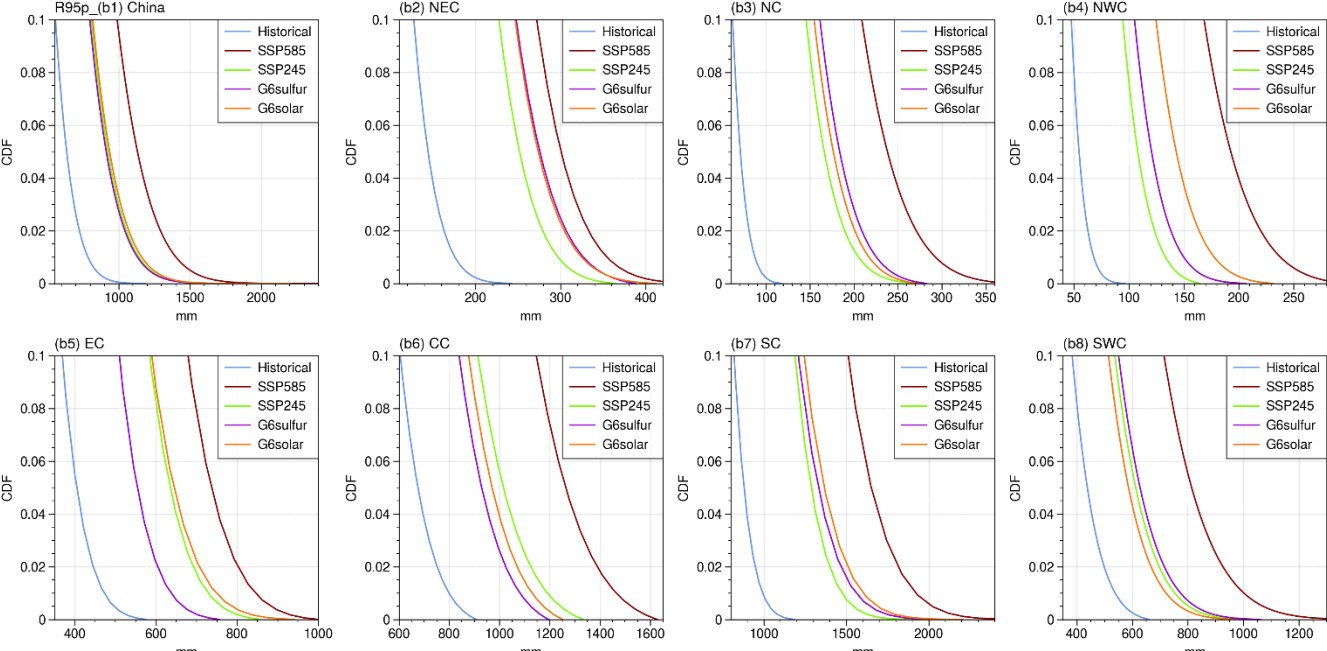

**Figure 6. Cumulative distribution functions of RX1day (a1-a8) and R95p (b1-b8) in China and 7 subregions. The same processing was applied to all CDFs figures, including the current one, to enhance clarity: image enlargement, vertical axis adjustment (0% to 10%), and exclusion of Aphrodite data.**

In summary, by the late 21st century, eastern China is projected to experience an increase in heavy rain events and a heightened risk of flooding under the high-emission SSP5-8.5 scenario, with UKESM1 simulations indicating a strengthening of both RX1day and R95p, signaling more stronger precipitation events driven by elevated GHG emissions. Under the G6 scenario, SRM effectively mitigates the increase of RX1day, RX5day and R95p compared to SSP5-8.5 scenario in all regions, particularly in east China and QTP region.

The frequency extreme index change in CWD has been calculated and shown in Fig.7. For the SSP5-8.5 scenario (2071-2100) relative to the PD (1981-2010), the ensemble mean of UKESMI predicts a significant decrease (p-value < 0.05) in southwest China (Fig.7a), particularly in the south SWC (QTP), with up to 30-day reduction. This reduction could be influenced by the East Asian and South Asian monsoons under the complex terrain of QTP (Wang et al., 2018).

Although CWD is projected to decrease under SSP5-8.5, precipitation amounts increase (Fig.3), suggesting that daily extreme precipitation intensity may rise in southern areas China in the future (Zhu et al., 2018). However, increased CWD occurs in mid-latitudes (mostly north of 30°N latitudes) but with a lesser extent (less than 20 days). A similar pattern of change is seen under SSP2-4.5, but with smaller magnitudes (Fig.7b). The experiments G6sulfur (Fig.7c) and G6solar (Fig.7d) exhibit



generally mitigated changes compared to SSP5-8.5, although the brown areas shown in Fig.7c are larger than in Fig.7a in NWC.


The absolute threshold rainstorm index, R50mm, is defined as recommended by ETCCDI, with a threshold set to daily precipitation of 50 mm by the China Meteorological Administration (CMA) (Sui et al., 2018). Under the SSP5-8.5 scenario (2071-2100) relative to the PD (1981-2010), the ensemble mean of UKESMI projects a significant increase (p-value < 0.05) in populous southern and eastern China (Fig.7e). This aligns with a prior study by Meng et al. (2021), which predicts an

increase in R50mm in the lower reaches of the Yangtze River basin and the coastal areas in SC (Meng et al., 2021), indicating a rise in rainstorm events in these regions by the end of the 21$^{st}$ century. This increase in rainstorm events contributes to an elevation in precipitation levels (as shown in Fig.3), exerting significant pressure on social economies and terrestrial ecosystems (as discussed by Peng et al., 2018). A similar pattern of change is observed under SSP2-4.5 (Fig.8f) and in the experiments G6sulfur (Fig.8g) and G6solar (Fig.8h), albeit with smaller magnitudes.


**Figure 7. Same as Figure 4, but for CWD (a-d) and R50mm (e-h).**

Compared to SSP5-8.5 (Fig.8a), G6sulfur significantly increases the CWD in south SWC (by up to 12 days), indicating sensitivity to globe warming within this region. Combined with Fig.7, G6sulfur effectively ameliorates the decrease in CWD under SSP5-8.5 in southeast QTP (approximately 90°E - 100°E). However, in southwest QTP, G6sulfur exacerbates the increase in CWD compared to high SSP5-8.5 scenario, possibly due to the complex terrain and varying altitude in QTP, linked



to the cooling effect of concentrated scattering aerosols across lower latitudes. G6sulfur primarily ameliorates the increase in CWD in mid-latitudes (north of 30°N). G6solar (Fig.8c) shows a similar magnitude in south CC and north SC as Fig.8a, suggesting that solar constant reduction does not have a significant effect in CWD compared to SSP5-8.5.

For rainstorm (R50mm), G6sulfur leads to a decrease in most part of China, with a significant decrease in south-eastern coastal areas and south QTP, up to 6 days compared to SSP5-8.5. Compared to SSP2-4.5 and G6solar, the differences are close to zero, suggesting that SRM yields nearly identical results to the SSP2-4.5 scenario. In all, SRM effectively ameliorates R50mm under SSP5-8.5 and provides statistically similar outcomes to SSP2-4.5 in China by the end of the 21$^{st}$ century.

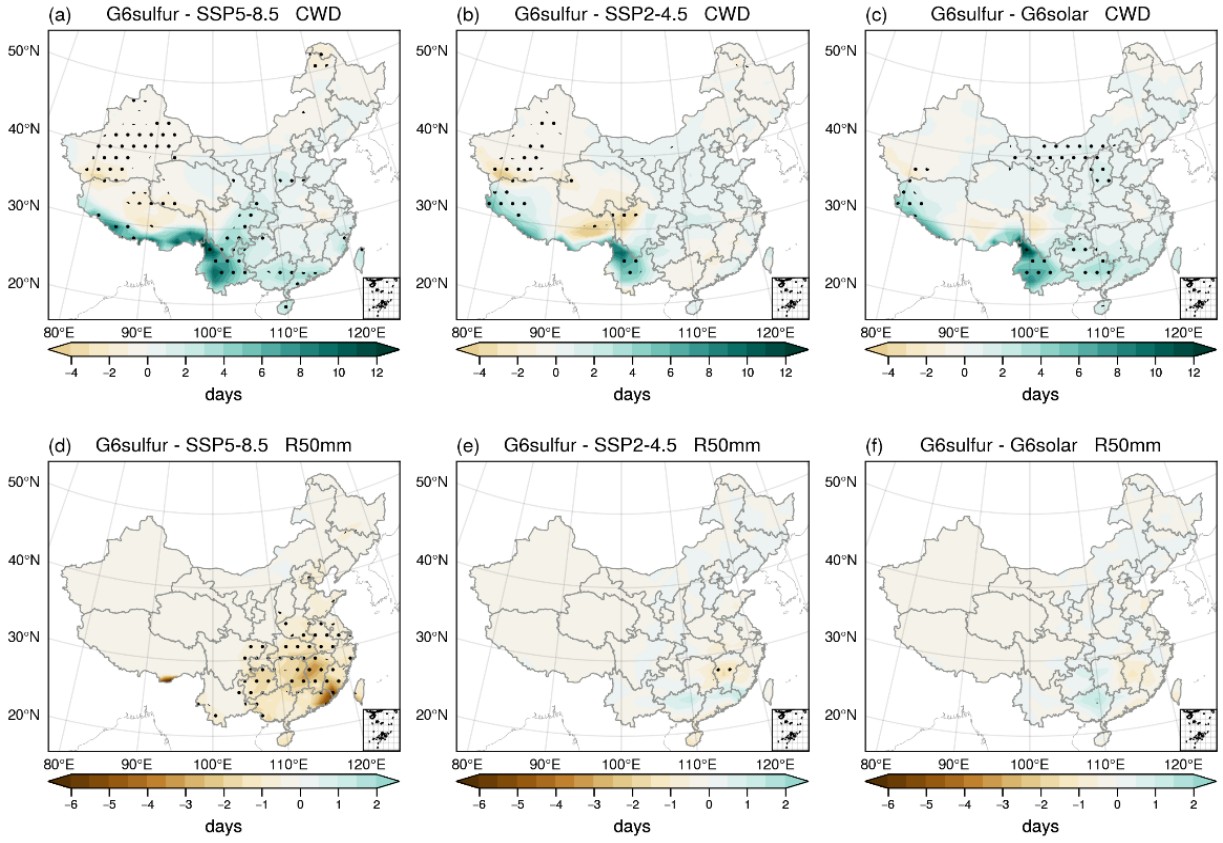

**Figure 8. Same as Figure 5, but for CWD (a-c) and R50mm (d-f).**

The maximum value of CWD in SWC exceeds 200 days, as shown in Fig.9, contributing to a general increase across China (Fig9(a1)). Meanwhile, CWD are projected to decrease, especially in SWC in Fig.9(a8)) and corresponds to the brown shaded areas in SWC as depicted in Fig.7(a-d).





The changes in CWD in the future are uncertain, as is whether they increase or decrease relative to PD. Furthermore, CWD under SSP5-8.5 and SSP2-4.5 scenarios does not consistently exhibit simultaneously increases or decreases across all regions. It is notable that in NC, and SC, G6sulfur (purple) provides similar results to the SSP2-4.5. When combined with Fig.8b and c (less than 2 days), suggests that G6sulfur yields statistically similar outcome to that of SSP2-4.5 in NC and SC. In SWC, the purple line and orange line also closely tracks the green line, indicating a similar cumulative distribution of CWD between G6sulfur and SSP2-4.5, and between G6solar and SSP2-4.5. However, there is an uneven spatial distribution, as seen in Fig.8b, and Fig.S5. Consequently, SRM is not expected to reach the levels of SSP2-4.5 in SWC. Interestingly, for EC, SSP2-4.5 yields almost identical statistics to SSP5-8.5, while both G6sulfur and G6solar show an increase compared to SSP scenarios. However, the negative values of CWD in EC in Table 2 and S2 indicate that SRM strategies cannot ameliorate the high values of CWD in the EC region.

Figure 8 highlights subtle differences between SSP2-4.5 and SRM strategies, a pattern also evident in the CDF analyses illustrating response comparisons. Therefore, moving forward, discussions regarding the CDFs of R50mm will be omitted.

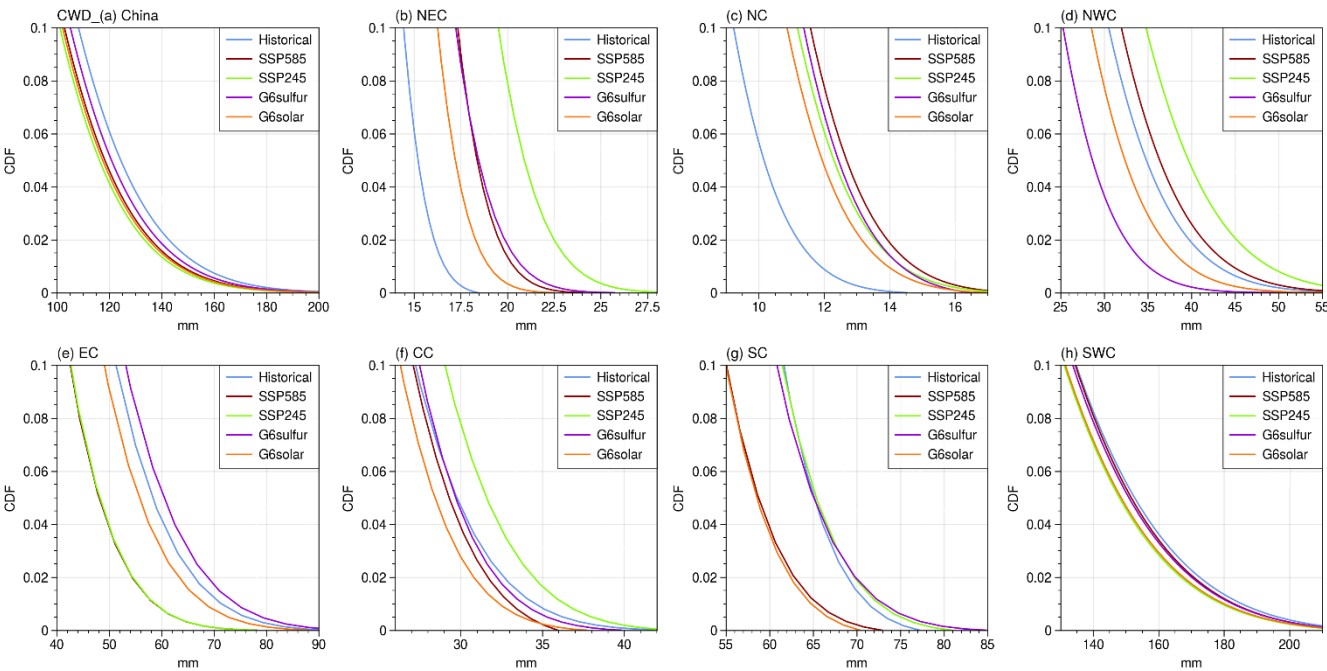

**Figure 9. Same as Figure 6, but for CWD in China and 7 subregions.**

In summary, CWD under SSP5-8.5 and SSP2-4.5 scenarios does not consistently exhibit simultaneously increases or decreases across all regions. CWD will significantly increase in QTP, and decrease in north China (almost in mid-latitudes, north of 30°N) but not serious in the future. The maximum increase occurs is in SWC, SAI effectively ameliorates the decrease of



CWD under SSP5-8.5 in southeast QTP (from about 90°E - 100°E). G6sulfur yields statistically similar outcome to that of SSP2-4.5 in NC and SC. SRM strategies yields results in R50mm nearly identical to those of the SSP2-4.5 scenario, and would have the most significant impact on weakening rainstorms in these densely populated regions.

### 3.2.2 Dry extreme changes

As for extreme wet indices, extreme dry conditions also occur in China, especially in north-western regions (Wang et al., 2017). The focus is on DD and CDD to study these changes and explore the impact of SAI.

For DD, the ensemble mean of UKESM simulations projects a significant increase (p-value < 0.05) in most of southern China (east of SWC, SC, and south of CC, EC) and a small part of western Xinjiang province (Fig.10a) for the SSP5-8.5 scenario
(2071-2100) relative to the PD (1981-2010). The largest increase, reaching up to 40 days, is observed in Fuzhou and Taiwan (southeast SC, near 120° E, 25°N). Decreases in DD are observed in northern and west China, including NEC, NC, NWC, west of SWC and north of CC, EC. Similar changes with smaller magnitudes are also observed under the SSP2-4.5 scenario (Fig.10b) and G6solar (Fig.10d). It is worth noting that, in comparison to the other three experiments, G6sulfur results in the most substantial increase in DD in western NWC (Fig.7c, Kunlun Mountains) in the future, indicating that G6sulfur
exacerbates drought conditions in Kunlun Mountains. This may be related to topography and slope, both of which play important roles in glacier change in the Kunlun Mountains (Niu et al., 2023).

Under SSP5-8.5 warming conditions (Fig.10e), there is a significant CDD decrease (p-value < 0.05) in north-western and northern China, with the most significant decrease in NWC, consistent with Xu et al., (2019). This implies that ignoring rising
temperatures seems to mitigate dry conditions (Xu et al., 2022a). In line with prior studies, there are notable north-south CDD variation in China (Feng et al., 2011).

The figure shows increased CDD in southern regions (along the middle and lower Yangtze River, south and parts of southwest China) but not significantly, hinting at potential increased droughts in southern China (Feng et al., 2011). These results also
align with the predicted decrease in the north and increase in the south in CDD by RegCM4 (Ji and Kang, 2015) and PRECIS (Meng et al., 2021) models. Smaller CDD changes are observed in most regions under SSP2-4.5 (Fig.10f), G6sulfur (Fig.10g) and G6solar (Fig.10h). It worth noting that G6sulfur shows a slight future increase in western China (Tarim Basin).



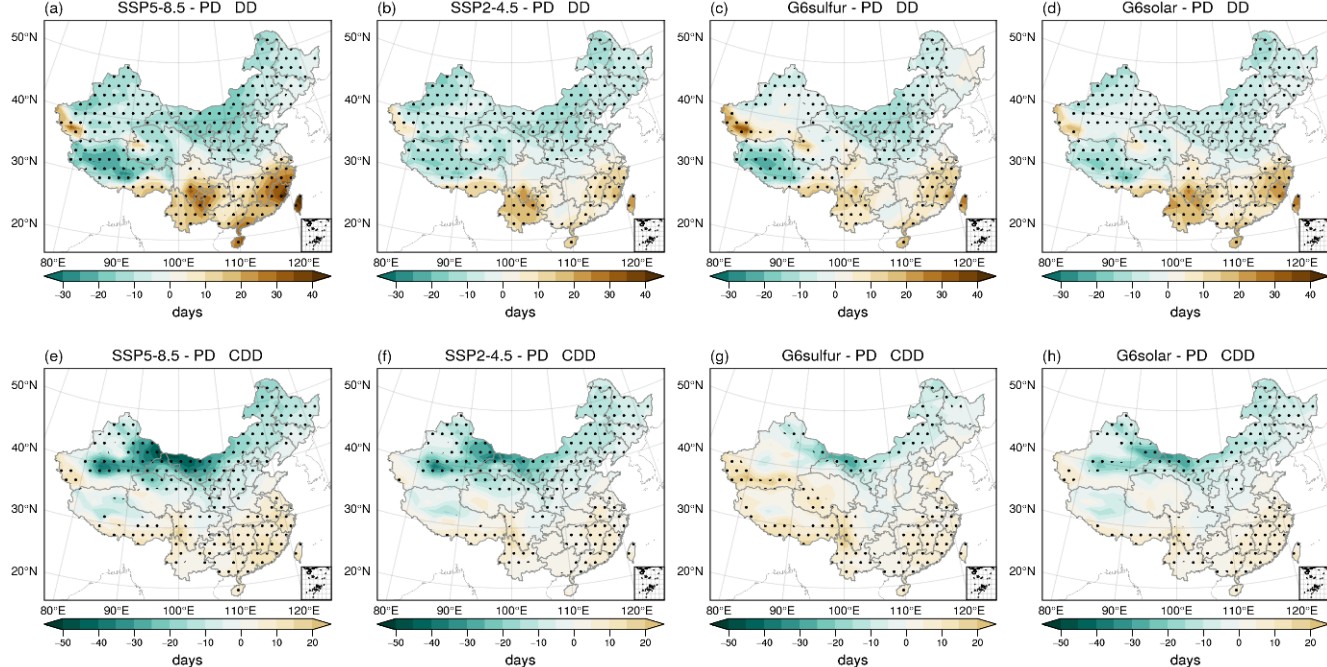


**Figure 10. Same as Figure 4, but for (a-d) DD (days) and (e-h) CDD (days).**

Fig.11 shows the impact of SAI on DD and CDD distribution in comparison to G6sulfur and other experiments. G6sulfur increases DD in northern and northwest China compared to SSP5-8.5 (Fig.11a), albeit mitigating the DD in a warmer climate

(Fig.10c). Compared to SSP2-4.5 (Fig.11b), G6sulfur leads to a further increase in western China near 35°N, up to 25 days. However, G6sulfur reduces dry climate in the south China compared to other experiments, despite an increased drought risk (as seen in Fig.10a-d). Comparisons confirm that G6sulfur increases CDD in most inland areas of China compared to SSP5-8.5 (Fig.11d), Only a few coastal areas show a reduction. G6sulfur increases the CDD almost across the entire China when compared to SSP2-4.5 (Fig.11e) and G6solar (Fig.11f).




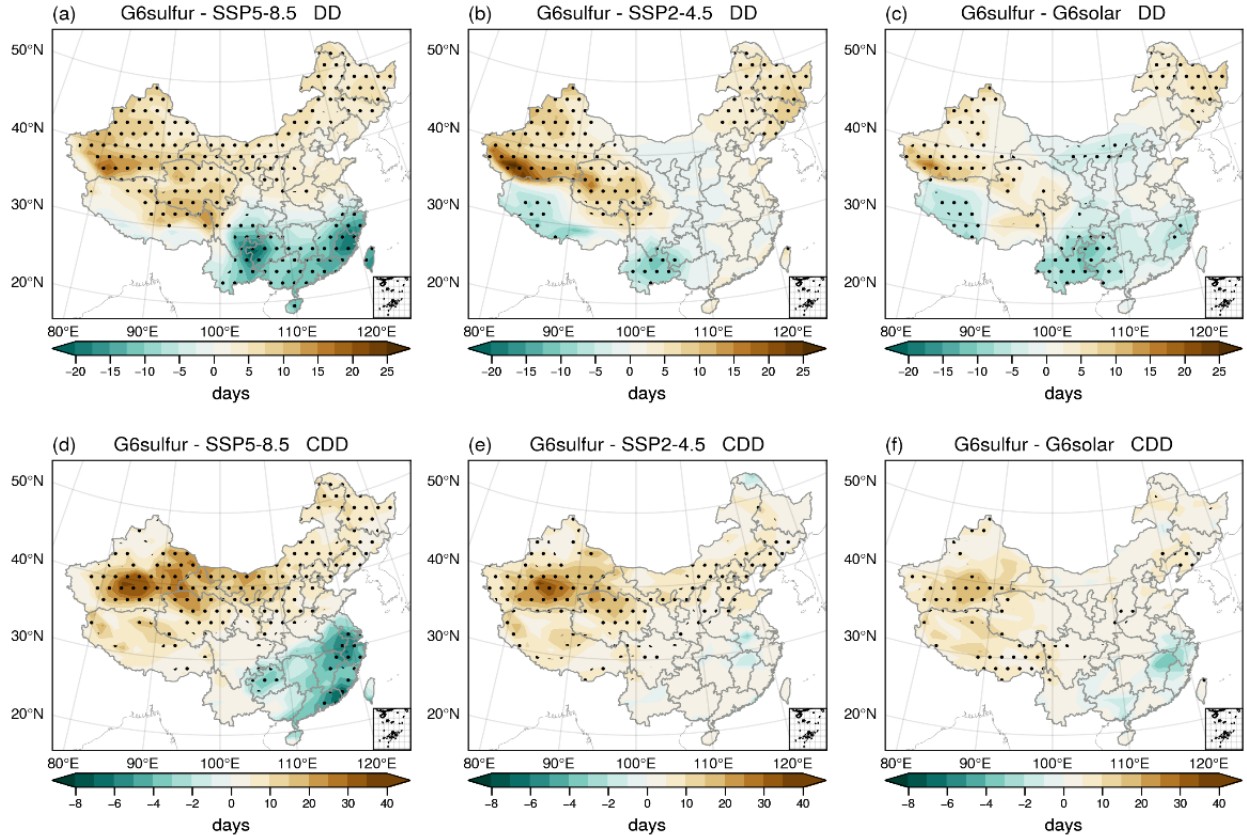

**Figure 11. Same as Figure 5, but for (a-c) DD (days) and (d-f) CDD (days).**

The tail of the DD CDFs in PD (blue) is consistently shifts to the right compared to the other lines in NEC, NC, and NWC
(Fig.12(a2) - (a4)), which corresponds to the declining trend (green shaded areas) in northern China as shown in Fig.10a-d.
The purple line surpasses the other three experiments in NWC, which explains the maximum increase value along Kunlun
Mountains in NWC in Fig.10a-d and Fig.11a-c. It is noteworthy that in CC and SC, G6sulfur yields similar statistic results as
SSP2-4.5.


The blue line surpasses the other four lines in NEC, NC and NWC, clarifying the decrease of CDD in the northern regions as
evident in Fig.10e-h, signifying a reduced drought risk in northern China in the future. Conversely, the red line consistently
stays to the left, while the purple line is positioned to the right compared to the other G6 experiment lines. This suggests that
G6sulfur and G6solar increase the drought risk when compared to the SSP5-8.5 scenario in northern regions, and the effect of
G6sulfur is more pronounced than that of G6solar. However, the distance between purple and other G6 experiments lines is
wider in NWC than in other regions, indicating the maximum increase in CDD under G6sulfur in NWC. This corresponds to
the maximum differences in NWC in Fig.11d-f. It is worth noting that G6solar outperforms G6sulfur compared to SSP2-4.5





in China (except for SWC region) as the distance between the orange line and the green line is smaller than that between the purple line and the green line.




**Figure 12. Same as Figure 6, but for DD (a1-a8) and CDD (b1-b8) in China and 7 subregions.**



In summary, both DD and CDD projected increase in the south and southeast China but decrease in the north and northwest regions by the end of the 21$^{st}$ century in 4 experiments. This reflects declining drought risk in northwest regions and increased extreme drought events in low-latitude southeast coastal areas in the future by G6 simulations. G6sulfur exacerbates drought conditions in Kunlun Mountains compared to other three experiments, and yields similar statistic results of DD as SSP2-4.5 in southeastern regions. SRM increase the drought risk when compared to SSP5-8.5 and SSP2-4.5 scenarios in northern regions

(NEC, NC and NWC), and the effect of G6sulfur is more pronounced than that of G6solar. G6solar outperforms G6sulfur compared to SSP2-4.5 in China (except for SWC region).

## 4. Summary and discussion

In this study, the effect of SAI on precipitation and the related extreme metrics is assessed over different sub-regions in China by comparing the G6sulfur experiment with the high (SSP5-8.5) and medium (SSP2-4.5) levels of GHG emissions and solar

constant reduction (G6solar) based on the coupled simulations of UKESM1. As both G6sulfur and G6solar aim to reduce future warming from the high-end GHG emission scenario (SSP5-8.5) to the medium GHG emission scenario (SSP2-4.5), the results reveal distinct responses of precipitation extreme indices to different statuses of external climate forcing. The conclusions of the paper are summarized as follows:

1. Under future warming scenarios (SSP5-8.5 and SSP2-4.5), most of sub-regions in China are projected to experience

increased precipitation and extreme wet climate events by the end of 21$^{st}$ century, particularly in the eastern and south-eastern coastal areas. This aligns with the prior study that found significant increases in precipitation extremes under the SSP5-8.5 and SSP2-4.5 scenarios (Xu et al., 2022b). Research indicates that the future reduction in anthropogenic aerosol emissions is expected to significantly enhance East Asian summer monsoon circulation and precipitation, potentially contributing to the increased precipitation extremes across China (Wang et al., 2016). SSP5-8.5 high

emission scenario projects a strengthening of future precipitation, while the G6sulfur and G6solar show ameliorated changes in precipitation.

2. With respect to SSP5-8.5, SRM effectively mitigates the increases in extreme rainfall intensities (RX1day, RX5day, and R95p) and frequency (R50mm), especially in the populated southeast areas. For RX1day, G6sulfur provides statistical outcomes more similar than G6solar compared to SSP2-4.5 in NC. While G6solar provides statistical

outcomes more similar than G6sulfur to those obtained from SSP2-4.5 in SWC. For R95p, G6solar provides statistically outcome more similar than G6sulfur to those obtained from SSP2-4.5 in EC, NC and SWC. SRM strategies weaken rainstorms in densely populated regions. SAI effectively ameliorates the decrease in CWD under SSP5-8.5 in southeast QTP (from about 90°E - 100°E), and yields statistically similar outcome to those of SSP2-4.5 in NEC, NC, EC, and SC.



3. Simulations of extreme drought events show a projected increase in south and southeast China. While there has been an improvement in mitigating drought conditions in Kunlun Mountains and Tarim Basin, G6sulfur exacerbates drought conditions compared to the other three experiments. G6sulfur yields similar statistic results of DD as SSP2-4.5 in southeastern regions (EC, CC and SC). The increased drought risk under G6sulfur is more pronounced than that under G6solar in northern regions (NEC, NC and NWC) when compared to the SSP5-8.5 and SSP2-4.5 scenarios. G6solar outperforms G6sulfur compared to SSP2-4.5 in China (except for SC region). Additionally, in NC and SWC, G6solar provides statistically similar results to SSP2-4.5.

The difference is shown in the magnitude as the use of SAI has more effects in reducing precipitation compared to solar constant reduction. The spatial patterns on extremely high and low precipitations under both intervention strategies are highly similar, in which this result aligns with the conclusions from (Visioni et al., 2022) on global extremes under SAI and solar constant reduction.

It is projected that the intensification of the hydrological cycle evident under SSP5-8.5 will become less intensified under both types of SRM in China. Previous studies, including (Niemeier et al., 2013) and (Ji et al., 2018), highlighted that the precipitation reduction response is robust under the SRM strategies. Our results of ETDCCI indices related to extreme precipitation and heavy rainfalls have concluded that SAI is mainly effective in mitigation in southeastern and southern China. The regions under the influence cover many important economic centres and port cities in China that are coastal and experience a high level of flooding risk. The use of SAI can be seen as a future option for preventing major flooding events. While both SRM strategies effectively reduce drying changes in most northern and western China, their implementation leads to an increased risk of drought within these regions compared to the SSP5-8.5 and SSP2-4.5 scenarios. Overall, our study shows efficacy of SAI in mitigating wet extremes in China. However, it highlights the unsuitability of SRM strategies for addressing drought risk in the northern and western regions.

As this study is solely focused on the precipitation and relevant extreme events based on the models, we cannot take socio-economic, biological, and other factors into account. Although many studies have also focused on areas such as crops (Cheng et al., 2019) and important modes of natural variability (Jones et al., 2020), their research is targeted on a wider and global scale, more regional analysis on China still requires future research. Also, the climate models and data remain uncertain, indicating the continuous improvement in models for simulations in deterring the future pathways of climate change and SRM. All the findings and conclusions in our paper extend the current understating of extreme hydrological responses to climate change and SAI in China. The regional analysis presents new insights into identifying the vulnerable areas under hydrological changes and how they may benefit from the SRM. We note that, owing to the second-order nature of the changes in climate extremes when compared to SSP2-4.5 (i.e. a relatively small signal to noise when compared to those from SSP5-8.5), that the analysis is very dependent on the model used in the analysis (UKESM1); other models may produce significantly different





results. It is therefore crucial to perform similar analyses with other state-of-the-art climate models to elucidate the robustness

of the results, and to inform policymakers of any potential detrimental influences of SRM.

The general message appears to be that both the G6sulfur and G6solar simulations abate most of the changes in detrimental extremes that are evident under the SSP5-8.5 scenario. Changes in extreme precipitation under G6sulfur and G6solar for all precipitation variables bar the consecutive wet days (CWD; where negligible changes occur compared to present day

conditions occur in NEC and EC regions and a increase in NWC) show only subtle second order differences as compared to SSP2-4.5. As the global mean temperature targets of G6sulfur, G6solar and SSP2-4.5 are nominally identical, this suggests that it is the global mean temperature change that is the dominant factor in driving changes in extreme precipitation.

While the general amelioration of precipitation changes under SAI as compared to those under SSP5-8.5 might seem a

somewhat obvious conclusion owing to the spin-down of the hydrological cycle under cooler temperatures (e.g. Tilmes et al, 2013), other studies have shown large-scale climatic shifts in key modes of climate variability that impact precipitations. For example, Haywood et al. (2013) and Jones et al. (2017) have modelled significant detrimental impacts on Sahelian precipitation and north Atlantic hurricane frequency under non-judicious SAI implementation owing to large-scale shifts in the Inter-Tropical Convergence Zone. Multi-model SAI simulations Jones et al. (2020, 2021) have shown detrimental impacts on the

North Atlantic Oscillation leading to rainfall deficits over the Iberian Peninsula above and beyond those evident in SSP5-8.5. Similarly, recent simulations of non-judicious deployment of an alternate SRM technique, that of marine cloud brightening, locked the climate into an extremely strong permanent La-Nina-like phase with associated detrimental impacts on sea-level rise over low-lying South Pacific islands (Haywood et al., 2023). It appears that, over large areas of China, any changes in detrimental extremes in precipitation are second order when compared to the benefits associated with reducing global mean

temperatures.

For dry days that influence drought (i.e. the DD and CDD diagnostics), the situation is rather different with increased dry days and continuous dry days as compared to SSP5-8.5. Again, this is consistent with SRM activities spinning down the hydrological cycle (e.g. Tilmes et al., 2013).


To conclude, it appears that changes in precipitation extremes related to flooding over the bulk of China that are induced under climate change may be abated by SRM, but changes in dry days relating to drought are likely to be enhanced. Large-scale shifts in precipitation patterns associated with changes in atmospheric dynamics noted in other SRM studies using climate models developed by the Hadley Centre (e.g. HadGEM2, UKESM1; Haywood et al., 2013, 2023; Jones et al., 2017; Jones et

al., 2020; 2021) do not appear to impact the bulk of China. Based on the same set of simulations as this paper, the study by Liang and Haywood (2023) demonstrated apparent side-effects of SRM as the simulated SAI scenario excebates the weakening the the subtropical westerly jet and further enhance the mid-latitude precipitation over China by modulating atmospheric rivers



over East Asia. Of course, we stress that the results from these simulations are model specific and further work with other
models needs to be performed to understand the robustness of these conclusions more generally.


## Code and data availability

All the UKESM1 model data for the SSP5-8.5, SSP2-4.5, GeoMIP G6sulfur and GeoMIP G6solar scenarios used in this work
are available from the Earth System Grid Federation (WCRP, 2022; https://esgf-node.llnl.gov/projects/cmip6). The
APHRODITE data have been downloaded from their official website, which is managed by the Data Integration and Analysis
System (DIAS, 2022; https://search.diasjp.net/en/dataset/APHRO_MA).

## Supplement

The supplement related to this article is available online at:

## Author contributions

The majority of this work was completed when OW was visiting the University of Exeter in the UK under a scholarship from
the China Scholarship Council. JL and JH devised the experiment, led the analysis. OW wrote the paper with contributions
from all the co-authors.

## Competing interests

The contact author has declared that neither of the authors has any competing interests.

## Disclaimer

Publisher's note: Copernicus Publications remains neutral with regard to jurisdictional claims in published maps and
institutional affiliations.

## Acknowledgements

We are grateful to the University of Exeter for providing the academic platform for this study and the Met Office for doing the
numerical calculations in this work. The authors thank Andy Jones, for help with running the G6sulfur and G6solar experiments.



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
