# Peer review of "Projected future changes in extreme precipitation over China under stratospheric aerosol intervention in the UKESM1 climate model"

_EGUsphere, 2023_

## Referee Comment (RC1)

This paper examines the impact of changes in different measures of precipitation over China under a high emissions scenario and with a small ensemble of climate intervention simulations at the end of the century. One climate model (UKESM1) is used for comparison with two different realisations of stratospheric aerosol injection - G6solar, using a constant solar dimming, and G6sulfur, with gradually increasing injections of sulfur into the stratosphere. The paper is well organised and generally clearly and well written. My main criticism is that the discussion does not evaluate the results with respect to other research that was carried out from other experiments such as GLENS. Suggestions for other relevant literature is at the end together with references made in the comments below.

Major comments:

Consider reducing the content of Section 2.2. A lot of this is repetition from the literature that is cited and doesn't necessarily need to be included in this article.

To avoid confusion, consider changing any reference to "present-day" to the control period and sticking with this consistently, rather than switching between 'historical', 'baseline' and 'present-day'. The WMO has adopted 1990-2020 as the "current climate period", which suggests that the period used in this article is historical.

Is there a benefit in the data validation against APHRODITE? The article does not include an assessment of extreme precipitation metrics with respect to observations, nor do the results or conclusions refer back to the observations. I think you could drop this, and instead refer to other assessments of the validity of extreme precipitation in climate models such as (Sillmann et al., 2013; Donat et al., 2020; Tebaldi et al., 2021).

Given that this research uses one climate model, and three ensemble members for each scenario it isn't really appropriate to state that G6sulfur/G6solar abates or ameliorates climate change as depicted by SSP5-8.5. There aren't sufficient model members to remove model uncertainty, and without observations we do not know which model realisation (if any) adequately simulates the effects of SAI. All you can state with confidence is that using this methodology and data, the SAI experiments produce results consistent with a lower emissions target. I would prefer to see all of the statements on improvements with respect to climate change removed, or at least reduce the emphasis of the statements.

Minor comments:

L28 change impacts to efficacy

L30 rephrase this sentence as noted above.

L31 remove trends

L42 is the higher risk of flooding associated with increased extreme precipitation? Remove the time periods studied, this is implicit.

L44-60 is this level of detail on historic events warranted? You do not examine the changes in jets or other sources of extremes.

L44 Why was summer of 2020 anomalous - it seems in keeping with the other extreme events you reported.

L61 change appears to has, and cite relevant literature such as (Donat et al., 2016; Pendergrass and Knutti, 2018), which also discuss changes in the hydrological cycle.

L64 Should causes go before faster?

L66 Update this to the more nuanced and recent research that shows extreme precipitation generally goes up everywhere (Pendergrass and Knutti, 2018)

L79 SAI does not mitigate anthropogenic climate warming, it may mitigate some of the impacts.

L79-89 This paragraph needs rephrasing to explain that SAI is premised on reproducing the effects associated with volcanic eruptions. However, you do not need to list the different volcanic eruptions themselves - just point to a large body of literature that supports these effects.

L99 include other recent research that explored changes in temperature and precipitation in other SAI experiments not just the GeoMIP archive (e.g. Tye et al., 2022; Simpson et al., 2019).

L100 What about (Tew et al., 2023)?

L105 remove maximise the signal-to-noise in the simulations as

L114 remove according

L126 See comment above, but at the very least remove The GeopMIP G6sulfur simulations that reduce global mean temperatures from the SSP5-8.5 scenario to the SSP2-4.5 are described in detail elsewhere.

L143-145 remove this last sentence.

L154 I believe that the extreme indices were defined by the WCRP not IPCC.

Table 1: the authors should be Frich and Klein Tank. Also refer to (Sillmann et al., 2013; Zhang et al., 2011) for the correct definitions

How did you calculate the 95th percentile? Did you bootstrap the individual years to avoid data inhomogeneities (Zhang et al., 2005)

L161 This may not be relevant if you remove the Aphrodite data as suggested above. However, I am concerned about regridding the larger data to the smaller grid. No additional information is gained in this respect (just several grid boxes with the same values) and may show errors and biases that are not true. Instead it would be more robust to regrid the observations to match that of UKESM. See https://climatedataguide.ucar.edu/climate-tools/regridding-overview for more information.

L165 remove instead of the more commonly used Student's t=test. Wilcoxon Rank Sum Testse work as

L167 change "with p-value <0.05 suggesting" to "with a 5% confidence level of"

L169-182 How did you establish the CDFs? Did you fit distributions, or are these empirical CDFs from the data? Did you examine the uncertainty in the CDFs, and were they fitted for each model member (correct) or the model mean (as the figures suggest)? I am also wary about CDFs for very small sample sizes - i.e. 30 values of the annual maximum rainfall.

L185-194 put this into the previous section

L193 Include this statement in the figure caption instead of the text - and check which way you have represented significance, this is opposite from the figure.

L211 Comment on the increase in drought in west and Taiwan under G6sulfur.

L215 Change this statement to something like projected changes are similar to those of SSP2-45, meaning that the SAI simulations are approximately successful.

L246 There are no absolute values>100mm or no increases from the control period >100mm?

L250 remove (p-value <0/05) and every other instance - this has already been stated.

L251 remove which is generally. Is there really only one research paper on increases in extreme precipitation in this region?

L257 Remove this sentence.

L265 should this be depicting?

L266 remove sentence "The comparisons confirm…."

L278 remove "This suggests that G6sulfur…." This is the results section, so discussions aren't appropriate.

L280 Stick to reporting the differences between the simulations in this section, then interpret (with appropriate caveats) in the discussion sections and remove the sentences on the implications or efficacy of SSPs vs G6.

L291 Remove ameliorating

Table 2: I am not sure that this adds to the interpretation of the results and could be removed. If retained, re-phrase as difference between G6 and SSP or something similar.

L311 Why is this interesting? Elaborate please.

L325 It might be more meaningful to look at the relative changes (e.g. in percentage terms) rather than absolute values. With regard to the "arbitrary" regions, why are they somewhat arbitrary? Surely they relate to some geographical or political definition, the point to make is that they may not correspond with climatological regions. Note that smaller regions would just emphasise noise in the results.

L329/30 remove this sentence.

Figure 6 as noted above the uncertainty across model members should be included in these curves. Please also check the colour scheme for colour blind appropriateness, and use the same x-axis for each variable for all regions (i.e. one x-axis for Rx1day, another for R95p). This also applies to Figure 9 and 12.

L350 Decreases in CWD do not necessarily equate to reductions in precipitation intensity. You can only make this interpretation if there is a reduction in the total number of wet days AND an increase or no change in the annual total.

L356 R50mm is not one of the formal ETCCDI indices, it is user defined.

L371 remove effectively ameliorates the

L373-375 This is discursive and needs more references to support it (and moving to the discussion section). There is likely a combination at play including changes in the location of the jet streams and ITCZ, as well as interactions with topography and changes in maritime temperature gradients.

L380/1 Remove this sentence

L387 CWD=200 days is right at the far end of the tail, I don't think it's appropriate to make this statement without any error estimates or uncertainty information. Further, the duration estimates of CWD and CDD add up to longer than a year - this is particularly obvious in comparison with Figure 12. Please check.

L402 Should this refer to Figure 9? This statement would be more meaningful with uncertainty envelopes to clearly demonstrate whether there are or are not differences between each model.

L413 Remove , and would have the ….

L458 It is not noteworthy that these yield similar results - that's the objective of the cooling. Remove this sentence.

L468 Remove last sentence of paragraph.

L475-481 There are other aspects related to drought risk - not least evaporation - that don't show up in the dry day count, such a strong statement about changes in drought risk aren't appropriate. See (Cheng et al., 2019; Dagon and Schrag, 2019, 2016) for more results related to climate intervention.

G6solar and G6 Sulfur are different ways of simulating a possible climate, we have no way of knowing whether one or the other is more valid without observations, and so can't be described as outperforming each other. See (Bednarz et al., 2022; Visioni et al., 2021) for more discussion on this.

Section 4

I suggest condensing the bullet points to short concluding statements, which can then be followed by the explanations. Given that the focus of this article is on the climate intervention, it would also make more sense to emphasise those results rather than the future projections that have been published elsewhere. This is also the point to discuss how valid the results are with respect to other research - including experiments outside the GeoMIP project.

References

Aswathy, V. N., O. Boucher, M. Quaas, U. Niemeier, H. Muri, J. Mülmenstädt, and J. Quaas. "Climate Extremes in Multi-Model Simulations of Stratospheric Aerosol and Marine Cloud Brightening Climate Engineering." Atmospheric Chemistry and Physics 15, no. 16 (August 2015): 9593–9610. https://doi.org/10.5194/acp-15-9593-2015.

Bal, Prasanta Kumar, Raju Pathak, Saroj Kanta Mishra, and Sandeep Sahany. "Effects of Global Warming and Solar Geoengineering on Precipitation Seasonality." Environmental Research Letters 14, no. 3 (March 13, 2019): 034011. https://doi.org/10.1088/1748-9326/aafc7d.

Bala, G., P. B. Duffy, and K. E. Taylor. "Impact of Geoengineering Schemes on the Global Hydrological Cycle." Proceedings of the National Academy of Sciences 105, no. 22 (June 2008): 7664–69. https://doi.org/10.1073/pnas.0711648105.

Barnes, Elizabeth A., James Wilson Hurrell, and Lantao Sun. "Detecting Changes in Global Extremes under the GLENS-SAI Climate Intervention Strategy," July 6, 2022. https://doi.org/10.1002/essoar.10511813.1.

Bednarz, E. M., Visioni, D., Banerjee, A., Braesicke, P., Kravitz, B., and MacMartin, D. G.: The overlooked role of the stratosphere under a solar constant reduction, Geophysical Research Letters, https://doi.org/10.1029/2022GL098773, 2022.

Cheng, W., MacMartin, D. G., Dagon, K., Kravitz, B., Tilmes, S., Richter, J. H., Mills, M. J., and Simpson, I. R.: Soil Moisture and Other Hydrological Changes in a Stratospheric Aerosol Geoengineering Large Ensemble, Journal of Geophysical Research: Atmospheres, 124, 12773–12793, https://doi.org/10.1029/2018JD030237, 2019.

Curry, Charles L., Jana Sillmann, David Bronaugh, Kari Alterskjaer, Jason N. S. Cole, Duoying Ji, Ben Kravitz, et al. "A Multimodel Examination of Climate Extremes in an Idealized Geoengineering Experiment." Journal of Geophysical Research: Atmospheres 119, no. 7 (April 2014): 3900–3923. https://doi.org/10.1002/2013JD020648.

Dagon, K. and Schrag, D. P.: Exploring the Effects of Solar Radiation Management on Water Cycling in a Coupled Land–Atmosphere Model*, Journal of Climate, 29, 2635–2650, https://doi.org/10.1175/JCLI-D-15-0472.1, 2016.

Dagon, K. and Schrag, D. P.: Quantifying the effects of solar geoengineering on vegetation, Climatic Change, 153, 235–251, https://doi.org/10.1007/s10584-019-02387-9, 2019.

Donat, M. G., Lowry, A. L., Alexander, L. V., O'Gorman, P. A., and Maher, N.: More extreme precipitation in the world's dry and wet regions, Nature Climate Change, 6, 508–513, https://doi.org/10.1038/nclimate2941, 2016.

Donat, M. G., Sillmann, J., and Fischer, E. M.: Changes in climate extremes in observations and climate model simulations. From the past to the future, in: Climate Extremes and Their Implications for Impact and Risk Assessment, edited by: Sillmann, J., Sippel, S., and Russo, S., Elsevier, 31–57, https://doi.org/10.1016/B978-0-12-814895-2.00003-3, 2020.

Pendergrass, A. G. and Knutti, R.: The Uneven Nature of Daily Precipitation and Its Change, Geophysical Research Letters, 45, 11,980-11,988, https://doi.org/10.1029/2018GL080298, 2018.

Sillmann, J., Kharin, V. V., Zhang, X., Zwiers, F. W., and Bronaugh, D.: Climate extremes indices in the CMIP5 multimodel ensemble: Part 1. Model evaluation in the present climate, Journal of Geophysical Research: Atmospheres, 118, 1716–1733, https://doi.org/10.1002/jgrd.50203, 2013.

Simpson, I. R., Tilmes, S., Richter, J. H., Kravitz, B., MacMartin, D. G., Mills, M. J., Fasullo, J. T., and Pendergrass, A. G.: The Regional Hydroclimate Response to Stratospheric Sulfate Geoengineering and the Role of Stratospheric Heating, Journal of Geophysical Research: Atmospheres, 124, 12587–12616, https://doi.org/10.1029/2019JD031093, 2019.

Tebaldi, C., Dorheim, K., Wehner, M., and Leung, R.: Extreme metrics from large ensembles: investigating the effects of ensemble size on their estimates, Earth Syst. Dynam., 12, 1427–1501, https://doi.org/10.5194/esd-12-1427-2021, 2021.
Tew, Y. L., Tan, M. L., Liew, J., Chang, C. K., and Muhamad, N.: A review of the effects of solar radiation management on hydrological extremes, IOP Conf. Ser.: Earth Environ. Sci., 1238, 012030, https://doi.org/10.1088/1755-1315/1238/1/012030, 2023.

Tye, M. R., Dagon, K., Molina, M. J., Richter, J. H., Visioni, D., Kravitz, B., and Tilmes, S.: Indices of Extremes: Geographic patterns of change in extremes and associated vegetation impacts under climate intervention, Earth System Dynamics, 13, 1233–1257, https://doi.org/10.5194/esd-13-1233-2022, 2022.

Visioni, D., MacMartin, D. G., and Kravitz, B.: Is Turning Down the Sun a Good Proxy for Stratospheric Sulfate Geoengineering?, Geophys Res Atmos, 126, e2020JD033952, https://doi.org/10.1029/2020JD033952, 2021.

Wei, Liren, Duoying Ji, Chiyuan Miao, Helene Muri, and John C. Moore. "Global Streamflow and Flood Response to Stratospheric Aerosol Geoengineering." Atmospheric Chemistry and Physics 18, no. 21 (November 2018): 16033–50. https://doi.org/10.5194/acp-18-16033-2018.

Zhang, X., Hegerl, G., Zwiers, F. W., and Kenyon, J.: Avoiding Inhomogeneity in Percentile-Based Indices of Temperature Extremes, Journal of Climate, 18, 1641–1651, https://doi.org/10.1175/JCLI3366.1, 2005.

Zhang, X., Alexander, L., Hegerl, G. C., Jones, P., Tank, A. K., Peterson, T. C., Trewin, B., and Zwiers, F. W.: Indices for monitoring changes in extremes based on daily temperature and precipitation data, Wiley Interdisciplinary Reviews: Climate Change, 2, 851–870, https://doi.org/10.1002/wcc.147, 2011.

---

## Author Comment (AC1)

**Dear referee,**

Thank you very much for the valuable comments on our manuscript. We have considered all the comments carefully, and will revise the manuscript according to the comments and suggestions by you and other referees, as well as community reviewers. Below is a point-by-point response your comments.

With warm regards,

Ou Wang, Ju Liang, Yuchen Gu, Jim M. Haywood*, Ying Chen, Chenwei Fang, Qin`geng Wang*

**General comment:**

This paper examines the impact of changes in different measures of precipitation over China under a high emissions scenario and with a small ensemble of climate intervention simulations at the end of the century. One climate model (UKESM1) is used for comparison with two different realisations of stratospheric aerosol injection - G6solar, using a constant solar dimming, and G6sulfur, with gradually increasing injections of sulfur into the stratosphere. The paper is well organised and generally clearly and well written. My main criticism is that the discussion does not evaluate the results with respect to other research that was carried out from other experiments such as GLENS. Suggestions for other relevant literature is at the end together with references made in the comments below.

**Major comments:**

Consider reducing the content of Section 2.2. A lot of this is repetition from the literature that is cited and doesn't necessarily need to be included in this article.

**Response:**

Thank you very much for your suggestion. We have realized many repetitions from the literature are not necessary. The section 2.2 will be revised as:

"In this study, data from G6sulfur and G6solar experiments are used from the sixth phase of GeoMIP from the U.K. Earth System Model UKESM1 (Sellar et al., 2019). UKESM1 is a fully coupled Earth system model with an atmospheric resolution of

1.25°latitude by 1.875°longitude (Storkey et al., 2018; Walters et al., 2019; Mulcahy et al., 2018, Sellar et al., 2019), and contributes to both CMIP6 and GeoMIP6 (Jones et al., 2020). The Scenario MIP high GHG forcing scenario SSP5-8.5 (O'neill et al., 2016) is used as the baseline scenario of both G6solar and G6sulfur experiments (Kravitz et al., 2015). UKESM1 simulates $SO_2$ injection in the stratosphere along the Greenwich meridian at an altitude of 18-20 km between 10° N and 10° S in G6sulfur over the period 2020-2100 (Kravitz et al., 2021; Haywood et al., 2022). A parallel experiment to G6sulfur, the G6solar experiment, reduces ScenarioMIP Tier 1 high forcing scenario to the medium forcing scenario by reducing solar irradiance. Notably, it is anticipated that G6solar will exhibit reduced inter-model disparities in the spatial distribution of forcing when compared to G6sulfur owing to model differences in representing the complexities of the sulfur cycle within global models. Therefore, G6solar is proposed as a parallel experiment to G6sulfur for the purpose of comparing the impacts of solar reduction with those of stratospheric aerosols (Kravitz et al., 2015)."

To avoid confusion, consider changing any reference to "present-day" to the control period and sticking with this consistently, rather than switching between 'historical', 'baseline' and 'present-day'. The WMO has adopted 1990-2020 as the "current climate period", which suggests that the period used in this article is historical.

**Response:**

Thank you. We have changed all references to "present-day" to "control period" throughout the manuscript.

Is there a benefit in the data validation against APHRODITE? The article does not include an assessment of extreme precipitation metrics with respect to observations, nor do the results or conclusions refer back to the observations. I think you could drop this, and instead refer to other assessments of the validity of extreme precipitation in climate models such as (Sillmann et al., 2013; Donat et al., 2020; Tebaldi et al., 2021)

**Response:**

Thank you for this important comment. Indeed, this study has benefits in the data validation against APHRODITE. First of all, the observations were used to validate the direct results of the model (i.e., amount of precipitation). We think this is the most fundamental for our study. In this regard, further description and analysis will be added in the revised manuscript. In addition, a scatter plot will be added as a new panel in

Figure 2 (as shown below), and the bias as a percent instead of an absolute value between the observations and the models (ensemble mean and the three model members) will be indicated, as suggested by another referee (RC3). Please refer to the response to the RC3 for more detail. More comparison on the extreme precipitation metrics between the observations and the model results will be performed and discussed in the revised manuscript.

[Figure]

**Figure 2(d) Scatter plots between the observations and model results at different level of precipitation during the control period (CP).**

The observations were classified into several level (intervals): P10 (the smallest 10%), P10-50, P50-90, P90-95, and P95 (the largest 5%).

In addition, according to your suggestion, some assessments in other relevant studies (e.g., Sillmann et al., 2013; Donat et al., 2020; Tebaldi et al., 2021) on the validity of extreme precipitation in climate models will also be mentioned in our revised manuscript. We appreciate it very much for providing the valuable references.

Given that this research uses one climate model, and three ensemble members for each scenario, it isn't really appropriate to state that G6sulfur/G6solar abates or ameliorates

climate change as depicted by SSP5-8.5. There aren't sufficient model members to remove model uncertainty, and without observations we do not know which model realisation (if any) adequately simulates the effects of SAI. All you can state with confidence is that using this methodology and data, the SAI experiments produce results consistent with a lower emissions target. I would prefer to see all of the statements on improvements with respect to climate change removed, or at least reduce the emphasis of the statements.

**Response:**

We agree with you, and thank you very much for the suggestion that makes our paper more rigorous and scientific. Really, considering many possible uncertainties in the climate models, as well as in the research scenarios, we cannot assert that G6sulfur/G6solar abates or ameliorates climate change as depicted by SSP5-8.5. Accordingly, relevant statements in the manuscript will be revised to reflect the limitations of our methodology and the uncertainties associated with the findings.

**Minor comments:**

L28 change impacts to efficacy

**Response:**

L28: "impacts" has been changed to "efficacy".

L30 rephrase this sentence as noted above.

**Response:**

L30: The sentence "While the results from both G6sulfur and G6solar show encouraging abatement of many of the impacts on detrimental extreme events that are evident in SSP5-8.5 there are some exceptions." has been change to "In all, the results from both G6sulfur and G6solar are encouraging, showing a reduction in the efficacy of detrimental extreme events, consistent with the lower emissions target of SSP2-4.5."

L31 remove trends

**Response:**

L31: "trends" has been removed as suggested.

L42 is the higher risk of flooding associated with increased extreme precipitation? Remove the time periods studied, this is implicit.

**Response:**

Yes, the higher risk of flooding is indeed associated with increased extreme precipitation. This is indicated in the results by Ying et al. (2014), where the flood risk is understood as an extreme climate index. Indeed, the time periods are implicit, and will be removed.

L44-60 is this level of detail on historic events warranted? You do not examine the changes in jets or other sources of extremes.

**Response:**

The details on historic events are mostly quoted from published papers or media reports. To be honest, we could not warrant their reliability since we have not conducted further investigation in this regard. Now, according to the comments by another referee (RC2), we have come to realize that the detailed description on the historic events is not necessary as it is not directly relevant to the study here. Therefore, we the part (L44-60) will be deleted or shortened in our revised manuscript.

L44 Why was summer of 2020 anomalous - it seems in keeping with the other extreme events you reported.

**Response:**

The using of the word "anomalous" was not appropriate. Besides 2020, flooding events also frequently happened in other years. We will correct it later. In addition, as mentioned above, the detailed description on the historic events will be shortened in our revised manuscript.

L61 change appears to has, and cite relevant literature such as (Donat et al., 2016; Pendergrass and Knutti, 2018), which also discuss changes in the hydrological cycle.

**Response:**

The sentence has been changed as: "On a global scale, climate change has been influencing hydroclimatic conditions (Donat et al., 2016; Pendergrass and Knutti, 2018)."

L64 Should causes go before faster?

**Response:**

Sorry for the wording mistake. The sentence should be "Climate change causes faster evaporation, and higher atmospheric temperature induce more moisture-laden air in the storm tracks."

L66 Update this to the more nuanced and recent research that shows extreme precipitation generally goes up everywhere (Pendergrass and Knutti, 2018)

**Response:**

We will update that based on relevant studies (e.g., Pendergrass and Knutti, 2018).

L79 SAI does not mitigate anthropogenic climate warming, it may mitigate some of the impacts.

**Response:**

You are right. The sentence will be revised as: "To some extent, SAI partially counteract climate warming by injecting reflective particles, or their gaseous precursors, into the stratosphere (Zarnetske et al., 2021)."

L79-89 This paragraph needs rephrasing to explain that SAI is premised on reproducing the effects associated with volcanic eruptions. However, you do not need to list the different volcanic eruptions themselves - just point to a large body of literature that supports these effects.

**Response:**

Thank you for this comment which has helped make our manuscript more concise and less verbose. The paragraph (L79-89) will be changed as: "SAI is premised on replicating the effects associated with volcanic eruptions, wherein reflective particles or their gaseous precursors are injected into the stratosphere. These resultant aerosols reflect and scatter solar radiation back into space, leading to a cooling effect that counterbalances the warming caused by increased concentrations of greenhouse gases (e.g. Bluth et al., 1992; Self et al., 1996; Robock, 2000; Soden et al., 2002; Haywood et al., 2014; Schmidt et al., 2018). In addition to reducing the temperature, SAI also influences tropospheric and stratospheric ozone, terrestrial ecosystem, terrestrial carbon, and hydrological cycle by changing the physical climate system and atmospheric chemistry. Numerous studies support these effects associated with volcanic eruptions and their simulation through SAI techniques (e.g.Liang and Haywood., 2023; Jones et

al., 2018; Jones et al., 2020; Cao, 2018; Plazzotta et al., 2019; Lee et al., 2021; Visioni et al., 2022; Imai et al., 2020; Mclandress et al., 2011). ”

L99 include other recent research that explored changes in temperature and precipitation in other SAI experiments not just the GeoMIP archive (e.g. Tye et al., 2022; Simpson et al., 2019).

L100 What about (Tew et al., 2023)?

**Response:**

Thank you for the suggestion. We have incorporated other recent research, including studies exploring changes in temperature and precipitation in various SAI experiments beyond the GeoMIP archive, such as those by Tye et al. (2022), Simpson et al. (2019), and Tew et al. (2023).

L105 remove maximise the signal-to-noise in the simulations as

L114 remove according

L126 See comment above, but at the very least remove The GeopMIP G6sulfur simulations that reduce global mean temperatures from the SSP5-8.5 scenario to the SSP2-4.5 are described in detail elsewhere.

L143-145 remove this last sentence.

**Response:**

Your above suggestions are all accepted in revising the manuscript.

L154 I believe that the extreme indices were defined by the WCRP not IPCC.

Table 1: the authors should be Frich and Klein Tank. Also refer to (Sillmann et al., 2013; Zhang et al., 2011) for the correct definitions

**Response:**

Thank you for pointing out the mistake, and we have corrected it in the text.

How did you calculate the 95th percentile? Did you bootstrap the individual years to avoid data inhomogeneities (Zhang et al., 2005)

**Response:**

For each grid, the 95th percentile was calculated based on 30 years (1981-2010) of precipitation data. We calculated the 95th percentile directly without using

bootstrapping methods, as recommended for calculating temperature indices (https://www.climdex.org/learn/indices/#index-TX90p).

**Response:**

Thank you for your valuable suggestion. The observations were regridded to match that of UKESM.

L165 remove instead of the more commonly used Student's t=test. Wilcoxon Rank Sum Testse work as
**Response:**

The text was removed.

L167 change "with p-value <0.05 suggesting" to "with a 5% confidence level of"
**Response:**

The expression was changed.

L169-182 How did you establish the CDFs? Did you fit distributions, or are these empirical CDFs from the data? Did you examine the uncertainty in the CDFs, and were they fitted for each model member (correct) or the model mean (as the figures suggest)? I am also wary about CDFs for very small sample sizes - i.e. 30 values of the annual maximum rainfall.
**Response:**

Thank you for pointing out this problem that we didn't explain it clearly about how we established the CDFs. In fact, for establishing the CDFs, firstly, for an extreme precipitation index at each grid point, the yearly mean of the ensemble model members was calculated. Then, the annual extreme precipitation indices for each grid point was

obtained by averaging over yearly means during the 30 years. Finally, the cumulative probability distribution of the extreme precipitation index over all grid points was statistically analysed for each of the seven regions, as well as the whole China. Therefore, we have a large number of samples for calculating the CDFs, stead of 30 values.

We computed the empirical cumulative distribution functions (ECDFs) of our data using histograms. To achieve a smoother representation of the distribution, we applied a Gaussian smoothing technique. By doing so, we were able to obtain smoothed representations of the empirical distributions, which provided clearer insights into the underlying patterns of the data.

As far as the uncertainty in the CDFs is concerned, in the original manuscript, we just placed emphasis on the uncertainty in the direct results of the UKESM, which we think is the most fundamental for our study. The model results (amount of precipitation) were validated with the observations (APHRODITE), and only means of the ensemble models were considered. In our revised manuscript, examination on the uncertainty in CDFs will be added by comparing the model results with that from the APHRODITE (for the historical period) and comparing the results among different models (for the future).

L185-194 put this into the previous section.

**Response:**

Thank you for your suggestion. We'd like to clarify that since we have already employed the Wilcoxon test for significance testing, the field significance calculations mentioned here are redundant, and was not used in this study. Therefore, we have removed this portion (L185-194) in the revised manuscript. We are sorry for this confusion.

L193 Include this statement in the figure caption instead of the text - and check which way you have represented significance, this is opposite from the figure.

**Response:**

As said above, the description of the method is removed.

L211 Comment on the increase in drought in west and Taiwan under G6sulfur.

**Response:**

At present, we are not sure about the mechanism about the increase in drought in west and Taiwan under G6sulfur. We think this is possibly because of the effects of relatively complex terrain in the areas, in particular, high mountains, which may disturb the atmospheric circulation and block transport of the moisture air. We will add some explanations in the revised manuscript.

L215 Change this statement to something like projected changes are similar to those of SSP2-45, meaning that the SAI simulations are approximately successful.

**Response:**

The statement has been revised as "G6sulfur (Fig. 3c) shows projected changes similar to those of SSP2-4.5 (Fig. 3b), indicating that the SAI simulations are approximately successful."

L246 There are no absolute values>100mm or no increases from the control period >100mm?

**Response:**

Thank you for pointing out this problem. We have clarified as "In the other three G6 models, the increase in RX5day is considerably smaller than that under SSP5-8.5, with none exceeding 100 mm compared to the control period (Fig.S2b-d)."

L250 remove (p-value <0.05) and every other instance - this has already been stated. Removed.

L251 remove which is generally. Is there really only one research paper on increases in extreme precipitation in this region?

The words are removed. Some other research papers (e.g., Qin and Xie, 2016; Peng et al., 2018) are include.

L257 Remove this sentence.

Removed.

L265 should this be depicting?

Yes, corrected. L266 remove sentence "The comparisons confirm…."

L278 remove "This suggests that G6sulfur…." This is the results section, so discussions aren't appropriate.

Removed. We appreciate it very much for pointing out the grammar and wording problems.

L280 Stick to reporting the differences between the simulations in this section, then interpret (with appropriate caveats) in the discussion sections and remove the sentences on the implications or efficacy of SSPs vs G6.

**Response:**

Thank you for the thoughtful suggestion. We have removed the interpret in line280-4, and now the revised sentence is "In comparison to SSP2-4.5 (Fig.S3b), G6sulfur exhibits an increase in RX5day, primarily in the region between 100°E and 120°E. For 'G6sulfur-G6solar'(Fig.S3c), positive values of RX5day are more pronounced in certain areas between 100°E and 120°E, especially in the low latitude zone between 20°N and 30°N.". In revising the manuscript, we will stick to reporting the differences between the simulations, and interpret (with appropriate caveats) in the discussion sections.

L291 Remove ameliorating

**Response:**

L291: "Ameliorating" has been removed.

Table 2: I am not sure that this adds to the interpretation of the results and could be removed. If retained, re-phrase as difference between G6 and SSP or something similar.

**Response:**

We think that Table 2 provides a useful summary of the results. If you look at Table 2, you get the general idea that SRM does 'good' things for the precipitation extremes, but 'bad' things for the droughts and dry days. That is a very useful take-home message. So, we would like to retain the table. Some descriptions and discussions about the results will be reconsidered and revised.

L311 Why is this interesting? Elaborate please.

**Response:**

We find this interesting because it highlights a unique pattern in the data. Despite observing mitigation effects in other regions, we notice that G6solar does not exhibit the same effect in the SC region. In other words, unlike other regions, G6solar does not mitigate the extreme high values of the RX1day index in the SC region. This suggests a nuanced relationship between the G6solar and their impact on RX1day in the SC region, warranting further investigation.

L325 It might be more meaningful to look at the relative changes (e.g. in percentage terms) rather than absolute values. With regard to the "arbitrary" regions, why are they somewhat arbitrary? Surely they relate to some geographical or political definition, the point to make is that they may not correspond with climatological regions. Note that smaller regions would just emphasise noise in the results.

**Response:**

We agree that relative changes such as in percentage terms would be more meaningful in some cases. However, in the case of our study, a great deal of the results are small values, and consequently, the relative changes (percentages) could be very large even for minor absolute changes. For this reason, we think the absolute changes can be more appropriate here. In the revised manuscript, we will add relative changes for some results or discussions where they are appropriate.

We agree that the conclusions would be different if based on different criteria for dividing the regions. Unfortunately, in this regard, there is no standard criteria for dividing the regions. In this study, the division of regions is a conventional way and has been widely adopted in many statistics reports and relevant studies (e.g., Luo et al., 2017; Fan et al., 2020; Yang and Shao, 2021; Liang et al. 2023). The sentence "It should be stressed here that the regions that are chosen for aggregation are somewhat arbitrary and the results could well change should smaller sub-regions be chosen for analysis." is not expressed accurately, which is removed. Some discussion on the uncertainty from regions' division will be added, including the scaling effect (smaller regions would just emphasise noise in the results).

L329/30 remove this sentence.

**Response:**

L329/30: The sentence has been removed.

Figure 6 as noted above the uncertainty across model members should be included in these curves. Please also check the colour scheme for colour blind appropriateness, and use the same x-axis for each variable for all regions (i.e. one x-axis for Rx1day, another for R95p). This also applies to Figure 9 and 12.

**Response:**

Yes, it would be more informative if the uncertainty across model members be included in the CDF curves in Figure 6. We do have the results of the three model members, and we tried to added results of each model along with the curves to indicate the uncertainty. However, since the curves are closely overlapped, we couldn't find a way to make the figure clear. Actually, the lines or colour blocks could be blended and overlapped together, and make the figure difficult to distinguish. For this reason, we gave up the idea of directly including the results of model members in the figures. Instead, we will add some statistical metrics of the model members in our revised manuscript. In this way, we think, at least to some extent, the uncertainty across model members can be indicated.

The colour scheme has been checked for colour blind appropriateness. For example, the purple line has been changed to black.

For the suggestion on using the same x-axis for each variable for all regions, there is also a difficulty. Because the range of index changes varies big across different regions, when plotting them on a large-scale x-axis, the curves with small range (or values) could be very close or even overlapped with each other, and difficult to be distinguished. For this reason, x-axis is not the same for all the regions. This is also the case for Figure 9 and 12.

L350 Decreases in CWD do not necessarily equate to reductions in precipitation intensity. You can only make this interpretation if there is a reduction in the total number of wet days AND an increase or no change in the annual total.

**Response:**

Thank you for this thoughtful comment. After recalculating the total number of wet days, the sentence in lines 349-350 has been revised. "While CWD is projected to decrease under SSP5-8.5, annual total precipitation amounts are projected to increase (Fig. 3). Additionally, the total number of wet days is expected to decrease in the future. These findings suggest that daily extreme precipitation intensity may rise in southern areas of China in the future (Zhu et al., 2018)."

 R50mm is not one of the formal ETCCDI indices, it is user defined.

**Response:**

Thank you for your reminding. We have revised the sentence as follows:
"The R50mm index is derived from the Rnnmm index, as suggested by ETCCDI. The Rnnmm index represents the count of precipitation above a user-chosen threshold. In this case, the threshold is set to 50 mm, as recommended by the China Meteorological Administration (CMA)."

L371 remove effectively ameliorates the

**Response:**

"effectively ameliorates the" has been removed.

L373-375 This is discursive and needs more references to support it (and moving to the discussion section). There is likely a combination at play including changes in the location of the jet streams and ITCZ, as well as interactions with topography and changes in maritime temperature gradients.

**Response:**

Thank you for your suggestion. More references in this regard will be included in our revised manuscript, and move it to discussion section.

L380/1 Remove this sentence.

**Response:**

L380/1: The sentence has been removed.

L387 CWD=200 days is right at the far end of the tail, I don't think it's appropriate to make this statement without any error estimates or uncertainty information. Further, the duration estimates of CWD and CDD add up to longer than a year - this is particularly obvious in comparison with Figure 12. Please check.

**Response:**

Thank you for the comments. As mentioned above, for the uncertainty in the CDFs, examination on the uncertainty in CDFs will be performed and added by comparing the model results with that from the APHRODITE (for the historical period) and comparing

the results among different models (for the future). Basing on that, relevant statements will be checked and the manuscript will be revised accordingly.

The phenomenon of the combined CDD and CWD exceeding 365 days in the same region arises because the high values of CWD and CDD may occur at different grid points, resulting in the possibility of the total exceeding 365 days.

L402 Should this refer to Figure 9? This statement would be more meaningful with uncertainty envelopes to clearly demonstrate whether there are or are not differences between each model.

**Response:**

Yes, the statement refers to Figure 9. Also, as mentioned previously, since the curves are closely overlapped, we couldn't find an appropriate way to include the results of model members in the figures. Instead, we will add some statistical metrics of the model members in our revised manuscript. In this way, we think, at least to some extent, the uncertainty across model members can be indicated.

L413 Remove, and would have the ….

Removed.

L458 It is not noteworthy that these yield similar results - that's the objective of the cooling. Remove this sentence.

The sentence has been removed.

L468 Remove last sentence of paragraph.

**Response:**

L468: The last sentence of the paragraph has been removed.

L475-481 There are other aspects related to drought risk - not least evaporation - that don't show up in the dry day count, such a strong statement about changes in drought risk aren't appropriate. See (Cheng et al., 2019; Dagon and Schrag, 2019, 2016) for more results related to climate intervention.

**Response:**

The sentence line476-481 has changed to "This reflects a potential decrease in drought risk in northwest regions and an increase in extreme drought events in low-latitude southeast coastal areas in the future according to four G6 simulations. Changes in precipitation affect soil moisture, thereby influencing evapotranspiration (ET) and

ultimately precipitation patterns. Assessing whether changes in DD and CDD affect drought risk also requires consideration of variations in ET and soil moisture (Cheng et al., 2019; Dagon and Schrag, 2016). Furthermore, solar radiation management (SRM) increases drought risk compared to SSP5-8.5 and SSP2-4.5 scenarios in northern regions (NEC, NC, and NWC)."

G6solar and G6 Sulfur are different ways of simulating a possible climate, we have no way of knowing whether one or the other is more valid without observations, and so can't be described as outperforming each other. See (Bednarz et al., 2022; Visioni et al., 2021) for more discussion on this.

**Response:**

We agree that we cannot tell which one is better between G6solar and G6sulfur. What we did in our study is comparing both the results of G6sulfur and G6solar in extreme precipitation events against the lower emission target (SSP2-4.5) or that in control period. We will add more discussion on this according to the references you suggested.

Section 4 I suggest condensing the bullet points to short concluding statements, which can then be followed by the explanations. Given that the focus of this article is on the climate intervention, it would also make more sense to emphasise those results rather than the future projections that have been published elsewhere. This is also the point to discuss how valid the results are with respect to other research - including experiments outside the GeoMIP project.

**Response:**

Thank you very much for the thoughtful and valuable suggestions. In revising the manuscript, we will consider to condense the bullet points, and add more explanations. In addition, more comparisons will be included and discussed with relevant results from published studies elsewhere. Thanks to so many thoughtful and valuable suggestions by you and other referees, we believe our manuscript will be improved greatly.

**References**

Donat, M., Alexander, L. V., Yang, H., Durre, I., Vose, R., Dunn, R. J., Willett, K. M., Aguilar, E., Brunet, M., and Caesar, J.: Updated analyses of temperature and precipitation extreme indices

since the beginning of the twentieth century: The HadEX2 dataset, Journal of Geophysical Research: Atmospheres, 118, 2098-2118, https://doi.org/10.1002/jgrd.50150, 2013.

Donat, M. G., Lowry, A. L., Alexander, L. V., O'Gorman, P. A., and Maher, N.: More extreme precipitation in the world's dry and wet regions, Nature Climate Change, 6, 508-513, https://doi.org/10.1038/nclimate2941, 2016.

Held, I. M. and Soden, B. J.: Robust responses of the hydrological cycle to global warming, J. Climate., 19, 5686-5699, https://doi.org/10.1175/JCLI3990.1, 2006.

Irvine, P. J., Kravitz, B., Lawrence, M. G., and Muri, H.: An overview of the Earth system science of solar geoengineering, Wiley Interdisciplinary Reviews: Climate Change, 7, 815-833, https://doi.org/10.1002/wcc.423, 2016.

Liang, J., Meng, C., Wang, J., Pan, X., and Pan, Z.: Projections of mean and extreme precipitation over China and their resolution dependence in the HighResMIP experiments, Atmospheric Research, 293, 106932, https://doi.org/10.1016/j.atmosres.2023.106932, 2023.

Miller, B. B. and Carter, C.: The test article, J. Sci. Res., 12, 135–147, doi:10.1234/56789, 2015.

Niu, S., Sun, M., Wang, G., Wang, W., Yao, X., and Zhang, C.: Glacier Change and Its Influencing Factors in the Northern Part of the Kunlun Mountains, Remote Sensing, 15, 3986, https://doi.org/10.3390/rs15163986, 2023.

Pendergrass, A. G. and Knutti, R.: The uneven nature of daily precipitation and its change, Geophysical Research Letters, 45, 11,980-911,988, https://doi.org/10.1029/2018GL080298, 2018.

Peng, Y., Zhao, X., Wu, D., Tang, B., Xu, P., Du, X., and Wang, H.: Spatiotemporal variability in extreme precipitation in China from observations and projections, Water, 10, 1089, https://doi.org/10.3390/w10081089, 2018.

Qin, P. and Xie, Z.: Detecting changes in future precipitation extremes over eight river basins in China using RegCM4 downscaling, Journal of Geophysical Research: Atmospheres, 121, 6802-6821, https://doi.org/10.1002/2016JD024776, 2016.

Ricke, K., Wan, J. S., Saenger, M., and Lutsko, N. J.: Hydrological consequences of solar geoengineering, Annual review of earth and planetary sciences, 51, 447-470, https://doi.org/10.1146/annurev-earth-031920-083456, 2023.

Smith, A. A., Carter, C., and Miller, B. B.: More test articles, J. Adv. Res., 35, 13–28, doi:10.2345/67890, 2014.

Tank, A. K. and Können, G.: Trends in indices of daily temperature and precipitation extremes in Europe, 1946–99, Journal of Climate, 16, 3665-3680, https://doi.org/10.1175/1520-0442(2003)016<3665:TIIODT>2.0.CO;2, 2003.

Zarnetske, P. L., Gurevitch, J., Franklin, J., Groffman, P. M., Harrison, C. S., Hellmann, J. J., Hoffman, F. M., Kothari, S., Robock, A., and Tilmes, S.: Potential ecological impacts of climate intervention by reflecting sunlight to cool Earth, Proceedings of the National Academy of Sciences, 118, e1921854118, https://doi.org/10.1073/pnas.1921854118, 2021.

---

## Author Comment (AC3)

**Dear reviewer,**

Thank you for your thoughtful and valuable comments, which are of great help for improving our paper. We have carefully considered all the comments by you and other reviewers, and prepared to revise the manuscript accordingly. A point-by-point response to your comments is provided below.

Thank you again for your valuable time and work, and we look forward to your positive decision on our paper.

With warm regards,

Ou Wang, on behave of all co-authors

**General comments:**

This study uses the UK Earth System Model (UKESM1) simulation results to examine the effect of solar radiation modification (SRM) geoengineering on precipitation extremes in China. As part of the GeoMIP project, UKESM1 was used to conduct two sets of SRM simulations: stratospheric aerosol injection (G6sulfur) and solar constant reduction (G6solar). Both G6sulfur and G6solar simulations are designed in such a way that global mean surface temperature under the scenario of SSP5-8.5 was brought down to the level under SSP2-4.5. Using a set of precipitation extreme indices, the authors investigated the effect of G6sulfur and G6solar on precipitation extremes for different regions of China. The authors found that compared to SSP5-8.5, both G6sulfur and G6solar ameliorate precipitation extremes over different parts of China, but increase drought risks in some northern part of China. The authors also compared the similarities and differences between precipitation extreme response to G6sulfur and G6solar for different regions of China. The analysis of this paper itself is largely sound, but I do not recommend its publication in ACP in the present form for the following reasons:

I see little science in this study. I have to say that I have not carefully examined the Results part, which is just the description of figures with little scientific insight. What the authors did is just to compare simulated precipitation extremes over different regions of China under SS5-8.5, SSP2-4.5, G6solar, and G6sulfur. Regional climate extremes are strongly dependent on the SRM scenarios (location, timing, and intensity of SAI and solar reduction). Also, regional climate extremes are strongly dependent on climate models. If one uses another climate model and/or another SAI strategy, most results presented in this paper might be different. At least, the authors should use multiple model results from GeoMIP instead of just one model. Also, the authors should try to investigate some science underlying the presented precipitation extreme comparisons. For example, why the difference between G6sulfur and G6solar? In the present form, this paper just presents simulation results from a specific model with little interpretations. At least for me, I see little science here.

**Response:**

We greatly appreciate your time and effort in reviewing our work and providing detailed and insightful comments on the manuscript, which are very helpful for improving the quality and clarity of our paper.

Indeed, in our present study, we mainly dedicated to compare the simulated precipitation extremes over different regions of China under different scenarios (SS5-8.5, SSP2-4.5, G6solar, and G6sulfur), with special focus on potential impacts of the SAI on precipitation extremes. We admit that there lack of sufficient interpretations on the results, particularly in terms of mechanism linking the response to the impacts. This is partially because of our limited knowledge in relevant fields. However, considering relatively scarce research and limited knowledge on the impact of SAI over East Asia, we think our findings are useful for deepening some understanding of the potential mitigation strategies of climate change. Thanks to the valuable suggestions provided by you as well as other referees, more interpretations will be added in our revised manuscript so as to make up the deficiency, at least to some extent.

We understand your concern regarding the use of only one climate model. We agree that the one model may not capture the diversity among different models, and employing multiple models for analysis could enhance the robustness of the findings and provide a more comprehensive perspective. Given this limitation in our study, we have realized that the title rather over-reached in terms of what is presented in the paper and changed the title to "Projected future changes in extreme precipitation over China under stratospheric aerosol intervention in the UKESM1 climate model".

We chose the UKESM1 model due to its extensive validation in prior studies and its reputation as a reliable tool for simulating climate dynamics. UKESM1, as described by Sellar et al. (2019), represents a significant advancement over its predecessor, HadGEM2-ES, with enhanced complexity in its components and internal coupling. The model performs admirably, maintaining a stable pre-industrial state and demonstrating strong agreement with observations across various contexts (Sellar et al., 2019). Furthermore, we conducted a validation of precipitation against APHRODITE data, which demonstrated that the model has a very credible performance. Previous studies also utilized the standalone UKESM1 model to evaluate the physical and biogeochemical state of the global ocean component (Yool et al., 2020), to assess the impact of both SAI and MCB on standard meteorological variables (Haywood et al., 2023), and to research other meteorological related (Haywood et al., 2022; Wells et al., 2023; Jones et al., 2020; Jones et al., 2022; Visioni et al., 2021).

In our study, while some biases were observed, UKESM1 reasonably captures precipitation patterns, particularly in eastern China, when compared with APHRODITE data from 1981-2010. Additionally, as noted in previous research (Liang and Haywood, 2023), UKESM1 is currently the sole model capable of providing outputs of pressure-level winds and specific humidity data every 6 hours, satisfying the requirements of the ARDT (Atmospheric Radiation Detection and Tracking) method. Furthermore, Tian et al.(2021) validated the UKESM1-0-LL simulation, demonstrating robust agreement between simulated and observed precipitation in China from 1961 to 2014, surpassing that of the CMIP6 multi-model ensemble (MME). Although acknowledging that a single model may not fully encompass the complexity of all climate variations, we

believe that UKESM1 offers a valuable initial assessment of the potential impacts of SRM strategies in different regions of China.

Certainly, future research could benefit from incorporating a broader range of models to validate our findings and further explore inter-model differences. This would contribute to a more comprehensive understanding of the effects of SRM on precipitation extremes and yield more robust conclusions.

About the difference between G6sulfur and G6solar, G6solar serves as a parallel experiment to G6sulfur, aiming to compare the effects of solar reduction with those of stratospheric aerosols. G6solar adopts the same setup as G6sulfur, but geoengineering is achieved through solar irradiance reduction. Specifically, the inter-model differences in the spatial distribution of forcing are expected to be smaller in G6solar than in G6sulfur, offering valuable insights into the effects of uncertainties in stratospheric sulfate aerosol transport (Kravitz et al., 2015). These have already been explained in L134-7 of the article.

Additionally, we would like to note that, when assessing impacts, it is common to focus on the most relevant metrics that are influenced. For example, the recent paper by Mari Tye (https://esd.copernicus.org/articles/13/1233/2022/):

*Tye, M. R., Dagon, K., Molina, M. J., Richter, J. H., Visioni, D., Kravitz, B., and Tilmes, S.: Indices of extremes: geographic patterns of change in extremes and associated vegetation impacts under climate intervention, Earth Syst. Dynam., 13, 1233–1257, https://doi.org/10.5194/esd-13-1233-2022, 2022.*

In Tye et al. (2022), a single model is used: Community Earth System Model (CESM1), for assessing extremes in temperature, precipitation and vegetation. Actually, Tye et al. (2022) provide no more than a discussion of impacts in the conclusion section and some general links to large scale dynamics that have been noted in other papers. While it should be more meaningful to delve into the causal mechanisms linking the response to the impacts, simply documenting impacts in extremes (as in Tye et al. (2022)) also appears worthy, at least to some. Other examples of published work that use a single model and focus on the impacts include, but are not limited to:

*Muthyala, R., Bala, G., & Nalam, A. (2018). Regional scale analysis of climate*

*extremes in an SRM geoengineering simulation, Part 2: temperature extremes. Current Science, 1036-1045.*

*Tilmes, S., Sanderson, B. M., & O'Neill, B. C. (2016). Climate impacts of geoengineering in a delayed mitigation scenario. Geophysical Research Letters, 43(15), 8222-8229.*

*Jones, A.C., Hawcroft, M.K., Haywood, J.M., Jones, A., Guo, X. and Moore, J.C., 2018. Regional climate impacts of stabilizing global warming at 1.5 K using solar geoengineering. Earth's Future, 6(2), pp.230-251.*

**Specific comments:**

Lines 36-59: This first paragraph of the Introduction part is very lengthy and most part is not directly relevant to the study here. For example, the detailed description of extreme precipitation in Zhengzhou and Beijing is not needed at all.
**Response:**

Thank you very much for pointing out the redundancy of the introduction part. In the revised manuscript, we will make (actually we have made) the first paragraph (lines 36-59) more concise and pertinent to the study, including shortening the description of extreme precipitation in Zhengzhou and Beijing.

Lines 61-72: This paragraph can also be substantially shortened and combined with the first paragraph.
**Response:**

According to your suggestion, we have substantially shortened the second paragraph (line 61-72), and combined it with the first paragraph.

Line 85: check the grammar here. ',the climate'
**Response:**

The error has been corrected. It should be "on the climate".

Lines 90-101: The use of 'prediction' in this paragraph is not appropriate.

**Response:**

  We have replaced "prediction" with "projection" in the paragraph.

**Response:**

  Yes, we agree that different scenario of SAI deployment would have different effects on the climate, in particular, the spatial and temporal distribution of precipitation. The statement (in lines 98-99) "SAI will exert a negative radiative forcing and reduce near-surface air temperature (including temperature means and extremes) (Pinto et al., 2020), and precipitation (Liu et al., 2021)" has be revised as: "Previous studies indicated SAI would exert a negative radiative forcing and reduce near-surface air temperature (including temperature means and extremes) (Pinto et al., 2020), and precipitation (Liu et al., 2021). However, the climate effects in terms of magnitude as well as spatial and temporal distribution depend largely on the scenario of SAI deployment. Furthermore, as suggested by some studies, although SAI can effectively counteract anthropogenic global warming at the global scale, it cannot fully offset the effects at regional scale (Tilmes et al., 2013; Niemeier et al., 2013; Simpson et al., 2019)".

**Response:**

  Thank you for pointing out this issue. In the response to your comment at the beginning, we have answered this point. Please refer to our explanation there.

**Response:**

  We'd like to clarify that since we have already employed the Wilcoxon test for the significance testing, the field significance analysis mentioned here are not necessary, and in fact, it was not used in this study. Therefore, the description (line185-194) should

be deleted. We are sorry for the confusion due to our carelessness.

**Line 199: The word of 'accurate' is not appropriate here.**

**Response:**

The "accurate" was replaced with "similar to the observed precipitation" in line 199.

**Line 133: 'reducing the solar constant or increasing SAI'. Check grammar and spelling here.**

**Response:**

The sentence should be changed to "increasing SAI".

**Lines 225-226: I don't understand what 'SAI is sensitive to global warming' means.**

**Response:**

The sentence has been revised as "This suggests that the effect of SAI on future precipitation is more widespread and remarkable compared to that of SSP5-8.5."

**Line 245: Where are 'the other three G6 models'?**

**Response:**

Sorry for our carelessness. The "models" should be "simulations", and it has been corrected in the revised manuscript. The other three simulations refer to that of SSP2-4.5, G6sulfur, and G6solar.

**References**

Haywood, J. M., Jones, A., Johnson, B. T., and McFarlane Smith, W.: Assessing the consequences of including aerosol absorption in potential stratospheric aerosol injection climate intervention strategies, Atmos. Chem. Phys, 22, 6135-6150, https://doi.org/10.5194/acp-22-6135-2022, 2022.

Haywood, J. M., Jones, A., Jones, A. C., Halloran, P., and Rasch, P. J.: Climate intervention using marine cloud brightening (MCB) compared with stratospheric aerosol injection (SAI) in the UKESM1 climate model, Atmos. Chem. Phys, 23, 15305-15324, https://doi.org/10.5194/acp-23-15305-2023, 2023.

Jones, A. C., Hawcroft, M. K., Haywood, J. M., Jones, A., Guo, X., and Moore, J. C.: Regional climate impacts of stabilizing global warming at 1.5 K using solar geoengineering, Earth's Future, 6, 230-251, https://doi.org/10.1002/2017EF000720, 2018.

Jones, A., Haywood, J. M., Jones, A. C., Tilmes, S., Kravitz, B., and Robock, A.: North Atlantic Oscillation response in GeoMIP experiments G6solar and G6sulfur: why detailed modelling is needed for understanding regional

implications of solar radiation management, Atmos. Chem. Phys., 21, 1287-1304,https://doi.org/10.5194/acp-21-1287-2021, 2020.

Jones, A., Haywood, J. M., Scaife, A. A., Boucher, O., Henry, M., Kravitz, B., Lurton, T., Nabat, P., Niemeier, U., and Séférian, R.: The impact of stratospheric aerosol intervention on the North Atlantic and quasi-biennial oscillations in the geoengineering model intercomparison project (GeoMIP) G6sulfur experiment, Atmos. Chem. Phys. 22, 2999-3016, https://doi.org/10.5194/acp-22-2999-2022, 2022.

Kravitz, B., Robock, A., Tilmes, S., Boucher, O., English, J. M., Irvine, P. J., Jones, A., Lawrence, M. G., MacCracken, M., and Muri, H.: The geoengineering model intercomparison project phase 6 (GeoMIP6): Simulation design and preliminary results, Geosci. Model Dev., 8, 3379-3392, https://doi.org/10.5194/gmd-8-3379-2015, 2015.

Liang, J. and Haywood, J.: Future changes in atmospheric rivers over East Asia under stratospheric aerosol intervention, Atmos. Chem. Phys, 23, 1687-1703, https://doi.org/10.5194/acp-23-1687-2023, 2023.

Muthyala, R., Bala, G., and Nalam, A.: Regional scale analysis of climate extremes in an SRM geoengineering simulation, Part 2: temperature extremes, Current Science, 1036-1045, http://www.jstor.org/stable/26495197., 2018.

Niemeier, U., Schmidt, H., Alterskjær, K., and Kristjánsson, J.: Solar irradiance reduction via climate engineering: Impact of different techniques on the energy balance and the hydrological cycle, J. Geophys. Res.-Atmos., 118, 11,905-911,917, https://doi.org/10.1002/2013JD020445, 2013.

Sellar, A. A., Jones, C. G., Mulcahy, J. P., Tang, Y., Yool, A., Wiltshire, A., O'connor, F. M., Stringer, M., Hill, R.,and Palmieri, J.: UKESM1: Description and evaluation of the UK Earth System Model, J. Adv. Model. Earth Sy., 11, 4513-4558, https://doi.org/10.1029/2019MS001739, 2019.

Simpson, I., Tilmes, S., Richter, J., Kravitz, B., MacMartin, D., Mills, M. J., Fasullo, J., and Pendergrass, A. G.: The regional hydroclimate response to stratospheric sulfate geoengineering and the role of stratospheric heating, J. Geophys. Res.-Atmos., 124, 12587-12616, https://doi.org/10.1029/2019JD031093, 2019.

Tian, J., Zhang, Z., Ahmed, Z., Zhang, L., Su, B., Tao, H., and Jiang, T.: Projections of precipitation over China based on CMIP6 models, Stoch Environ Res Risk Assess., 35, 831-848, 2021.

Tilmes, S., Fasullo, J., Lamarque, J. F., Marsh, D. R., Mills, M., Alterskjaer, K., Muri, H., Kristjánsson, J. E., Boucher, O., and Schulz, M.: The hydrological impact of geoengineering in the Geoengineering Model Intercomparison Project (GeoMIP), J. Geophys. Res.-Atmos., 118, 11,036-011,058, https://doi.org/10.1002/jgrd.50868, 2013.

Tilmes, S., Sanderson, B. M., and O'Neill, B. C.: Climate impacts of geoengineering in a delayed mitigation scenario, Geophys. Res. Lett., 43, 8222-8229, https://doi.org/10.1002/2016GL070122, 2016.

Tye, M. R., Dagon, K., Molina, M. J., Richter, J. H., Visioni, D., Kravitz, B., and Tilmes, S.: Indices of extremes: geographic patterns of change in extremes and associated vegetation impacts under climate intervention, Earth Syst. Dynam., 13, 1233–1257, https://doi.org/10.5194/esd-13-1233-2022, 2022.

Visioni, D., MacMartin, D. G., Kravitz, B., Boucher, O., Jones, A., Lurton, T., Martine, M., Mills, M. J., Nabat, P., and Niemeier, U.: Identifying the sources of uncertainty in climate model simulations of solar radiation modification with the G6sulfur and G6solar Geoengineering Model Intercomparison Project (GeoMIP) simulations, Atmos. Chem. Phys., 21, 10039-10063, https://doi.org/10.5194/acp-21-10039-2021, 2021.

Wells, A. F., Jones, A., Osborne, M., Damany-Pearce, L., Partridge, D. G., and Haywood, J. M.: Including ash in UKESM1 model simulations of the Raikoke volcanic eruption reveals improved agreement with observations, Atmos. Chem. Phys., 23, 3985-4007, https://doi.org/10.5194/acp-23-3985-2023, 2023.

Yool, A., Palmiéri, J., Jones, C. G., de Mora, L., Kuhlbrodt, T., Popova, E. E., Nurser, A. G., Hirschi, J., Blaker, A. T., and Coward, A. C.: Evaluating the physical and biogeochemical state of the global ocean component of UKESM1 in CMIP6 historical simulations, Geosci Model Dev 2020, 1-68, https://doi.org/10.5194/gmd-14-3437-2021, 2020.

---

## Author Comment (AC4)

Dear reviewer,

Thank you very much for your thoughtful comments and constructive suggestions. We appreciate your recognition of the potential value in our study and your encouragement of using innovative approaches that integrate measurements and models. We have considered all the comments by you and other reviewers carefully, and will revise (actually, we have mostly revised) the manuscript comprehensively. A point-by-point response to each of your comments is provided below.

Thank you again for your valuable time and work, and we look forward to your positive decision on our paper.

With warm regards,

Ou Wang, on behave of all co-authors

**General comments:**

I think this could be a useful study with some work. I always like creative ways of integrating measurements and models. The analysis is also carefully done and focuses on clearly important issues (extremes).

While I don't dispute any of the findings, my biggest issue is with the explanations. There is a lot of reporting of the results but not much interpretation other than (sometimes) speculating about mechanisms. Given that the authors have a great deal of climate model output at their disposal, they could look into some of these mechanisms. I would point out specific examples, but this seems to be a general issue in Section 3.

Also, there is a lot of discussion of different indices, but they mostly show the same thing. That's not a problem, but the way you're describing them makes it seem like you're going through a laundry list of indices. I'd like to see more insight. Digging into the results in Table 2 would be interesting. For example, _why_ does CWD not behave like the other indices? What's special about those two regions that have the

**Response:**

First of all, we'd like to note that we have realized that the title of our original manuscript was rather over-reached in terms of what is presented in the paper and changed it to "Projected future changes in extreme precipitation over China under stratospheric aerosol intervention in the UKESM1 climate model", which is more relevant to the results of the article.

We have to admit that interpretation of the results and speculation about the mechanisms are insufficient, partially because our knowledge in this area is still limited. Though it can't be fully made up for the time being, we are trying to improve it by adding some more interpretations and discussions in our revised manuscript. Nevertheless, the main purpose is to demonstrate the possible impacts of SAI by comparisons among different scenarios. Really, it would be more interesting to discuss on causal mechanisms linking the response to the impacts, considering the relatively scarce research and limited knowledge on the impact of SAI over East Asia, we think simply documenting the impacts in extremes (as in Tye et al. (2022)) also appears worthy, at least to some.

Thank you for your suggestion for digging into the results in Table 2. We are going to add following discussions/statements:

At line 394-397: "The positive value in NC and NWC indicates that SAI experiments produce results that are closer to the PD conditions, which suggests the effect of SAI in the northwest arid regions is more significant in reducing precipitation compared to SSP5-8.5 scenario. However, the relative effect is not obvious due to the small magnitude of CWD in these regions."

At Line 457-459: "As shown in Table 2, the DD is positive in the SC region, meaning G6sulfur effectively lowers the threshold for extreme DD events compared to SSP5-8.5. This suggests that the SAI is more effective for DD in the humid region.

At Line 469-472: "The positive value of the CDD index in the SC and SWC regions indicates that G6sulfur notably reduces the threshold for extreme CDD events compared to SSP585, thereby approaching PD conditions in these regions. This suggests that G6sulfur has the potential to mitigate the decrease of CDD in the SC and

alleviate the increase of CDD in the SWC. The ameliorating effect of DD and CDD in the SC region under G6sulfur may be related to the strengthening of the anti-cyclonic circulation associated with the subtropical gyre, which appears to increase under G6 compared to SSP5-8.5 (Liang and Haywood, 2023). This intensification results in an increased inflow of moist air from the ocean at 850hPa and a greater supply of moisture to the southern region of the area."

**Specific comments:**

Figure 1 and Section 2.1: Any reason you don't include the Tibetan plateau?
**Response:**

As shown in Figure1 in the paper, the Tibetan plateau is indeed represented in the brown areas, divided into two parts in SWC and NWC. The division of the regions follows a conventional approach that has been widely adopted in many statistical reports and relevant studies (e.g., Luo et al., 2017; Fan et al., 2020; Yang and Shao, 2021; Liang et al., 2023), despite that there is no standard criteria for such divisions. We will clarify this in the revised manuscript.

Lines 169-182: This seems like a long way of saying that you used survival functions, which are a perfectly reasonable thing to use for what you want to do.
**Response:**

We appreciate your recognition of the method we used in the article. The description might be too lengthy, and we considering to make it more concise.

Lines 186-187: This is not consistent with my understanding of what field significance does. I would appreciate more description as to what you mean.
**Response:**

Thank you for pointing this out. We'd like to clarify that since we have already employed the Wilcoxon test for the significance testing, the field significance analysis mentioned here are not necessary, and in fact, it was not used in this study. Therefore, the description (line185-194) should be deleted. We are sorry for the confusion due to

our carelessness.

**Response:**

Thank you for your valuable suggestion. Really, it would be meaningful to show the bias in terms of relative changes, in addition to the absolute value. When trying to do so (adding a panel showing the bias as percentages), a problem we found is that, since a great deal of the results (daily precipitation) are small values, the relative changes (percentages) could be very large even for minor absolute changes (particularly for those in western and northern areas), and this would make the results confused.

For this reason, a panel of scatter plots comparing the observations with the model results are added, as shown below. The new panel has 4 sub-panels, with each comparing the observations with the mean of ensemble model and the three model members, respectively. The observations (daily precipitation) during the control period in China were classified in to several intervals: P10 (the smallest 10%), P10-50, P50-90, P90-95, and P95 (the largest 5%). In order to indicate the bias as a percent, relative changes (compared to the observations) for different intervals have been calculated, as listed in following table.

The scatter plots indicate a close relationship between the observations and the model results. However, the model results are generally higher than observations, possibly because of the different resolution of the data. Since our study has been mainly focused on the relative changes between the future results and that of control period for different scenarios, the systematic bias would not affect the conclusions significantly. As expected, relative changes are very large at small values (below the 10th percentile), both for the ensemble mean and the model members. For the results at the 10-50th and 50-90th percentiles, relative changes are around 30%. When larger than the 95th percentile, relative changes are relatively small, near 15%. The differences among ensemble members are not significant, which suggests the uncertainty in the ensembled results is reasonable and acceptable.

What mentioned above will be included in our revised manuscript.

[Figure]

**Figure 2(d) Scatter plots between the observations and model results at different level of precipitation during the control period (CP).**

The observations were classified into several intervals: P10 (the smallest 10%), P10-50, P50-90, P90-95, and P95 (the largest 5%).

Table: Relative changes of the model results (compared to the observations)

| intervals | Ensemble mean | r1i1p1f2 | r4 i1p1f2 | r8i1p1f2 |
|---|---|---|---|---|
| <P10 | 89.81% | 93.95% | 89.44% | 86.04% |
| P10-50 | 30.05% | 30.38% | 31.85% | 27.13% |
| P50-90 | 30.50% | 28.95% | 31.36% | 31.16% |
| P90-95 | 24.03% | 22.79% | 24.85% | 24.44% |
| >P95 | 15.76% | 15.09% | 16.27% | 15.92% |

Lines 225-226: I'm not sure what this means. SAI is sensitive to global warming?

**Response:**

We are sorry for the mistakes. The sentence has been revised as "This suggests that the effect of SAI on future precipitation is more widespread and remarkable compared to that of SSP5-8.5,"

Line 265:   Typo (depicting)

**Response:**

We have corrected 'dipicting' to 'depicting. '

Line 293:   Aggregated how?

**Response:**

Sorry for the wording mistake. The "aggregated" should be "presented". The sentence (lines 291-293) should be "For each index, the differences between the maximum values under the G6sulfur and SSP5-8.5 scenarios compared to the current period are presented in Table 2."

Line 295: Be more specific about "the opposite". Also, what are 0 values in the table? (I can figure it out, but you need a description.)

**Response:**

To be more specific about "the opposite" and "0 values", detailed explanation is provided: "Specifically, positive values indicate that the index under the G6sulfur are closer to that under the current period, compared to the SSP5-8.5 scenario, meaning that the mitigation effect of G6sulfur is significant. On the contrary, negative values mean the enhanced effect under G6sulfur compared to SSP5-8.5 scenario. In addition, 0 values in the table represent instances where there is no difference between the maximum index values under G6sulfur and SSP5-8.5, suggesting no significant impact on extreme precipitation events."

Lines 324-325:   It's difficult to put these numbers in context.   Is 100 mm a lot for these regions?

**Response:**

The 100 mm here is in comparison with other regions. The sentence (lines 324-325) should be revised as: "in EC and CC, there are substantial decreases of more than 100mm compared to other regions, whereas in NEC and NC, there are some increases of about 50mm."

Line 341:   I don't know if "effectively mitigates" is the correct phrasing.   Be more specific.

**Response:**

The sentence "SRM effectively mitigates the increase of RX1day, RX5day and R95p compared to SSP5-8.5 scenario in all regions" has been changed to "SRM results are encouraging, showing a reduction in the efficacy of detrimental extreme events, similar to the lower emissions target of SSP2-4.5"

Lines 391ff:   I'll be honest, I had a hard time with this entire paragraph.   I'm really not sure I understand it.

**Response:**

Thank you for your feedback on this paragraph. We have revised it to improve clarity and coherence: "It is notable that in NC, and SC, G6sulfur (purple) provides similar results to the SSP2-4.5. When combined with Fig.8b and c (less than 2 days), suggests that G6sulfur yields statistically similar outcome to that of SSP2-4.5 in NC and SC. In SWC, the 395 purple line and orange line also closely tracks the green line, indicating a similar cumulative distribution of CWD between G6sulfur and SSP2-4.5, and between G6solar and SSP2-4.5. However, there is an uneven spatial distribution, as seen in Fig.8b, and Fig.S5. Consequently, SRM is not expected to reach the levels of SSP2-4.5 in SWC. Interestingly, for EC, SSP2-4.5 yields almost identical statistics to SSP5-8.5, while both G6sulfur and G6solar show an increase compared to SSP scenarios. However, the negative values of CWD in EC in Table 2 and S2 indicate that SRM strategies cannot ameliorate the high values of CWD 400 in the EC region."

Line 520:   ETCCDI

**Response:**

Thank you so much for the careful suggestion. The "ETDCCI" has been corrected to "ETCCDI"

**Reference**

Liang, J. and Haywood, J.: Future changes in atmospheric rivers over East Asia under stratospheric aerosol intervention, Atmos. Chem. Phys., 23, 1687–1703, https://doi.org/10.5194/acp-23-1687-2023, 2023.

Tye, M. R., Dagon, K., Molina, M. J., Richter, J. H., Visioni, D., Kravitz, B., and Tilmes, S.: Indices of extremes: geographic patterns of change in extremes and associated vegetation impacts under climate intervention, Earth Syst. Dynam., 13, 1233–1257, https://doi.org/10.5194/esd-13-1233-2022, 2022.

---

## Author Response (AR1)

**Dear editor and referees,**

First of all, we greatly appreciate your thoughtful and valuable comments and suggestions, which have been a great help in improving our manuscript. After careful consideration and revision, we think all the comments have been appropriately addressed, and the revised manuscript could meet the quality standards of ACP. Below is a point-by-point response to all referee comments, generally including comments from referees, our responses, and changes in the manuscript. In addition, a marked-up manuscript version showing the changes made (using track changes in Word) is provided following the response.

Once again, we are very grateful for your kind help, and looking forward to your further comments and positive decision on our manuscript.

Thank you for your consideration of publication.

With warm regards,

Qin`geng Wang, Prof. (State Key Laboratory of Pollution Control and Resources Reuse, School of Environment, Nanjing University, Nanjing, 210023, China)

Jim M. Haywood, Prof. (Department of Mathematics, Faculty of Environment, Science and the Economy, University of Exeter, Exeter EX4 4QE, UK)

**Response to Referee #1**

**General comment:**

This paper examines the impact of changes in different measures of precipitation over China under a high emissions scenario and with a small ensemble of climate intervention simulations at the end of the century. One climate model (UKESM1) is used for comparison with two different realisations of stratospheric aerosol injection - G6solar, using a constant solar dimming, and G6sulfur, with gradually increasing injections of sulfur into the stratosphere. The paper is well organised and generally clearly and well written. My main criticism is that the discussion does not evaluate the results with respect to other research

that was carried out from other experiments such as GLENS. Suggestions for other relevant literature is at the end together with references made in the comments below.

**Major comments:**

Consider reducing the content of Section 2.2. A lot of this is repetition from the literature that is cited and doesn't necessarily need to be included in this article.

**Response:**

Thank you very much for your suggestion. We have realized the repetitions from the literature are not necessary, and revised the section 2.2, as in lines 139-166.

To avoid confusion, consider changing any reference to "present-day" to the control period and sticking with this consistently, rather than switching between 'historical', 'baseline' and 'present-day'. The WMO has adopted 1990-2020 as the "current climate period", which suggests that the period used in this article is historical.

**Response:**

Thank you. We have changed "present-day" to "control period" throughout the manuscript.

Is there a benefit in the data validation against APHRODITE? The article does not include an assessment of extreme precipitation metrics with respect to observations, nor do the results or conclusions refer back to the observations. I think you could drop this, and instead refer to other assessments of the validity of extreme precipitation in climate models such as (Sillmann et al., 2013; Donat et al., 2020; Tebaldi et al., 2021)

**Response:**

Thank you for this important comment. Indeed, this study has got a great benefit in the data validation against APHRODITE. The observations were used to validate the direct results of the model (i.e., amount of precipitation), which we think is the most fundamental for this study. To further compare the results between simulations and observations, particularly focusing on extreme precipitation values, in the

revised manuscript (line 237), a scatter plot between the ensemble simulations and observations is provided as a new panel in Figure 2 (Fig.2d). In addition, the scatter plots between the three model results and the observations are provided in Figure S1. In the scatter plots, the daily observed precipitations were classified into several intervals: P10 (the smallest 10%), P10-50, P50-90, P90-95, and P95 (the largest 5%). The scatter plots (also shown below) indicate a close relationship between the observations and the simulations. However, the simulations are generally higher than observations, possibly because of the different resolution of the data. Since our study has been mainly focused on the relative changes between the future results and that of control period for different scenarios, the systematic bias would not affect the conclusions significantly. The above explanation is also added in the revised manuscript (lines 230-235).

[Figure]

**Figure 2(d) Scatter plot between the observations and simulations at different level of precipitation.**
The observations were classified into several level (intervals): P10 (the smallest 10%), P10-50, P50-90, P90-95, and P95 (the largest 5%).

In order to indicate the bias as a percent, relative changes (compared to the observations) for different intervals have been calculated, and the results are listed in a new Table2 added in the revised manuscript

75 (also shown below). As expected, the relative changes are very large at small values (below the 10th percentile), both for the ensemble mean and the model members. For the results at the 10-50th and 50-90th percentiles, the relative changes are around 30%. When larger than the 95th percentile, the relative changes are relatively small, near 15%. The differences among ensemble members are not significant, which suggests the uncertainty in the ensembled results is reasonable and acceptable. Please see lines
80 248-255 for the table and discussions.

**Table 2: Relative changes of the model results (compared to the observations)**

| intervals | Ensemble mean | r1i1p1f2 | r4 i1p1f2 | r8i1p1f2 |
|-----------|---------------|----------|-----------|----------|
| <P10 | 89.81% | 93.95% | 89.44% | 86.04% |
| P10-50 | 30.05% | 30.38% | 31.85% | 27.13% |
| P50-90 | 30.50% | 28.95% | 31.36% | 31.16% |
| P90-95 | 24.03% | 22.79% | 24.85% | 24.44% |
| >P95 | 15.76% | 15.09% | 16.27% | 15.92% |

Given that this research uses one climate model, and three ensemble members for each scenario, it isn't really appropriate to state that G6sulfur/G6solar abates or ameliorates climate change as depicted by
85 SSP5-8.5. There aren't sufficient model members to remove model uncertainty, and without observations we do not know which model realisation (if any) adequately simulates the effects of SAI. All you can state with confidence is that using this methodology and data, the SAI experiments produce results consistent with a lower emissions target. I would prefer to see all of the statements on improvements with respect to climate change removed, or at least reduce the emphasis of the statements.

90 **Response:**

We agree with you, and thank you very much for the suggestion that makes our paper more rigorous and scientific. Really, considering many possible uncertainties in the climate models, as well as in the research scenarios, we cannot assert that G6sulfur/G6solar abates or ameliorates climate change as depicted by SSP5-8.5. Accordingly, relevant statements in the manuscript have been removed.

95

**Minor comments:**

**Response:**

Changed (Line 32).

**Response:**

The sentence "While the results from both G6sulfur and G6solar show encouraging abatement of many of the impacts on detrimental extreme events that are evident in SSP5-8.5 there are some exceptions." has been change to "While the G6sulfur and G6solar show encouraging potential abatement of the impacts from detrimental extreme events which are similar with the lower emissions target of SSP2-4.5, there are some exceptions." Please see lines 33-35.

**Response:**

Removed (line 35).

L42 is the higher risk of flooding associated with increased extreme precipitation? Remove the time periods studied, this is implicit.

**Response:**

Yes, the higher risk of flooding is indeed associated with increased extreme precipitation. This is indicated in the results by Ying et al. (2014), where the flood risk is understood as an extreme climate index. Indeed, the time periods are implicit, and has been removed (line 47).

L44-60 is this level of detail on historic events warranted? You do not examine the changes in jets or other sources of extremes.

**Response:**

The details on historic events are mostly quoted from published papers or media reports. To be honest,
125  we could not warrant their reliability since we have not conducted deep investigation in this regard. Now, according to the comments by another referee (RC2), we have realized that the detailed description on the historic events is not directly relevant to the study here, and not necessary. Therefore, relevant contents have been deleted or shortened in our revised manuscript (lines 50-66).

130  L44 Why was summer of 2020 anomalous - it seems in keeping with the other extreme events you reported.
**Response:**
The using of the word "anomalous" was not appropriate. Besides 2020, flooding events also frequently happened in other years. In the revised manuscript, this has been corrected (line 49). In addition, as mentioned above, the detailed description on the historic events has been shortened.
135

L61 change appears to has, and cite relevant literature such as (Donat et al., 2016; Pendergrass and Knutti, 2018), which also discuss changes in the hydrological cycle.
**Response:**
The sentence has been changed as: "On a global scale, climate change has been influencing
140  hydroclimatic conditions (Donat et al., 2016; Pendergrass and Knutti, 2018)." Please see lines 66-67.

L64 Should causes go before faster?
**Response:**
Relevant contents have been deleted as mentioned above.
145

L66 Update this to the more nuanced and recent research that shows extreme precipitation generally goes up everywhere (Pendergrass and Knutti, 2018)
**Response:**
It has been updated according to Pendergrass and Knutti (2018). Please see lines 71-72.
150

L79 SAI does not mitigate anthropogenic climate warming, it may mitigate some of the impacts.

**Response:**

You are right. The sentence has been revised as: "To some extent, SAI partially counteract climate warming by injecting reflective particles, or their gaseous precursors, into the stratosphere (Zarnetske et al., 2021)." Please see lines 86-87.

L79-89 This paragraph needs rephrasing to explain that SAI is premised on reproducing the effects associated with volcanic eruptions. However, you do not need to list the different volcanic eruptions themselves - just point to a large body of literature that supports these effects.

**Response:**

Thank you for this comment which has helped make our manuscript more concise and less verbose. The paragraph has been changed as: "To some extent, SAI can partially counteract climate warming by injecting reflective particles, or their gaseous precursors, into the stratosphere (Zarnetske et al., 2021). In addition to reducing the temperature, SAI also influences tropospheric and stratospheric ozone, terrestrial ecosystem, terrestrial carbon, and hydrological cycle by changing the physical climate system and atmospheric chemistry. Numerous studies support these effects associated with volcanic eruptions and their simulation through SAI techniques (e.g. Imai et al., 2020; Mclandress et al., 2011; Jones et al., 2018, 2020; Liang and Haywood., 2023; Lee et al., 2021; Plazzotta et al., 2019; Visioni et al., 2022)." Please see lines 86-91 of our revised manuscript.

L99 include other recent research that explored changes in temperature and precipitation in other SAI experiments not just the GeoMIP archive (e.g. Tye et al., 2022; Simpson et al., 2019).
L100 What about (Tew et al., 2023)?

**Response:**

Thank you for the suggestion. We have incorporated other recent research, including studies exploring changes in temperature and precipitation in various SAI experiments beyond the GeoMIP archive, such as those by Tye et al. (2022), Simpson et al. (2019), and Tew et al. (2023). Please see lines 113, 116, 119.

180 L105 remove maximise the signal-to-noise in the simulations as

L114 remove according

**Response:**

    Removed

185 L126 See comment above, but at the very least remove The GeopMIP G6sulfur simulations that reduce global mean temperatures from the SSP5-8.5 scenario to the SSP2-4.5 are described in detail elsewhere.

**Response:**

    According to your above suggestions, the manuscript has been revised. Please see lines 147-152.

190 L143-145 remove this last sentence.

**Response:**

    Removed.

L154 I believe that the extreme indices were defined by the WCRP not IPCC.

195 **Response:**

    Thank you for pointing out the mistake, and we have corrected it in the manuscript (line 179).

Table 1: the authors should be Frich and Klein Tank. Also refer to (Sillmann et al., 2013; Zhang et al., 2011) for the correct definitions

200 **Response:**

    The mistake has been corrected (line 184).

How did you calculate the 95th percentile? Did you bootstrap the individual years to avoid data inhomogeneities (Zhang et al., 2005)

205 **Response:**

    For each grid, the 95th percentile was calculated based on 30 years (1981-2010) of daily precipitation data. We calculated the 95th percentile directly without using bootstrapping methods, as recommended

for calculating temperature indices (https://www.climdex.org/learn/indices/#index-TX90p). Relevant explanation has been added in the manuscript (line 285-286).

210

L161 This may not be relevant if you remove the Aphrodite data as suggested above. However, I am concerned about regridding the larger data to the smaller grid. No additional information is gained in this respect (just several grid boxes with the same values) and may show errors and biases that are not true. Instead it would be more robust to regrid the observations to match that of UKESM. See

215 https://climatedataguide.ucar.edu/climate-tools/regridding-overview for more information.

**Response:**

Thank you for your valuable suggestion. The observations were re-gridded to match that of the UKESM (line 187-188).

220 L165 remove instead of the more commonly used Student's t=test. Wilcoxon Rank Sum Test work as

**Response:**

The text was removed.

L167 change "with p-value <0.05 suggesting" to "with a 5% confidence level of"

225 **Response:**

The expression was changed (line 193).

L169-182 How did you establish the CDFs? Did you fit distributions, or are these empirical CDFs from the data? Did you examine the uncertainty in the CDFs, and were they fitted for each model member

230 (correct) or the model mean (as the figures suggest)? I am also wary about CDFs for very small sample sizes - i.e. 30 values of the annual maximum rainfall.

**Response:**

Thank you for pointing out this problem that we didn't explain it clearly about how we established the CDFs. In fact, for establishing the CDFs, firstly, for an extreme precipitation index at each grid point,

235 the yearly mean of the ensemble model members was calculated. Then, the annual extreme precipitation

indices for each grid point was obtained by averaging over yearly means during the 30 years. Finally, the cumulative probability distribution of the extreme precipitation index over all grid points was statistically analysed for each of the seven regions, as well as the whole China. Therefore, we have a large number of samples for calculating the CDFs, instead of 30 values.

We computed the empirical cumulative distribution functions (ECDFs) of our data using histograms. To achieve a smoother representation of the distribution, we applied a Gaussian smoothing technique. By doing so, we were able to obtain smoothed representations of the empirical distributions, which provided clearer insights into the underlying patterns of the data.

As far as the uncertainty in the CDFs is concerned, in the original manuscript, we just placed emphasis on the uncertainty in the direct results of the UKESM, which we think is the most fundamental for our study. The model results (amount of precipitation) were validated with the observations (APHRODITE), and only means of the ensemble models were considered. In our revised manuscript, more comparisons between the model results and the observations have been provided. Please see lines 230-255.

L185-194 put this into the previous section.

**Response:**

Thank you for your suggestion. We'd like to clarify that since we have already employed the Wilcoxon test for significance testing, the field significance calculations mentioned here are redundant, and was not used in this study. Therefore, we have removed this portion in the revised manuscript. We are sorry for this confusion.

L193 Include this statement in the figure caption instead of the text - and check which way you have represented significance, this is opposite from the figure.

**Response:**

As said above, the description of the method has been removed.

L211 Comment on the increase in drought in west and Taiwan under G6sulfur.

**Response:**

265     Thank you for pointing out these detailed signals. We thank the reviewers for pointing out these detailed signals. These signals of increase in drought are possibly linked to the intensification and northward shift of the Western Pacific Subtropical High; however, due to the limited reliability of small-scale signals in the non-storm-resolving GCMs like UKESM1, we decided not to discuss these small-scale signals in the revised manuscript.

270

L215 Change this statement to something like projected changes are similar to those of SSP2-45, meaning that the SAI simulations are approximately successful.

**Response:**

The statement has been changed to "G6sulfur (Fig. 3c) shows projected changes are similar to those

275     of SSP2-4.5 (Fig. 3b), indicating that the SAI simulations are approximately successful." Please see lines 262-263.

L246 There are no absolute values>100mm or no increases from the control period >100mm?

**Response:**

280     Thank you for pointing out this problem. We have clarified it as "In the other three G6 scenarios, the increase in RX5day is considerably smaller than that under SSP5-8.5, with none exceeding 100 mm compared to the control period (Fig.S3a-d)." Please see lines 294-295.

L250 remove (p-value <0.05) and every other instance - this has already been stated.

285     Removed.

L251 remove which is generally. Is there really only one research paper on increases in extreme precipitation in this region?

The words are removed. Some other research papers (e.g., Qin and Xie, 2016; Peng et al., 2018) are

290     included in revised manuscript (line 300).

Removed.

We have corrected 'dicipting' to 'depicting. ' Line 314.

The sentence has been removed.

300

Removed.

**Response:**

Thank you for the thoughtful suggestion. To stick to reporting the differences, the sentence has been
revised as "In comparison to SSP2-4.5 (Fig.S4b), G6sulfur exhibits an increase in RX5day, primarily in
310   the region between 100°E and 120°E. For 'G6sulfur-G6solar'(Fig.S4c), positive values of RX5day are
more pronounced in certain areas between 100°E and 120°E, especially in the low latitude zone between
20°N and 30°N." Please see lines 327-331. In addition, relevant interpretations (with appropriate caveats)
have been condensed in the discussion section (Lines 633-637).

**Response:**

"Ameliorating" has been removed.

Table 2: I am not sure that this adds to the interpretation of the results and could be removed. If retained,
320    re-phrase as difference between G6 and SSP or something similar.

**Response:**

We think that Table 2 (now is Table 3) provides a useful summary of the results. If you look at Table 3, you get the general idea that SRM does 'good' or 'bad' things on the extreme indices threshold. That is a very useful take-home message. So, we would like to retain the table. Some descriptions and
325    discussions about the results have been revised (Lines 343-349, 441-443, 509-510).

L311 Why is this interesting? Elaborate please.

**Response:**

We find this interesting because it highlights a unique pattern in the data. Despite observing
330    mitigation effects in other regions, we notice that while G6solar mitigate the overall RX1day, it exacerbates the maximum RX1day values beyond SSP5-8.5. This suggests a nuanced relationship between the G6solar and their impact on RX1day in the SC region, warranting further investigation.

L325 It might be more meaningful to look at the relative changes (e.g. in percentage terms) rather than
335    absolute values. With regard to the "arbitrary" regions, why are they somewhat arbitrary? Surely they relate to some geographical or political definition, the point to make is that they may not correspond with climatological regions. Note that smaller regions would just emphasise noise in the results.

**Response:**

We agree that relative changes such as in percentage terms would be more meaningful in some cases.
340    However, in the case of our study, a great deal of the results are small values, and consequently, the relative changes (percentages) could be very large even for minor absolute changes. For this reason, we think the absolute changes can be more appropriate here.

We agree that the conclusions would be different if based on different criteria for dividing the regions. Unfortunately, in this regard, there is no standard criteria for dividing the regions. In this study, the
345    division of regions is a conventional way and has been widely adopted in many statistics reports and relevant studies (e.g., Luo et al., 2017; Fan et al., 2020; Yang and Shao, 2021; Liang et al. 2023). The

sentence "It should be stressed here that the regions that are chosen for aggregation are somewhat arbitrary and the results could well change should smaller sub-regions be chosen for analysis." is not expressed accurately, which has been removed.

350

L329/30 remove this sentence.

**Response:**

The sentence has been removed.

355 Figure 6 as noted above the uncertainty across model members should be included in these curves. Please also check the colour scheme for colour blind appropriateness, and use the same x-axis for each variable for all regions (i.e. one x-axis for Rx1day, another for R95p). This also applies to Figure 9 and 12.

**Response:**

Yes, it would be more informative if the uncertainty across model members be included in the CDF
360 curves in Figure 6. We do have the results of the three model members, and we tried to added results of each model along with the curves to indicate the uncertainty. However, since the curves are closely overlapped, we couldn't find a way to make the figure clear. The lines or colour blocks could be blended and overlapped together, and make the figure difficult to distinguish. For this reason, we gave up the idea of directly including the results of model members in the figures. Instead, we have added some statistical
365 metrics of the model members (as a new table) in our revised manuscript (lines 230-255). In this way, we think, at least to some extent, the uncertainty across model members can be indicated.

The colour scheme has been checked for colour blind appropriateness.

For the suggestion on using the same x-axis for each variable for all regions, there is a difficulty. Because the range of index changes varies big across different regions, when plotting them on a large-
370 scale x-axis, the curves with small range (or values) could be very close or even overlapped with each other, and difficult to be distinguished. For this reason, x-axis is not the same for all the regions. We use the same x-axis in Figure 6 and appropriately adjusted the x-axis in Figure 9 and Figure12, according to different range of index changes.

L350 Decreases in CWD do not necessarily equate to reductions in precipitation intensity. You can only make this interpretation if there is a reduction in the total number of wet days AND an increase or no change in the annual total.

**Response:**

Thank you for this thoughtful comment. The sentence has been removed.

L356 R50mm is not one of the formal ETCCDI indices, it is user defined.

**Response:**

Thank you for your reminding. We have revised the sentence as follows:

"The R50mm index is derived from the Rnnmm index, as suggested by ETCCDI. The Rnnmm index represents the count of precipitation above a user-chosen threshold. In this case, the threshold is set to 50 mm, as recommended by the China Meteorological Administration (CMA)." See lines 404-405.

L371 remove effectively ameliorates the

**Response:**

It has been removed.

L373-375 This is discursive and needs more references to support it (and moving to the discussion section). There is likely a combination at play including changes in the location of the jet streams and ITCZ, as well as interactions with topography and changes in maritime temperature gradients.

**Response:**

Thank you for your suggestion. More references in this regard have been included in our revised manuscript (lines 423-426). Because the discussion section provides a summary of the findings, we believe it is more appropriate to include it there.

L380/1 Remove this sentence.

**Response:**

The sentence has been removed.

 CWD=200 days is right at the far end of the tail, I don't think it's appropriate to make this statement without any error estimates or uncertainty information. Further, the duration estimates of CWD and CDD add up to longer than a year - this is particularly obvious in comparison with Figure 12. Please check.

**Response:**

Thank you for the comments. As mentioned above, some examination on the uncertainty in CDFs has been added by comparing the model results with that from the APHRODITE (for the historical period) and comparing the results among different models (for the future).

The phenomenon of the combined CDD and CWD exceeding 365 days in the same region arises because the high values of CWD and CDD may occur at different grid points, resulting in the possibility of the total exceeding 365 days.

L402 Should this refer to Figure 9? This statement would be more meaningful with uncertainty envelopes to clearly demonstrate whether there are or are not differences between each model.

**Response:**

The statement refers to Figure 8. Also, as mentioned previously, since the curves are closely overlapped, we couldn't find an appropriate way to include the results of model members in the figures. Instead, we have added some statistical metrics of the model members (as a new table2, lines 246-247) in our revised manuscript. Meanwhile, relative changes (compared to the observations) for different intervals have been calculated, as listed in table2(lines 248-255). In this way, we think, at least to some extent, the uncertainty across model members can be indicated.

L413 Remove, and would have the ….

Removed.

L458 It is not noteworthy that these yield similar results - that's the objective of the cooling. Remove this sentence.

The sentence has been removed.

**Response:**

The last sentence of the paragraph has been removed.

L475-481 There are other aspects related to drought risk - not least evaporation - that don't show up in the dry day count, such a strong statement about changes in drought risk aren't appropriate. See (Cheng et al., 2019; Dagon and Schrag, 2019, 2016) for more results related to climate intervention.

**Response:**

The sentence has been changed to "This reflects a potential decrease in drought risk in northwest

440     regions and an increase in extreme drought events in low-latitude southeast coastal areas in the future according to four G6 simulations. Changes in precipitation affect soil moisture, thereby influencing evapotranspiration (ET) and ultimately precipitation patterns. Assessing whether changes in DD and CDD affect drought risk also requires consideration of variations in ET and soil moisture (Cheng et al., 2019; Dagon and Schrag, 2016). Furthermore, solar radiation management (SRM) increases drought risk

445     compared to SSP5-8.5 and SSP2-4.5 scenarios in northern regions (NEC, NC, and NWC)." Please see lines 535-540 of the revised manuscript.

G6solar and G6sulfur are different ways of simulating a possible climate, we have no way of knowing whether one or the other is more valid without observations, and so can't be described as outperforming

each other. See (Bednarz et al., 2022; Visioni et al., 2021) for more discussion on this.

**Response:**

We agree that we cannot tell which one is better between G6solar and G6sulfur, and removed the related sentences (lines 329, 520-522, 545, and 582). What we did in our study is comparing both the results of G6sulfur and G6solar in extreme precipitation events against the lower emission target (SSP2-

455     4.5) or that in control period. We have added more discussion on this according to the references you suggested. Please see lines 609-610.

Section 4 I suggest condensing the bullet points to short concluding statements, which can then be followed by the explanations. Given that the focus of this article is on the climate intervention, it would also make more sense to emphasise those results rather than the future projections that have been published elsewhere. This is also the point to discuss how valid the results are with respect to other research - including experiments outside the GeoMIP project.

**Response:**

According to your thoughtful and valuable suggestions, we have revised our manuscript. The concluding statements have been shortened, and more emphasises have been placed on results on the climate intervention, instead of published future projections. For relevant revisions, please see lines 547-559 in the revised manuscript.

**Response to Referee #2**

**General comments:**

This study uses the UK Earth System Model (UKESM1) simulation results to examine the effect of solar radiation modification (SRM) geoengineering on precipitation extremes in China. As part of the GeoMIP project, UKESM1 was used to conduct two sets of SRM simulations: stratospheric aerosol injection (G6sulfur) and solar constant reduction (G6solar). Both G6sulfur and G6solar simulations are designed in such a way that global mean surface temperature under the scenario of SSP5-8.5 was brought down to the level under SSP2-4.5. Using a set of precipitation extreme indices, the authors investigated the effect of G6sulfur and G6solar on precipitation extremes for different regions of China. The authors found that compared to SSP5-8.5, both G6sulfur and G6solar ameliorate precipitation extremes over different parts of China, but increase drought risks in some northern part of China. The authors also compared the similarities and differences between precipitation extreme response to G6sulfur and G6solar for different

regions of China. The analysis of this paper itself is largely sound, but I do not recommend its publication in ACP in the present form for the following reasons:

485

I see little science in this study. I have to say that I have not carefully examined the Results part, which is just the description of figures with little scientific insight. What the authors did is just to compare simulated precipitation extremes over different regions of China under SS5-8.5, SSP2-4.5, G6solar, and G6sulfur. Regional climate extremes are strongly dependent on the SRM scenarios (location, timing, and

490 intensity of SAI and solar reduction). Also, regional climate extremes are strongly dependent on climate models. If one uses another climate model and/or another SAI strategy, most results presented in this paper might be different. At least, the authors should use multiple model results from GeoMIP instead of just one model. Also, the authors should try to investigate some science underlying the presented precipitation extreme comparisons. For example, why the difference between G6sulfur and G6solar? In

495 the present form, this paper just presents simulation results from a specific model with little interpretations. At least for me, I see little science here.

**Response:**

We greatly appreciate your time and effort in reviewing our work and providing detailed and

500 insightful comments on the manuscript, which are very helpful for improving the quality and clarity of our paper.

Indeed, in our present study, we mainly dedicated to compare the simulated precipitation extremes over different regions of China under different scenarios (SS5-8.5, SSP2-4.5, G6solar, and G6sulfur), with special focus on potential impacts of the SAI on precipitation extremes. We admit that there lack of

505 sufficient interpretations on the results, particularly in terms of mechanism linking the response to the impacts. This is partially because of our limited knowledge in relevant fields. However, considering relatively scarce research and limited knowledge on the impact of SAI over East Asia, we think our

findings are useful for deepening some understanding of the potential mitigation strategies of climate change. Thanks to your valuable comments, more interpretations on the effects of G6sulfur (e.g., in lines 423-426) and relevant mechanistic analyses (e.g., in lines 466-468, 537-538) have been added in our revised manuscript. We hope this could make up the deficiency to some extent.

We understand your concern regarding the use of only one climate model. We agree that the one model may not capture the diversity among different models, and employing multiple models for analysis could enhance the robustness of the findings and provide a more comprehensive perspective. Given this limitation in our study, we have realized that the paper title was rather over-reached in terms of what was presented in the paper and has been changed to "Projected future changes in extreme precipitation over China under stratospheric aerosol intervention in the UKESM1 climate model".

We chose the UKESM1 model due to its extensive validation in prior studies and its reputation as a reliable tool for simulating climate dynamics. UKESM1, as described by Sellar et al. (2019), represents a significant advancement over its predecessor, HadGEM2-ES, with enhanced complexity in its components and internal coupling. The model performs admirably, maintaining a stable pre-industrial state and demonstrating strong agreement with observations across various contexts (Sellar et al., 2019). Furthermore, we conducted a validation of precipitation against APHRODITE data, which demonstrated that the model has a very credible performance. Previous studies also utilized the standalone UKESM1 model to evaluate the physical and biogeochemical state of the global ocean component (Yool et al., 2020), to assess the impact of both SAI and MCB on standard meteorological variables (Haywood et al., 2023), and to research other meteorological related (Haywood et al., 2022; Wells et al., 2023; Jones et al., 2020; Jones et al., 2022; Visioni et al., 2021).

In our study, while some biases were observed, UKESM1 reasonably captures precipitation patterns, particularly in eastern China, when compared with APHRODITE data from 1981-2010. Additionally, as noted in previous research (Liang and Haywood, 2023), UKESM1 is currently the sole model capable of providing outputs of pressure-level winds and specific humidity data every 6 hours, satisfying the

requirements of the ARDT (Atmospheric Radiation Detection and Tracking) method. Furthermore, Tian et al.(2021) validated the UKESM1-0-LL simulation, demonstrating robust agreement between simulated and observed precipitation in China from 1961 to 2014, surpassing that of the CMIP6 multi-model ensemble (MME). Although acknowledging that a single model may not fully encompass the complexity of all climate variations, we believe that UKESM1 offers a valuable initial assessment of the potential impacts of SRM strategies in different regions of China.

Certainly, future research could benefit from incorporating a broader range of models to validate our findings and further explore inter-model differences. This would contribute to a more comprehensive understanding of the effects of SRM on precipitation extremes and yield more robust conclusions.

About the difference between G6sulfur and G6solar, G6solar serves as a parallel experiment to G6sulfur, aiming to compare the effects of solar reduction with those of stratospheric aerosols. G6solar adopts the same setup as G6sulfur, but geoengineering is achieved through solar irradiance reduction. Specifically, the inter-model differences in the spatial distribution of forcing are expected to be smaller in G6solar than in G6sulfur, offering valuable insights into the effects of uncertainties in stratospheric sulfate aerosol transport (Kravitz et al., 2015). These have been explained in the revised manuscript (lines 152-159).

Additionally, we would like to note that, when assessing impacts, it is common to focus on the most relevant metrics that are influenced. For example, the recent paper by Mari Tye (https://esd.copernicus.org/articles/13/1233/2022/):

*Tye, M. R., Dagon, K., Molina, M. J., Richter, J. H., Visioni, D., Kravitz, B., and Tilmes, S.: Indices of extremes: geographic patterns of change in extremes and associated vegetation impacts under climate intervention, Earth Syst. Dynam., 13, 1233–1257, https://doi.org/10.5194/esd-13-1233-2022, 2022.*

In Tye et al. (2022), a single model (Community Earth System Model, CESM1) is used for assessing extremes in temperature, precipitation and vegetation. While it should be more meaningful to delve into the causal mechanisms linking the response to the impacts, simply documenting impacts in extremes (as

in Tye et al. (2022)) also appears worthy, at least to some. Other examples of published work that use a single model and focus on the impacts include, but are not limited to:

560    *Muthyala, R., Bala, G., & Nalam, A. (2018). Regional scale analysis of climate extremes in an SRM geoengineering simulation, Part 2: temperature extremes. Current Science, 1036-1045.*

*Tilmes, S., Sanderson, B. M., & O'Neill, B. C. (2016). Climate impacts of geoengineering in a delayed mitigation scenario. Geophysical Research Letters, 43(15), 8222-8229.*

565    *Jones, A.C., Hawcroft, M.K., Haywood, J.M., Jones, A., Guo, X. and Moore, J.C., 2018. Regional climate impacts of stabilizing global warming at 1.5 K using solar geoengineering. Earth's Future, 6(2), pp.230-251.*

**Specific comments:**

570

Lines 36-59: This first paragraph of the Introduction part is very lengthy and most part is not directly relevant to the study here. For example, the detailed description of extreme precipitation in Zhengzhou and Beijing is not needed at all.

**Response:**

575    Thank you very much for pointing out the redundancy of the introduction part. We have made the first paragraph more concise and pertinent to the study, including shortening the description of extreme precipitation in Zhengzhou and Beijing. Please see lines 40-66 in the revised manuscript.

Lines 61-72: This paragraph can also be substantially shortened and combined with the first paragraph.

580    **Response:**

According to your suggestion, we have substantially shortened the second paragraph, and combined it with the first paragraph. Please see lines 66-79.

Line 85: check the grammar here. ',the climate'

585    **Response:**

It should be "on the climate". However, this sentence has been removed for brevity.

Lines 90-101: The use of 'prediction' in this paragraph is not appropriate.

**Response:**

590    We have replaced "prediction" with "projection" in the paragraph.

Lines 98-99: Whether SAI would decrease precipitation depends on the scenario of SAI deployment. Also, instead of Pinto et al. 2020 and Liu and et al. 2021, more influential papers on the climate effect of SAI should be cited.

595    **Response:**

Yes, we agree that different scenario of SAI deployment would have different effects on the climate, in particular, the spatial and temporal distribution of precipitation. In light of this consideration, the statement "SAI will exert a negative radiative forcing and reduce near-surface air temperature (including temperature means and extremes) (Pinto et al., 2020), and precipitation (Liu et al., 2021)" has been revised

600    as: " However, the climate effects in terms of magnitude as well as spatial and temporal distribution depend largely on the scenario of SAI deployment. Furthermore, as suggested by some studies, although SAI can effectively counteract anthropogenic global warming at the global scale, it cannot fully offset the effects at regional scale (Tilmes et al., 2013; Niemeier et al., 2013; Tye et al,. 2022)". Please see lines 113-116 of the revised manuscript.

605

Line 120: Why only use results from a single model? Why not use multi-model results from GeoMIP?

**Response:**

Thank you for pointing out this issue. In the response to your comment at the beginning, we have answered this point. Please refer to our explanation there.

Lines 186-187: I don't quite understand this sentence.

**Response:**

We are sorry for the confusion due to our carelessness. Since we have already employed the Wilcoxon test for the significance testing, the field significance analysis mentioned here are not necessary, and in fact, it was not used in this study. Therefore, the description (line185-194 in original manuscript) has been deleted.

Line 199: The word of 'accurate' is not appropriate here.

**Response:**

The "accurate" was replaced with "similar to the observed precipitation" (line 224).

Line 133: 'reducing the solar constant or increasing SAI'. Check grammar and spelling here.

**Response:**

The sentence has been removed for brevity.

Lines 225-226: I don't understand what 'SAI is sensitive to global warming' means.

**Response:**

The sentence has been revised as "This suggests that the effect of SAI on future precipitation is more widespread and remarkable compared to that of SSP5-8.5." Please see lines 273-274.

Line 245: Where are 'the other three G6 models'?

**Response:**

Sorry for our carelessness. The "models" should be "scenarios", and it has been corrected in the revised manuscript (line 294). By the way, the other three G6 simulations refer to that of SSP2-4.5, G6sulfur, and G6solar.

**Response to Referee #3**

**General comments:**

I think this could be a useful study with some work. I always like creative ways of integrating measurements and models. The analysis is also carefully done and focuses on clearly important issues (extremes).

While I don't dispute any of the findings, my biggest issue is with the explanations. There is a lot of
645 reporting of the results but not much interpretation other than (sometimes) speculating about mechanisms. Given that the authors have a great deal of climate model output at their disposal, they could look into some of these mechanisms. I would point out specific examples, but this seems to be a general issue in Section 3.

Also, there is a lot of discussion of different indices, but they mostly show the same thing. That's
650 not a problem, but the way you're describing them makes it seem like you're going through a laundry list of indices. I'd like to see more insight. Digging into the results in Table 2 would be interesting. For example, _why_ does CWD not behave like the other indices? What's special about those two regions that have the opposite sign?

**Response:**

655 First of all, we'd like to note that we have realized that the title of the original manuscript was rather over-reached in terms of what is presented in the paper and changed it to "Projected future changes in extreme precipitation over China under stratospheric aerosol intervention in the UKESM1 climate model", which is more relevant to the results of the article.

We have to admit that interpretation of the results and speculation about the mechanisms are
660 insufficient, partially because of our limited knowledge in relevant areas. Though it can't be fully made up for the time being, we are trying to make improvements by adding some more interpretations and discussions in our revised manuscript (e.g., in lines 423-426, 466-468, 537-538). Nevertheless, the main

purpose is to demonstrate the possible impacts of SAI by comparations among different scenarios. Really, it would be more interesting to discuss on causal mechanisms linking the response to the impacts, considering the relatively scarce research and limited knowledge on the impact of SAI over East Asia, we think simply documenting the impacts in extremes (as in Tye et al. (2022)) also appears worthy, at least to some.

Thank you for your suggestion for digging into the results in Table 3 (the Table 2 has been changed to Table 3 in the revised manuscript). We have added following discussions/statements in the revised manuscript:

At lines 441-443: "In the regions projected to experience an increase of CWD in NE and NWC, the positive value (in Table3) indicates that SAI experiments produce results of threshold that are closer to the CP conditions. However, the relative effect is not obvious due to the small magnitude of CWD in these regions."

At Lines 509-510: "As shown in Table 3, the DD is positive in the SC region, meaning G6sulfur effectively lowers the threshold for extreme DD events compared to SSP5-8.5. This suggests that the SAI is more effective for DD maximum in the humid region."

At Lines 522-529: "The positive value in Table 3 of the CDD index in the SC and SWC regions in Table 3 indicates that G6sulfur notably closes the threshold of CP extreme CDD events compared to SSP5-8.5, thereby approaching drought extremes of CP in these regions. This suggests that G6sulfur has the potential to mitigate the CDD extremes. The ameliorating effect of DD and CDD compared to SSP5-8.5 in the SC region under G6sulfur may be related to the strengthening of the anti-cyclonic circulation associated with the subtropical gyre, which appears to increase under G6 compared to SSP5-8.5 (Liang and Haywood, 2023). This intensification results in an increased inflow of moist air from the ocean at 850hPa and a greater supply of moisture to the southern region of the area."

**Specific comments:**

Figure 1 and Section 2.1: Any reason you don't include the Tibetan plateau?

690 **Response:**

As shown in Figure1, the Tibetan plateau is included and represented as the brown areas, which is divided into two parts in SWC and NWC. The division of the regions follows a conventional approach that has been widely adopted in many statistical reports and relevant studies (e.g., Luo et al., 2017; Fan et al., 2020; Yang and Shao, 2021; Liang et al., 2023).

695

Lines 169-182: This seems like a long way of saying that you used survival functions, which are a perfectly reasonable thing to use for what you want to do.

**Response:**

We have revised it more concise. Please see lines 194-207.

700

Lines 186-187: This is not consistent with my understanding of what field significance does. I would appreciate more description as to what you mean.

**Response:**

We are sorry for the confusion due to our carelessness. Since we have already employed the
705 Wilcoxon test for the significance testing, the field significance analysis mentioned here are not necessary, and in fact, it was not used in this study. Therefore, the part has been deleted.

Figure 2: Can you add a panel showing the bias as a percent instead of an absolute value?

**Response:**

710 Thank you for your valuable suggestion. Really, it would be meaningful to show the bias in terms of relative changes, in addition to the absolute value. When trying to do so (adding a panel showing the bias as percentages), a problem we met is that, since a great deal of the results (daily precipitation) are small values, the relative changes (percentages) could be very large even for minor absolute changes (particularly for those in western and northern areas), and this could make the results confused.

For this reason, a panel of scatter plot comparing the observations with the simulations (mean of the ensemble model) is added as Fig.2d in the revised manuscript (line 237), also shown below. In addition, comparisons between the observations and each of the three model members are also provided as supplementary material (Figure S1). The observations (daily precipitation) during the control period in China were classified in to several intervals: P10 (the smallest 10%), P10-50, P50-90, P90-95, and P95 (the largest 5%). In order to indicate the bias as a percent, relative changes (compared to the observations) for different intervals have been calculated, and listed in a new table2 added in our revised manuscript (lines 246-247), which is also provided below.

The scatter plots indicate a close relationship between the observations and the model results. However, the model results are generally higher than observations, possibly because of the different resolution of the data. Since our study has been mainly focused on the relative changes between the future results and that of control period for different scenarios, the systematic bias would not affect the conclusions significantly. As expected, relative changes are very large at small values (below the 10th percentile), both for the ensemble mean and the model members. For the results at the 10-50th and 50-90th percentiles, relative changes are around 30%. When larger than the 95th percentile, relative changes are relatively small, near 15%. The differences among ensemble members are not significant, which suggests the uncertainty in the ensembled results is reasonable and acceptable. What mentioned above have been included in our revised manuscript. Please see lines 248-255.

[Figure]

(Scatter plots)

**Figure Scatter plots between the observations and model results at different level of precipitation during the control period (CP). The first panel is provided as Fig.2d in the revised manuscript, and the other three panels are provided as Fig. S1 in the supplementary material.** The observations were classified into several intervals: P10 (the smallest 10%), P10-50, P50-90, P90-95, and P95 (the largest 5%).

Table2: Relative changes of the model results (compared to the observations)

| intervals | Ensemble mean | r1i1p1f2 | r4 i1p1f2 | r8i1p1f2 |
| --- | --- | --- | --- | --- |
| <P10 | 89.81% | 93.95% | 89.44% | 86.04% |
| P10-50 | 30.05% | 30.38% | 31.85% | 27.13% |
| P50-90 | 30.50% | 28.95% | 31.36% | 31.16% |
| P90-95 | 24.03% | 22.79% | 24.85% | 24.44% |
| >P95 | 15.76% | 15.09% | 16.27% | 15.92% |

Lines 225-226: I'm not sure what this means. SAI is sensitive to global warming?

**Response:**

We are sorry for the mistake. The sentence has been revised as "This suggests that the effect of SAI on future precipitation is more widespread and remarkable, compared to that of SSP5-8.5". Please see lines 273-274.

Line 265:   Typo (depicting)

**Response:**

We have corrected 'dicipting' to 'depicting. ' Line 314.

Line 293:   Aggregated how?

**Response:**

Sorry for the wording mistake. The "aggregated" should be "presented". The sentence has been revised, please see lines 340-349.

Line 295: Be more specific about "the opposite". Also, what are 0 values in the table?   (I can figure it out, but you need a description.)

**Response:**

To be more specific about "the opposite" and "0 values", an explanation is provided: "A positive difference suggests a mitigation effect of SAI, while a negative difference indicates exacerbation in index thresholds for projected increase regions. In regions where the projected index is decreasing, the meaning of positive and negative signs is opposite to that in regions where the index is projected to increase. In addition, the 0 values indicate there is almost no difference between the maximum index values under G6sulfur and SSP5-8.5, suggesting negligible impact of SAI on indices threshold." Please see lines 345–348 of the revised manuscript.

 It's difficult to put these numbers in context.    Is 100 mm a lot for these regions?

**Response:**

770     The sentence has been revised as: "in EC and CC, there are decreases of more than 100mm, whereas in NEC and NC, there are some increases of about 50mm." Please see lines 374-376.

   I don't know if "effectively mitigates" is the correct phrasing.    Be more specific.

**Response:**

775     The sentence "SRM effectively mitigates the increase of RX1day, RX5day and R95p compared to SSP5-8.5 scenario in all regions" has been changed to "SRM results are encouraging, showing a reduction in the detrimental extreme events, similar to the lower emissions target of SSP2-4.5". Please see lines 392-393.

780        I'll be honest, I had a hard time with this entire paragraph.    I'm really not sure I understand it.

**Response:**

        Thank you for your feedback on this paragraph. We have revised it as: "In the regions projected to experience an increase of CWD in NE and NWC, the positive value (in Table3) indicates that SAI
785     experiments produce results of threshold that are closer to the CP conditions. However, the relative effect is not obvious due to the small magnitude of CWD in these regions. It is notable that in NC, and SC, G6sulfur (black) provides similar results to the SSP2-4.5. Interestingly, for EC, SSP2-4.5 yields almost identical statistics to SSP5-8.5, while both G6sulfur and G6solar show an increase compared to SSP scenarios." Please see lines 441-451.

790

   ETCCDI

**Response:**

Thank you for pointing out the mistake, it should be "ETCCDI". However, this sentence has been removed for brevity.

795

**References**

Donat, M., Alexander, L. V., Yang, H., Durre, I., Vose, R., Dunn, R. J., Willett, K. M., Aguilar, E., Brunet, M., and Caesar, J.: Updated analyses of temperature and precipitation extreme indices since the beginning of the twentieth century: The HadEX2 dataset, Journal of Geophysical Research: Atmospheres, 118, 2098-2118, https://doi.org/10.1002/jgrd.50150, 2013.

800

Donat, M. G., Lowry, A. L., Alexander, L. V., O'Gorman, P. A., and Maher, N.: More extreme precipitation in the world's dry and wet regions, Nature Climate Change, 6, 508-513, https://doi.org/10.1038/nclimate2941, 2016.

Fan, H., Zhao, C., and Yang, Y.: A Comprehensive Analysis of the Spatio-Temporal Variation of Urban Air Pollution in China During 2014–2018, Atmos. Environ., 220, 117066, https://doi.org/10.1016/j.atmosenv.2019.117066, 2020.

805 Haywood, J. M., Jones, A., Johnson, B. T., and McFarlane Smith, W.: Assessing the consequences of including aerosol absorption in potential stratospheric aerosol injection climate intervention strategies, Atmos. Chem. Phys, 22, 6135-6150, https://doi.org/10.5194/acp-22-6135-2022, 2022.

Haywood, J. M., Jones, A., Jones, A. C., Halloran, P., and Rasch, P. J.: Climate intervention using marine cloud brightening (MCB) compared with stratospheric aerosol injection (SAI) in the UKESM1 climate model, Atmos. Chem. Phys, 23, 15305-

810 15324, https://doi.org/10.5194/acp-23-15305-2023, 2023.

Held, I. M. and Soden, B. J.: Robust responses of the hydrological cycle to global warming, J. Climate., 19, 5686-5699, https://doi.org/10.1175/JCLI3990.1, 2006.

Irvine, P. J., Kravitz, B., Lawrence, M. G., and Muri, H.: An overview of the Earth system science of solar geoengineering, Wiley Interdisciplinary Reviews: Climate Change, 7, 815-833, https://doi.org/10.1002/wcc.423, 2016.

815 Jones, A. C., Hawcroft, M. K., Haywood, J. M., Jones, A., Guo, X., and Moore, J. C.: Regional climate impacts of stabilizing global warming at 1.5 K using solar geoengineering, Earth's Future, 6, 230-251, https://doi.org/10.1002/2017EF000720, 2018.

Jones, A., Haywood, J. M., Jones, A. C., Tilmes, S., Kravitz, B., and Robock, A.: North Atlantic Oscillation response in GeoMIP experiments G6solar and G6sulfur: why detailed modelling is needed for understanding regional implications of solar radiation management, Atmos. Chem. Phys., 21, 1287-1304,https://doi.org/10.5194/acp-21-1287-2021, 2020.

820 Jones, A., Haywood, J. M., Scaife, A. A., Boucher, O., Henry, M., Kravitz, B., Lurton, T., Nabat, P., Niemeier, U., and Séférian, R.: The impact of stratospheric aerosol intervention on the North Atlantic and quasi-biennial oscillations in the geoengineering model intercomparison project (GeoMIP) G6sulfur experiment, Atmos. Chem. Phys. 22, 2999-3016, https://doi.org/10.5194/acp-22-2999-2022, 2022.

Kravitz, B., Robock, A., Tilmes, S., Boucher, O., English, J. M., Irvine, P. J., Jones, A., Lawrence, M. G., MacCracken, M.,

825 and Muri, H.: The geoengineering model intercomparison project phase 6 (GeoMIP6): Simulation design and preliminary results, Geosci. Model Dev., 8, 3379-3392, https://doi.org/10.5194/gmd-8-3379-2015, 2015.

Liang, J. and Haywood, J.: Future changes in atmospheric rivers over East Asia under stratospheric aerosol intervention, Atmos. Chem. Phys., 23, 1687–1703, https://doi.org/10.5194/acp-23-1687-2023, 2023.

Liang, J., Meng, C., Wang, J., Pan, X., and Pan, Z.: Projections of mean and extreme precipitation over China and their resolution dependence in the HighResMIP experiments, Atmospheric Research, 293, 106932, https://doi.org/10.1016/ j. atmosres.2023.106932, 2023.

Luo, J., Du, P., Samat, A., Xia, J., Che, M., Xue, Z.: Spatiotemporal Pattern of PM2.5 Concentrations in Mainland China and Analysis of Its Influencing Factors using Geographically Weighted Regression. Sci. Rep., 7, 40607. https://doi.org/10.1038/srep40607, 2017.

Muthyala, R., Bala, G., and Nalam, A.: Regional scale analysis of climate extremes in an SRM geoengineering simulation, Part 2: temperature extremes, Current Science, 1036-1045, http://www.jstor.org/stable/26495197, 2018.

Niemeier, U., Schmidt, H., Alterskjær, K., and Kristjánsson, J.: Solar irradiance reduction via climate engineering:Impact of different techniques on the energy balance and the hydrological cycle, J. Geophys. Res.-Atmos., 118, 11,905-911,917, https://doi.org/10.1002/2013JD020445, 2013.

Niu, S., Sun, M., Wang, G., Wang, W., Yao, X., and Zhang, C.: Glacier Change and Its Influencing Factors in the Northern Part of the Kunlun Mountains, Remote Sensing, 15, 3986, https://doi.org/10.3390/rs15163986, 2023.

Pendergrass, A. G. and Knutti, R.: The uneven nature of daily precipitation and its change, Geophysical Research Letters, 45, 11,980-911,988, https://doi.org/10.1029/2018GL080298, 2018.

Peng, Y., Zhao, X., Wu, D., Tang, B., Xu, P., Du, X., and Wang, H.: Spatiotemporal variability in extreme precipitation in China from observations and projections, Water, 10, 1089, https://doi.org/10.3390/w10081089, 2018.

Qin, P. and Xie, Z.: Detecting changes in future precipitation extremes over eight river basins in China using RegCM4 downscaling, Journal of Geophysical Research: Atmospheres, 121, 6802-6821, https://doi.org/10.1002/2016JD024776, 2016.

Ricke, K., Wan, J. S., Saenger, M., and Lutsko, N. J.: Hydrological consequences of solar geoengineering, Annual review of earth and planetary sciences, 51, 447-470, https://doi.org/10.1146/annurev-earth-031920-083456, 2023.

Sellar, A. A., Jones, C. G., Mulcahy, J. P., Tang, Y., Yool, A., Wiltshire, A., O'connor, F. M., Stringer, M., Hill, R.,and Palmieri, J.: UKESM1: Description and evaluation of the UK Earth System Model, J. Adv. Model. Earth Sy., 11, 4513-4558, https://doi.org/10.1029/2019MS001739, 2019.

Simpson, I., Tilmes, S., Richter, J., Kravitz, B., MacMartin, D., Mills, M. J., Fasullo, J., and Pendergrass, A. G.: The regional hydroclimate response to stratospheric sulfate geoengineering and the role of stratospheric heating, J. Geophys. Res.-Atmos., 124, 12587-12616, https://doi.org/10.1029/2019JD031093, 2019.

Tank, A. K. and Können, G.: Trends in indices of daily temperature and precipitation extremes in Europe, 1946–99, Journal of Climate, 16, 3665-3680, https://doi.org/10.1175/1520-0442(2003)016<3665:TIIODT>2.0.CO;2, 2003.

Tian, J., Zhang, Z., Ahmed, Z., Zhang, L., Su, B., Tao, H., and Jiang, T.: Projections of precipitation over China based on CMIP6 models, Stoch Environ Res Risk Assess., 35, 831-848, 2021.

Tilmes, S., Fasullo, J., Lamarque, J. F., Marsh, D. R., Mills, M., Alterskjær, K., Muri, H., Kristjánsson, J. E., Boucher, O., and Schulz, M.: The hydrological impact of geoengineering in the Geoengineering Model Intercomparison Project (GeoMIP), J. Geophys. Res.-Atmos., 118, 11,036-011,058, https://doi.org/10.1002/jgrd.50868, 2013.

Tilmes, S., Sanderson, B. M., and O'Neill, B. C.: Climate impacts of geoengineering in a delayed mitigation scenario, Geophys. Res. Lett., 43, 8222-8229, https://doi.org/10.1002/2016GL070122, 2016.

Tye, M. R., Dagon, K., Molina, M. J., Richter, J. H., Visioni, D., Kravitz, B., and Tilmes, S.: Indices of extremes: geographic patterns of change in extremes and associated vegetation impacts under climate intervention, Earth Syst. Dynam., 13, 1233–

1257, https://doi.org/10.5194/esd-13-1233-2022, 2022.

Visioni, D., MacMartin, D. G., Kravitz, B., Boucher, O., Jones, A., Lurton, T., Martine, M., Mills, M. J., Nabat, P., and Niemeier, U.: Identifying the sources of uncertainty in climate model simulations of solar radiation modification

870   with the G6sulfur and G6solar Geoengineering Model Intercomparison Project (GeoMIP) simulations, Atmos. Chem.Phys., 21, 10039-10063, https://doi.org/10.5194/acp-21-10039-2021, 2021.

Wells, A. F., Jones, A., Osborne, M., Damany-Pearce, L., Partridge, D. G., and Haywood, J. M.: Including ash in UKESM1 model simulations of the Raikoke volcanic eruption reveals improved agreement with observations, Atmos. Chem. Phys., 23, 3985-4007, https://doi.org/10.5194/acp-23-3985-2023, 2023.

875   Yang, J., Shao, M., 2021. Impacts of Extreme Air Pollution Meteorology on Air Quality in China. J. Geophys. Res. Atmos., 126, e2020JD033210. https://doi.org/10.1029/2020JD033210, 2021.

Yool, A., Palmiéri, J., Jones, C. G., de Mora, L., Kuhlbrodt, T., Popova, E. E., Nurser, A. G., Hirschi, J., Blaker, A. T., and Coward, A. C.: Evaluating the physical and biogeochemical state of the global ocean component of UKESM1 in CMIP6 historical simulations, Geosci Model Dev 2020, 1-68, https://doi.org/10.5194/gmd-14-3437-2021, 2020.

880   Zarnetske, P. L., Gurevitch, J., Franklin, J., Groffman, P. M., Harrison, C. S., Hellmann, J. J., Hoffman, F. M., Kothari, S., Robock, A., and Tilmes, S.: Potential ecological impacts of climate intervention by reflecting sunlight to cool Earth, Proceedings of the National Academy of Sciences, 118, e1921854118, https://doi.org/10.1073/pnas.1921854118, 2021.

---

## Author Response (AR2)

**Dear editor and reviewer,**

We greatly appreciate your thoughtful and valuable comments and suggestions in both rounds of the review, which have been crucial in improving our manuscript. Now, according to your latest feedback, this manuscript has been further revised. We have provided a marked-up version of the manuscript showing all changes made (using track changes in Word). Following is a point-by-point response to the comments.

With warm regards,

Ou Wang, on behalf of all co-authors

**Response to comments by referee #1:**

**General comment:**

The authors have worked hard to improve the manuscripts and addressed many of my previous concerns. However, some inconsistencies remain in the text and some fairly major changes are still necessary before this article can be accepted for publication.

**Response:**

Thank you very much for your comments. We have carefully considered each of the comments, and most of them have been adopted in improving our manuscript. Responses to all the comments are given point by point as follows.

I have not checked the ms beyond L270 as my comments are largely repetitive: The point of the G6 simulations is not whether they arrive at a "better" future than SSP5-8.5, a better future would be SSP1.19. Rather the point is whether using SAI to hold temperatures in line with SSP2-45 will have any other unforeseen consequences. Thus all results comparing back to SSP5-85 need to be removed, and statistical significance tests should be between SSP2-45 and G6 simulations. Please revise the remainder of the manuscript to reflect this.

**Response:**

We understand your suggestion that the G6sulfur simulations should only be compared against SSP2-4.5. This type of analysis highlights any differences that the impact of stratospheric aerosol injection (under the assumptions implicit within the G6sulfur strategy) might have when compared to a greenhouse gas warming scenario with the same global mean temperature.

While we acknowledge that this is important, and has been the focus of many SRM publications, including those using both SAI (e.g. Wells et al., 2024) and MCB (e.g. Haywood et al., 2023), this type of analysis does not directly present a 'risk-risk' analysis – i.e. the risk of unmitigated global warming (SSP5-8.5) against a world where greenhouse gas emissions are essentially unmitigated, but SAI is used to counter-balance a proportion of the global warming.

While we agree that the academic community may be interested in the residual climate impacts between a greenhouse gas world and a (GHG+SAI) world, the first-order impact of SAI is often overwhelmingly positive – at least when viewed as a physical risk-physical risk analysis. This fact is, unfortunately, often overlooked in the, largely more academic, presentations on SRM. Questions that the public and policy-makers ask are along the lines of i) Could SAI be effective in reducing the impacts of climate change? and ii) What are the residual climate impacts? Ignoring the first-order impacts (i.e. point (i)) and presenting only the second-order impacts (i.e. point (ii)) would distort the policy-relevance of the studies.

It is important to acknowledge that the G6 experiments (G6sulfur and G6solar) are based on SSP5-8.5; i.e., the external climate forcing of SAI is applied to the GHG emission scenario (Tilmes et al., 2022). Their comparisons to SSP5-85 directly quantify the role of aerosol-climate interaction in the effect of SAI on climate. It is important not only for the policy-makers but also for climate scientists who are interested in aerosol-climate feedback.

Studies such as those by Liang and Jim. (2023) have investigated the combined effects of GHG+SAI, demonstrating the simulated SAI strategy is effective at partly mitigating the projected future increase in atmospheric rivers (ARs) activity under SSP5–8.5 over the study region, while it induces significant increases (by up to 0.15 %) in the AR frequency over north-eastern China. These findings provide essential insights for shaping climate policies. They help policymakers understand how SAI can influence

precipitation patterns regionally, aiding in the formulation of evidence-based decisions for climate adaptation and mitigation strategies.

By presenting G6 against SSP5-85, we provide important information to the policy-makers. This is of particular interest and of relevance to policy-makers in China, who have not had a great deal of exposure to the science of SRM via SAI. Given the narrative and arguments laid out above, we respectfully request that the reviewer reconsider their insistence in removing this analysis; we feel it is too important to ignore.

At a general level please ensure you choose either precipitation or rainfall and stick with that terminology. Similarly for "global warming" "anthropogenic global warming" "warming at the global scale".

**Response:**

For the terms "precipitation" or "rainfall," different references use different words, but we have consistently used "precipitation" throughout the Abstract and main text and have ensured that all instances are correct. Additionally, we have checked and ensured that only "global warming" is used consistently throughout the text.

I remain unconvinced about the need to present the comparison with APHRODITE data. Your analysis centres around extreme precipitation, for which the models are not appropriate as they overestimate the maximum intensity. Furthermore, you do not (and do not need to!) bias correct the results; and I do not see where else the comparison to observations is used (not line 230-255 as stated in your response). You could instead simply state that the model has been effectively evaluated for adequacy elsewhere citing relevant papers (including your own paper Liang and Haywood 2023!). If you really wish to keep it, I suggest putting a summary of the analysis as an Appendix.

**Response:**

We agree that the validation should be moved to the Supplement. While it may not be highly relevant for these theoretical simulations, it provides reassuring that the model performs satisfactorily. We also refer to Liang and Haywood (2023) and other relevant articles. We have added the statement, please see lines 219-222.

The presentation of the results is still confusing. In all cases you present changes and statistical significance with respect to the control period, but state that SAI is approximately successful. I recommend breaking this up to illustrate both the changes with respect to the control period, and the differences at the end of the century from SSP2-45. Discuss the results with respect to these differences from SSP2-4.5 as that is the target. As the pattern of changes is largely the same for all indices, you don't need to show the results for the changes relative to control, but could explain that the patterns are similar to those seen in other indices and cite other literature for UKESM/CMIP6 and focus on the effects of SAI.

**Response:**

We understand your suggestion to break up the analysis to illustrate both the changes with respect to the control period (CP) and the differences at the end of the century from SSP2-4.5, as well as not to show the results for the changes relative to the CP but to explain that the patterns are similar to those seen in other indices. While we recognize the importance of this approach, we believe that presenting changes relative to the CP allows for a more straightforward visualization of future changes under different scenarios, providing clearer insights into the effectiveness of SAI in mitigating climate effects.

Additionally, we have included comparisons between G6solar and SSP2-4.5 for all indices and discussed the differences in results among various simulations. We have also added, for example, details on the mechanisms and reasons for SAI's effects relative to SSP2-4.5 in lines 254-256, as this information is crucial for understanding its effect.

We believe that this comprehensive approach provides a balanced view of the effect, addressing both the need for direct comparisons with SSP2-4.5 and the benefits of understanding changes relative to the control period. This dual perspective ensures that the policy relevance and scientific robustness of our findings are maintained.

**Specific Comments:**

L24 G6sulfur is.

**Response:**

Thank you. Already added the space "G6sulfur" and "is".

**Response:**

Thank you. Deleted.

L28-29 As stated for the previous version of the ms, this is not appropriate and should be reframed as matching the target FOR THIS MODEL only similar to L31

**Response:**

The sentence has been revised to "The G6sulfur and G6solar experiments show statistically similar results to those under SSP2-4.5 in extreme precipitation intensities of China in UKESM1. These results are encouraging." Please see lines 28-30.

**Response:**

Thank you. Deleted, as this sentence "the G6sulfur and G6solar show encouraging potential abatement of the impacts from detrimental extreme events which are similar with the lower emissions target of SSP2-4.5, there are some exceptions. For instance," is similar to before "The G6sulfur and G6solar experiments show statistically similar results to those under SSP2-4.5 in extreme precipitation intensities of China in UKESM1.".

L34 Should really highlight that although only one model, small ensemble etc., the hydrological effects are not as beneficial as those indicated for temperature and more research recommended. Leave statements about deployment to policy people.

**Response:**

Thank you for this important comment. The sentence has been changed to: "Given the limitations of the current model and the small ensemble size, and considering that the hydrological effects are less beneficial than those indicated for temperature, it is recommended that further, more comprehensive

research be performed, including using multiple models, to better understand these impacts." Please see line 35-38.

Inserted.

L43-52 You could still reduce the level of historical events as the focus of the article is not on these impacts. e.g. start a new paragraph at "Extreme precipitation…" then "For example severe flooding affected southern, eastern and parts of central China in the summer of 2020 (Jia et al 2020); extremely intense hourly and daily precipitation also occurred over Zhengzhou (southeast? China) in 2021 (Zhao et al, Dong et al). Further extreme precipitation occurred over southern China in 2022, together with flooding in the west of the Yangtze; and finally record-breaking precipitation occurred in 2023 across at least 15 cities and provinces in north-eastern China" Could cite the WMO State of the Climate report here to back up the record-breaker. WMO (2023) Provisional State of the Climate in 2023. World Meteorological Organization. Available at: https://www.nature.com/articles/nclimate3063 (Accessed: 15 February 2024).

**Response:**

Thank you for the thoughtful suggestion. The sentence has been revised to "Extreme precipitation events appear to have impacted China more frequently in recent years. For example, severe flooding affected southern, eastern, and parts of central China in the summer of 2020 (Jia et al., 2022); extremely intense hourly and daily precipitation also occurred over Zhengzhou (central China) in 2021 (Zhao et al., 2021; Dong et al., 2022). Typhoon Doksuri in 2023 resulted in significant flooding in China, setting records near Beijing, while Yunnan province experienced its most severe drought since 1961 (WMO, 2024). These events suggest a potential expansion of regions that could be influenced by increasing precipitation under the changing climate. On a global scale, climate change has been influencing hydroclimatic conditions (Donat et al., 2016; Pendergrass and Knutti, 2018)." Please see lines 45-58.

L50 Remove "Although not statistically robust" and "might tentatively" Dunn et al. (2020) and de Vries et al. (2023) show a robust signal in Rx1day globally.

De Vries, I.E. et al. (2023) 'Robust global detection of forced changes in mean and extreme precipitation despite observational disagreement on the magnitude of change', Earth System Dynamics, 14(1), pp. 81–100. Available at: https://doi.org/10.5194/esd-14-81-2023.

Dunn, R.J.H. et al. (2020) 'Development of an Updated Global Land In Situ-Based Data Set of Temperature and Precipitation Extremes: HadEX3', Journal of Geophysical Research: Atmospheres, 125(16). Available at: https://doi.org/10.1029/2019JD032263.

**Response:**

Removed. We thank the reviewer for recommending these helpful references.

L61 Change 'forecasted' to 'projected'.

**Response:**

Thank you. Changed.

L63 Again this is a policy statement best left to policy makers. The projections show an urgent need to mitigate (I.e. reduce carbon emissions) to avoid worse changes, and to adapt. However, as you go on, some have suggested that climate interventions may also support those actions to further abate the impacts from climate change.

**Response:**

Thank you for pointing out this problem. The sentence has been revised to "An increase in precipitation forecasted by current climate models, particularly that projected over the populated areas in East Asia, such as China (Liang and Haywood, 2023), indicates an urgent need for mitigation efforts (i.e., reducing carbon emissions) to prevent worsening impacts from climate change. However, it has been suggested that climate interventions could complement these actions in further mitigating the impacts of climate change." Please see lines 67-71.

L65 Technically this should be a reference to COP 15, UNFCCC rather than the IPCC special report.

195 **Response:**

Thank you for pointing out this mistake, the reference has been corrected. Please see lines 73-75.

L73-75 Move "Numerous studies…." To L69 before G6Sulfur - to give better background on the idea of a "natural analogue" in the form of volcanoes.

200 **Response:**

The sentence has been moved. Please see lines 79-81.

L71 Zarnetske also points out the negative consequences; it is worth making more of that to make this paper a balanced contribution.

205 **Response:**

Thank you for your advice. We have added "Thus, SAI would come with some adverse consequences, including stratospheric polar ozone depletion leading to increased surface UV radiation, and increased sulphate deposition to the surface (acid rain). Moreover, the potential risks of abrupt termination also reveal significant changes in temperature and precipitation velocities, with potential severe impacts on

210 ecological systems (e.g. Trisos et al., 2017). Despite the numerous drawbacks (e.g. Robock et al., 2015), its potential climate regulation effects make it arguably a plausible strategy to address escalating climate change challenges." Please see lines 87-92.

L75 New paragraph for "The latest phase of …".

215 **Response:**

This section has been moved to start a new paragraph at line 93.

L79 Description of model specifics (I.e. SSP) belongs in the Methods section.

**Response:**

220 Lines 79-83 has been removed to Method section. Please see lines 133-136.

L83 rephrase "Previous studies from a range of modeling experiments innate that SAI will exert….

**Response:**

The sentence has been rephrased in line 97.

225

L84 This is too generalised. Instead "reduce mean surface air temperature and may reduce global mean precipitation".

**Response:**

Thank you for your suggestion. The sentence has been revised. Please see lines 98-99.

230

L86 change to "when stratospheric sulfur is used to moderate global mean temperatures".

**Response:**

The sentence has been revised in lines 100-101.

235 L87 Remove "However, as suggested by some studies," and change to "can effectively moderate global mean temperature increases, it cannot…".

**Response:**

Thank you for the comments. The sentence has been changed to "Although SAI can effectively moderate global mean temperature increases, it cannot fully offset the effects at regional scale (Niemeier

240 et al., 2013; Tilmes et al., 2013; Tye et al., 2022)." in lines 102-104.

L92 geoengineering is redundant here.

**Response:**

The word "geoengineering" has been removed.

245

L93 maybe differences instead of changes? And "between scenarios of projected warming alone, and warming with solar geoengineering"

**Response:**

The sentence has been revised to "our study explores the differences in frequency and intensity of
extreme precipitation between the scenario of projected warming alone and warming with solar
geoengineering (G6sulfur, G6solar)." Please see lines 109-111.

L98 This should be 2100?

**Response:**

The number has been corrected in line 112.

L117 remove "Tier 1" this is confusing.

**Response:**

Removed.

L125 Define the control period here instead of calling it "Historical".

**Response:**

We've defined the control period as suggestion. Please see line 146.

L131 maybe "was created from spatial interpolation of gauge…"

**Response:**

The sentence has been revised in line 153.

L132 change "has been a" to "is a"

**Response:**

Changed. Please see line 155.

L140 include the references that are also in the Table caption here.

**Response:**

The references have been included on line 162.

L162-190 This section does not describe changes and so doesn't go with the title. It would be better removed altogether as stated above.

**Response:**

280    This sentence has been removed to the Supplement. As mentioned in the previous response, we have added the statement in lines 219-222.

L192 Should be "In all four simulations, most of the region is."

**Response:**

285    Revised. Please see line 224.

L193-195 See my general comment and consider removing these lines.

**Response:**

As stated in our response to the general comment, we need to retain these lines.

290

L205 some areas of southern China seem wetter than SSP2-45, and drier in the west for G6. See general comment. Even if statistically significant this still doesn't categorically show that SAI "effectively mitigates the increase" it demonstrates that it works for this set up in this model.

**Response:**

295    Thank you for pointing that out. The sentence has been revised to "This indicates SAI somewhat mitigates the increase in mean precipitation from the high GHG SSP5-8.5 scenario to the medium GHG SSP2-4.5 scenario across most of China. It is important to note that this finding is based on a single model, and future studies could validate these results using multiple models." in lines 245-248.

300 L212 space between R95p, for

**Response:**

Thank you, the space has been added.

L220 Should be SSP5-8.5

305 **Response:**

Thank you. It has been corrected.

L223 Why is this noteworthy? It seems to be the same location in all four panels.

**Response:**

310 Thank you for pointing that out. 'Noteworthy' was not accurately used; we have revised it to 'observed' in line 276.

L227 Presumably the combination of water vapour and south-westerly winds brings more frequent ARs as you demonstrated?

315 **Response:**

We agree that the increases in the north-eastward water vapour transport can facilitate the increase in AR frequency and partially the increase in extreme precipitation. However, not all the occurrences of extreme precipitation and their future changes can be attributed to ARs, particularly in southern China where precipitation extremes are linked to different types of weather systems (e.g. tropical cyclones and

320 easterly waves). Thus, A short statement relevant to this has been added to lines 285-287. We also stated that "the synoptic mechanisms behind these signals of increase still require further investigation" in lines 287-288.

L228 Spelling check all UK or all US throughout.

325 **Response:**

Thank you. The spelling has been checked throughout the paper to make sure the language is UK English.

L231 is this a statistically significant decrease with respect to SSP5-8.5?

330 **Response:**

Thank you for pointing out this issue. Strictly speaking, this sentence may not be accurate and has been removed.

 This would make more sense with SSP2-4.5 - G6sulfur, SSP2-45 - G6solar, and G6sulfur-G6solar because the description is about how areas are wetter than SSP2-45; and your reference point is whether the difference between SSP-2.45 and G6 are statistically distinguishable and what the consequences might be. Update the narrative to match the figures as suggested.

**Response:**

As our response to the general comment, as well as your suggestion, we have included 'G6solar-SSP2-45' and updated the description to match the figures. Please see lines 298-330.

 Change "effectively mitigates"

**Response:**

The sentence has been revised to "G6sulfur mitigates RX5day under SSP5-8.5, particularly in the eastern and south-western regions under UKESM1." Please see lines 317-318.

Table 3 as noted above, more compelling would be the difference in results between SSP2-4.5 and G6. Can SAI hold the space without having other consequences?

**Response:**

As suggested, we have included the difference in results between SSP2-4.5 and G6 (Table S3 and Table S4) in the Supplement. We have also added some related discussions, such as in lines 355-360 and lines 517-519.

 You still have not included in the article how the CDFs were established for each region!

**Response:**

Thanks for pointing out this issue. We have added a more detailed explanation: "To establish the CDFs for an extreme precipitation index, firstly, the yearly mean extreme precipitation index was calculated over the three ensemble model members and the 30 years at each grid point. Then, for each region and the whole of China, the empirical CDFs (ECDFs) of the extreme precipitation index were statistically established as histograms, based on the values over all the grids. To achieve a smooth

representation of the distribution, we applied a Gaussian smoothing technique. By doing so, we were able to obtain smoothed representations of the empirical distributions, which provided clearer insights into the underlying patterns of the data." Please see lines 180-185.

365 L286 Four G6 scenarios?

**Response:**

Thank you for pointing this out; we have corrected it to "four simulations."

**Reference**

370 Haywood, J. M., Jones, A., Jones, A. C., Halloran, P., and Rasch, P. J.: Climate intervention using marine cloud brightening (MCB) compared with stratospheric aerosol injection (SAI) in the UKESM1 climate model, Atmospheric Atmos. Chem. Phys, 23., 15305-15324, https://doi.org/10.5194/acp-23-15305-2023, 2023.

Liang, J. and Haywood, J.: Future changes in atmospheric rivers over East Asia under stratospheric aerosol intervention, Atmos. Chem. Phys., 23, 1687-1703, https://doi.org/10.5194/acp-23-1687-2023, 2023.

375 Robock, A., Jerch, K., & Bunzl, M.: 20 reasons why geoengineering may be a bad idea. Bull. Atom. Sci, 64, 14–59, https://doi.org/10.1080/00963402.2008.11461140, 2008.

Tilmes, S., Visioni, D., Jones, A., Haywood, J., Séférian, R., Nabat, P., Boucher, O., Bednarz, E. M., and Niemeier, U.: Stratospheric ozone response to sulfate aerosol and solar dimming climate interventions based on the G6 Geoengineering Model Intercomparison Project (GeoMIP) simulations, Atmos. Chem. Phys., 22, 4557–4579, https://doi.org/10.5194/acp-

380 22-4557-2022, 2022.

Trisos, C. H., Amatulli, G., Gurevitch, J., Robock, A., Xia, L., and Zambri, B.: Potentially dangerous consequences for biodiversity of solar geoengineering implementation and termination, Nat. Ecol. Evol., 2, 475-482, https://doi.org/10.1038/s41559-017-0431-0, 2018.

Wells, A. F., Henry, M., Bednarz, E. M., MacMartin, D. G., Jones, A., Dalvi, M., and Haywood, J. M.: Identifying climate

385    impacts from different Stratospheric Aerosol Injection strategies in UKESM1, Earth's Future., 12, e2023EF004358, https://doi.org/10.1029/2023EF004358, 2024.

---

## Author Response (AR3)

**Dear Dr. Simone Tilmes,**

Thank you for accepting our manuscript for publication. The "Short summary" has been corrected to English.

Sincerely yours,

Ou Wang (first author)